# Inference from Quantized Data via Normal Variance-Mean Mixtures

**Chenyu Gao** [1]  **Zhexian Yang** [1]  **Ziping Zhao** [1]

## Abstract

Inference from quantized data has received significant attention in recent years due to its broad applications in machine learning and signal processing. Existing likelihood-based approaches are often restricted to Gaussian assumptions or low-bit quantization settings, limiting modeling flexibility and robustness under complex data distributions. In this work, we study inference from quantized observations under the general normal variance-mean mixture (NVMM) framework, which encompasses distributions including Gaussian, $t$, generalized hyperbolic skew-$t$, and generalized hyperbolic distributions. Optimization under the NVMM framework is challenging because the underlying likelihood function involves multidimensional integrals that are difficult to evaluate due to multidimensional quantization and latent mixture variables. To address this difficulty, we propose an expectation conditional maximization (ECM) algorithm with latent-variable augmentations for both quantization and mixture modeling. By leveraging the conditional Gaussian structure of the NVMM family, the proposed method admits closed-form updates for all model parameters at each iteration, leading to an efficient and tractable optimization procedure. We further establish global linear convergence guarantees for the proposed ECM algorithm. Beyond basic parameter estimation, the proposed framework naturally extends to several structured learning and recovery tasks under the NVMM framework, including quantized regression, matrix completion, compressive sensing, and covariance estimation. Numerical experiments demonstrate the effectiveness and robustness of the proposed framework across a variety of quantized inference problems.

[1]School of Information Science and Technology, ShanghaiTech University, Shanghai, China. Correspondence to: Ziping Zhao <zipingzhao@shanghaitech.edu.cn>.

*Proceedings of the 43rd International Conference on Machine Learning*, Seoul, South Korea. PMLR 306, 2026. Copyright 2026 by the author(s).

## 1. Introduction

Quantization, which represents data using a finite number of bits, provides an efficient mechanism for data acquisition, storage, transmission, and computation in resource-limited settings (Roberts, 1962; Gray & Neuhoff, 2002; Widrow & Kollár, 2008; Kipnis & Reeves, 2021). By reducing the precision of continuous-valued observations, quantization enables scalable processing of large-scale and high-dimensional data while alleviating communication, memory, and hardware burdens. As a result, quantized observations arise naturally in a broad range of machine learning and signal processing applications. Representative examples include matrix recovery in recommendation systems (Davenport et al., 2014; Cai & Zhou, 2013; Bhaskar, 2016; Bottegal & Suykens, 2017; Gao et al., 2018), sparse signal recovery in compressed sensing (Boufounos & Baraniuk, 2008; Zymnis et al., 2009; Plan & Vershynin, 2013; Ai et al., 2014), and statistical learning tasks such as quantized regression and subspace estimation (Mayne, 1967; Gyorfi & Wegkamp, 2008; Chen et al., 2023; Chi & Fu, 2017; Dirksen et al., 2025). Quantization also plays a central role in wireless communications and sensing applications, including channel estimation, array detection, radar signal processing, spectrum sensing, and networked sensing (Choi et al., 2016; Stöckle et al., 2016; Plabst et al., 2018; Ren & Li, 2017; Ameri et al., 2019; Jin et al., 2020; Bar-Shalom & Weiss, 2002; Lu et al., 2024; Yang et al., 2025; Chi & Fu, 2017). In many of these applications, the objective is not merely to recover the quantized observations themselves, but rather to infer latent model structures and statistical quantities underlying the data generation process.

These applications can often be formulated through probabilistic modeling of latent quantities and structures from quantized observations. Depending on the application, the quantities of interest may correspond to low-order statistical quantities such as the mean (Papadopoulos et al., 2001; Dabeer & Masry, 2008) for distributed detection (Ribeiro & Giannakis, 2006a;b; Fang & Li, 2008), and the covariance matrix (Van Vleck & Middleton, 1966; Gray & Stockham, 1993; Liu & Lin, 2021; Dirksen et al., 2022; Xiao et al., 2023) for direction-of-arrival estimation (Eamaz et al., 2022; 2023; Bar-Shalom & Weiss, 2002), wireless power estimation (Mo et al., 2017), and spectrum sensing

(Yang et al., 2025). More generally, latent structures may also arise through low-rank, sparse, or low-dimensional representations, leading to structured inference problems such as matrix completion, compressive sensing, and subspace recovery.

However, quantization inevitably discards part of the information contained in the original observations, making inference from quantized data both challenging and practically important. In this paper, we study probabilistic inference of latent quantities and structures from quantized observations. Let $\boldsymbol{x} \in \mathbb{R}^d$ denote a random signal or latent variable of interest, and let $\boldsymbol{y}$ denote its quantized observation generated through

$$\boldsymbol{y} = \mathcal{Q}(\boldsymbol{x}), \tag{1}$$

where $\mathcal{Q} : \mathbb{R}^d \to \mathcal{K}^d$ denotes a quantization function, or quantizer, and $\mathcal{K} = \{k_1, \ldots, k_e\}$ is a finite quantization alphabet containing $e$ levels. Specifically, the $i$-th entry of $\mathcal{Q}(\boldsymbol{x})$ is defined as

$$[\mathcal{Q}(\boldsymbol{x})]_i = k_l, \quad \text{if } x_i \in [\tau_{l-1}, \tau_l), \tag{2}$$

where the quantization intervals satisfy

$$\bigcup_{l=1}^{e} [\tau_{l-1}, \tau_l) = \mathbb{R},$$

with $\tau_0 = -\infty$ and $\tau_e = \infty$. The number of bits required to represent each quantized value is given by $\log_2 e$. When $\log_2 e$ is small, the quantizer is referred to as coarse quantization. A particularly important special case is one-bit quantization, corresponding to $e = 2$. Under the additional conditions $k_1 = -1$, $k_2 = 1$, and $\tau_1 = 0$, the quantizer $\mathcal{Q}$ reduces to the element-wise signum function.

In this paper, we study maximum likelihood inference from quantized observations $\boldsymbol{y}$ under the normal variance-mean mixture (NVMM) framework. We assume that the latent signal $\boldsymbol{x}$ follows a general NVMM distribution (Barndorff-Nielsen et al., 1982; Barndorff-Nielsen, 1997; Polson & Scott, 2013), a flexible probabilistic family capable of modeling heavy tails, skewness, and heterogeneous covariance structures through its location, skewness, and scatter parameters. The NVMM framework encompasses several widely used distributions, including Gaussian, $t$, generalized hyperbolic skew-$t$ (GHST), and generalized hyperbolic (GH) distributions. Unlike existing approaches that are primarily restricted to one-bit quantization or Gaussian assumptions, the proposed framework accommodates arbitrary multi-bit quantization functions and general distributions within the NVMM family.

To solve the resulting maximum likelihood problem, we develop an expectation conditional maximization (ECM)

algorithm that jointly handles multidimensional quantization and latent mixture variables through suitable latent-variable augmentations. The proposed method alternates between an expectation step (E-step), where a surrogate objective function is constructed via Jensen's inequality, and a conditional maximization step (CM-step), where the surrogate is optimized with respect to different parameter blocks while keeping the remaining parameters fixed. By leveraging the conditional Gaussian structure of the NVMM family, the proposed ECM algorithm admits closed-form updates for the location, skewness, and scatter parameters. We further establish global linear convergence guarantees for the proposed algorithm. Beyond parameter estimation under the NVMM framework, the proposed probabilistic inference framework naturally extends to several structured learning and recovery problems, including quantized regression, matrix completion, compressive sensing, and covariance estimation. Extensive numerical experiments demonstrate the effectiveness and robustness of the proposed framework across a variety of quantized inference tasks.

## 2. Problem Formulation

A random vector $\boldsymbol{x} \in \mathbb{R}^d$ following an NVMM model can be represented as

$$\boldsymbol{x} = \boldsymbol{\mu} + z\boldsymbol{\xi} + (z\boldsymbol{\Sigma})^{\frac{1}{2}} \boldsymbol{\epsilon}, \tag{3}$$

where $\boldsymbol{\mu}$ is the location parameter, $\boldsymbol{\xi}$ is the skewness parameter, $\boldsymbol{\Sigma}$ is the scatter parameter, $z$ is a nonnegative random variable with density function $p(z)$, and $\boldsymbol{\epsilon}$ is a standard normal random vector with zero mean and identity covariance matrix, independent of $z$. Equivalently, conditional on $z$, the random vector $\boldsymbol{x}$ follows a Gaussian distribution with mean $\boldsymbol{\mu} + z\boldsymbol{\xi}$ and covariance matrix $z\boldsymbol{\Sigma}$. The conditional density function of $\boldsymbol{x}$ given $z$ is therefore

$$\begin{aligned} p(\boldsymbol{x} \mid z; \boldsymbol{\theta}) &= \frac{1}{(2\pi)^{\frac{d}{2}} \det(z\boldsymbol{\Sigma})^{\frac{1}{2}}} \\ &\quad \times \exp\left(-\frac{1}{2z} \|\boldsymbol{x} - \boldsymbol{\mu} - z\boldsymbol{\xi}\|_{\boldsymbol{\Sigma}^{-1}}^2\right), \end{aligned} \tag{4}$$

where

$$\boldsymbol{\theta} = (\boldsymbol{\mu}, \boldsymbol{\xi}, \boldsymbol{\Sigma})$$

collects the model parameters[1], and $\|\boldsymbol{x}\|_{\boldsymbol{A}}^2 = \boldsymbol{x}^\top \boldsymbol{A} \boldsymbol{x}$. Following the quantization model in (1), the density function

---

[1]In general, the distribution of $z$ in an NVMM model may also depend on additional parameters, which are assumed to be known and are not included in the estimation procedure throughout this paper for simplicity.

of the quantized observation $\boldsymbol{y}$ is given by

$$
\begin{aligned}
p(\boldsymbol{y}; \boldsymbol{\theta}) &= \int_{\mathcal{Q}^{-1}(\boldsymbol{y})} p(\boldsymbol{x}; \boldsymbol{\theta}) \mathrm{d}\boldsymbol{x} \\
&= \int_{\mathcal{Q}^{-1}(\boldsymbol{y})} \int_0^\infty p(\boldsymbol{x} \mid z; \boldsymbol{\theta}) p(z) \mathrm{d}z \mathrm{d}\boldsymbol{x},
\end{aligned} \tag{5}
$$

where $\mathcal{Q}^{-1}(\boldsymbol{y})$ denotes the preimage of the quantized observation $\boldsymbol{y}$ under the quantizer $\mathcal{Q}$. In particular, $\mathcal{Q}^{-1}(\boldsymbol{y})$ forms a hyper-rectangle in $\mathbb{R}^d$, whose projection onto the $i$-th coordinate corresponds to the quantization interval associated with $[\boldsymbol{y}]_i$.

Given $n$ independent and identically distributed observations $\boldsymbol{y}_1, \ldots, \boldsymbol{y}_n$, the maximum likelihood (ML) estimation problem is formulated as

$$
\max_{\boldsymbol{\theta}} \quad L(\boldsymbol{\theta}) = \sum_{t=1}^n \log p(\boldsymbol{y}_t; \boldsymbol{\theta}). \tag{6}
$$

Even evaluating the likelihood function in (5) is generally challenging, since it involves multidimensional integration induced jointly by the quantization region $\mathcal{Q}^{-1}(\boldsymbol{y})$ and the latent mixing variable $z$. In the special case where $d = 1$, $z$ is deterministic, and one-bit quantization is applied, the resulting model reduces to a univariate Gaussian setting for which closed-form estimators are available (Ribeiro & Giannakis, 2006b). In contrast, for multivariate quantization and general NVMM models, the likelihood function typically does not admit a closed-form expression, making the associated optimization problem substantially more difficult.

# 3. Related Work

Building upon the ML optimization framework in (6), a variety of downstream tasks can be addressed. Among various research directions, two fundamental lines of inquiry concern the estimation of the mean and covariance of $\boldsymbol{x}$, which we adopt as our starting point. Furthermore, ML-based parameter estimation can be extended to the estimation of structured probabilistic models, which constitute standard practice in machine learning. In the following, we demonstrate applications of the proposed model to problems such as quantized regression, quantized matrix completion, and quantized compressive sensing.

## 3.1. Quantized Mean Estimation

In (Papadopoulos et al., 2001), in order to estimate signals from wireless sensor networks, an optimization problem is formulated for estimating the mean under a one-dimensional Gaussian distribution assumption, based on multi-bit quantized data. To address this optimization problem, the authors propose an ECM-based algorithm. Subsequently, (Ribeiro & Giannakis, 2006b) focuses on the

specific scenario of one-bit quantization under a Gaussian distribution and derives a closed-form expression for mean estimation in this case. Further, (Fang & Li, 2008) extends this line of research by introducing an adaptive threshold into the one-bit quantization function, thereby improving mean estimation accuracy. In addition, (Dabeer & Masry, 2008) generalizes the problem to the multidimensional setting. (Ribeiro & Giannakis, 2006a) extends the model distribution to the generalized Gaussian distribution under the one-dimensional assumption.

## 3.2. Quantized Linear Regression

Quantized linear regression characterizes the mapping of a feature vector $\boldsymbol{x}$ to a discrete output $y$, modeled as

$$
y = \mathcal{Q}(\boldsymbol{\beta}^\top \boldsymbol{x} + \epsilon).
$$

In this formulation, the linear component $\boldsymbol{\beta x}$ is perturbed by an error term $\epsilon$, the primary objective is to estimate the parameter $\boldsymbol{\beta}$, and the error term $\epsilon$ is treated as a zero-mean random variable. When $\epsilon$ is assumed as a Gaussian variable, an EM-based algorithm has been proposed (Finesso et al., 1999). However, as the Gaussian distribution assumption exhibits poor performance when data contain outliers, subsequent studies (Chen et al., 2023; 2024) center on robust estimation methods for heavy-tailed data. Furthermore, robust estimation in classical linear regression accounts for both asymmetric and heavy-tailed noise (Fan et al., 2021). For instance, the noise term $\epsilon$ can be modeled using a GH distribution (Kim & Browne, 2024), which captures both of these characteristics simultaneously. In this paper, our proposed algorithm facilitates parameter estimation using quantized measurements under the assumption of a normal variance-mean mixture. In the normal variance-mean mixture, the $t$ distribution is heavy-tailed, and both the GHST and GH distributions exhibit heavy tails and asymmetry, this modeling framework enables robust estimation. In this case, the model becomes

$$
y = \mathcal{Q}\left(\boldsymbol{\beta}^\top \boldsymbol{x} + (z - \mathsf{E}[z])\xi + (z\sigma)^{\frac{1}{2}} \epsilon\right). \tag{7}
$$

## 3.3. Quantized Probabilistic Matrix Completion

The goal of the low-rank matrix completion problem (Candes & Recht, 2012) is to recover an unknown low-rank matrix $\boldsymbol{M} \in \mathbb{R}^{d_1 \times d_2}$ from an observed, yet incomplete, matrix. Let $\boldsymbol{X}$ denote a matrix whose entries are drawn from a normal variance-mean mixture distribution. We use $\mu_{ij}$ and $x_{ij}$ to represent the $(i, j)$-th entries of $\boldsymbol{M}$ and $\boldsymbol{X}$, respectively. Based on the normal variance-mean model, the relationship between $\mu_{ij}$ and $x_{ij}$ is given by

$$
x_{ij} = \mu_{ij} + z^{\frac{1}{2}} \sigma \epsilon, \tag{8}
$$

where $\mu_{ij}$ serves as the mean of $x_{ij}$. In the quantization scenario (Davenport et al., 2014), however, the matrix $\boldsymbol{X}$

is not directly accessible; instead, we observe its quantized counterpart $\boldsymbol{Y} = \mathcal{Q}(\boldsymbol{X})$. Moreover, the entire matrix $\boldsymbol{Y}$ is not available for observation. Let $\mathcal{O}$ denote the index set of the observed entries; specifically, if $(i,j) \in \mathcal{O}$, then $Y_{ij}$ is observed, otherwise $Y_{ij}$ is missing. Our goal is to recover $\boldsymbol{M}$ from incomplete $\boldsymbol{Y}$. The study of matrix completion was popularized following the Netflix Million Dollar Challenge, which posed the task: accurately predicting the values of those entries with a user–item matrix in which entries represent item ratings.

To recover $\boldsymbol{M}$ from quantized and corrupted observations, a seminal study introduced an ML framework for one-bit low-rank matrix completion (Davenport et al., 2014). Building on this work, random dithering was incorporated into the quantization function to improve recovery performance (Eamaz et al., 2024). Both approaches employed the nuclear norm to relax the low-rank constraint and used projected gradient descent for optimization. An alternative strategy reformulated the low-rank constraint via matrix factorization, followed by projected gradient descent (Bhaskar & Javanmard, 2015). Similarly, factorization combined with a majorization–minimization method was used to derive a surrogate objective, which was solved via the Gauss–Newton method (Liu et al., 2025). The framework was further extended from one-bit to multi-bit quantization using low-rank factorization together with projected gradient descent (Bhaskar, 2016). The former works (Davenport et al., 2014; Bhaskar & Javanmard, 2015; Bhaskar, 2016) are based on the Gaussian distribution assumption. However, observed data often exhibit heavy-tailed characteristics, which are induced by the presence of outliers (Chen et al., 2024). To analyze such data, robust modeling techniques are frequently employed (Shen et al., 2019; Chen et al., 2024). The normal variance-mean mixture class encompasses numerous classical heavy-tailed distributions that are extensively utilized in robust modeling. Consequently, this study employs the $t$ distribution and the symmetric GH distribution to perform robust estimation for the quantized matrix completion problem.

### 3.4. Quantized Compressive Sensing

One-bit compressive sensing ((Boufounos & Baraniuk, 2008)) aims to estimate a sparse signal lying in a known measurement subspace based on observed quantized data (Jacques et al., 2013; Chen et al., 2024). Existing methods for this problem generally fall into three categories. The first two primarily focus on 1-bit quantization without incorporating additive noise into the original measurements (Li et al., 2018). The first category, termed regularizer-class algorithms (Laska et al., 2011), introduces additional regularization terms to the classical compressive sensing recovery problem to enforce consistency between the sparse signal and the measurements. The second category, known

as penalty-class algorithms (Yan et al., 2012), models sign flips between quantized and recovered measurements as penalty terms. The third category assumes the presence of additive noise in the original measurements (Zymnis et al., 2009; Knudson et al., 2016). Our proposed model is developed as a further extension of the third approach. Similar to the quantized matrix completion problem, robust estimation represents a significant modeling framework in quantized compressed sensing (Plan & Vershynin, 2013; Dirksen & Mendelson, 2021; Jung et al., 2021). While previous research primarily focuses on characterizing the statistical errors of algorithms based on the properties of heavy-tailed data, this study utilizes the $t$ and symmetric GH distributions to derive efficient algorithms for robust estimation. We employ a normal variance-mean mixture model for noise modeling. Let $\boldsymbol{\Phi} \in \mathbb{R}^{d_1 \times d_2}$ be the known measurement matrix, and $\boldsymbol{\vartheta} \in \mathbb{R}^{d_2}$ be the sparse signal to be estimated. Based on the normal variance-mean model, the quantized compressed sensing model is given by

$$\boldsymbol{y} = \mathcal{Q}(\boldsymbol{x}), \quad \boldsymbol{x} = \boldsymbol{\Phi}\boldsymbol{\vartheta} + z^{\frac{1}{2}}\sigma\boldsymbol{\epsilon}, \tag{9}$$

where $\boldsymbol{\Phi}\boldsymbol{\vartheta}$ serves as the location parameter of $\boldsymbol{x}$.

### 3.5. Quantized Covariance/Correlation Estimation

In (Van Vleck & Middleton, 1966), the authors investigate the estimation of a correlation matrix from one-bit quantized zero-mean Gaussian measurements, based on the arcsine law (Lévy, 1940). However, in such one-bit zero-mean settings, the individual variance of each dimension cannot be determined, which prevents recovery of the full covariance matrix. To estimate the variance, the "dithering technique" is introduced, which is to set a threshold in the signum function (Liu & Lin, 2021). Based on the dithering technique, the variance can be obtained in closed form (Fang & Li, 2008), but the correlation matrix should be solved analytically. Consequently, subsequent studies (Dirksen et al., 2022; Eamaz et al., 2023; Xiao et al., 2023; Liu & Chou, 2025) have proposed various strategies for correlation estimation. The estimation approaches for correlation can be classified into two categories. The first category relies on the correlation coefficient function of the one-bit Gaussian distribution, with analyses usually restricted to pairwise interactions between dimensions in the multivariate setting. Since this function is generally computationally intractable, different approximate functions have been proposed to compute the correlation coefficient (Eamaz et al., 2023; Xiao et al., 2023; Liu & Chou, 2025). In (Eamaz et al., 2022; 2023), the authors introduce a known, time-varying threshold for the signum function and propose a modified arcsine law to express correlation coefficients as integrals, which are then approximated using Gauss–Legendre quadrature. The method in (Xiao et al., 2023) applies ML estimation to compute cor-

relation coefficients, which is equivalent to iteratively evaluating their approximate functions. The work (Liu & Chou, 2025) represents the correlation coefficient function as an infinite series via the one-bit Hermite law (Liu & Lin, 2021) and then approximates it using harmonic approximation. The second category (Dirksen et al., 2022; Chen et al., 2024) directly employs the sample covariance matrix computed using multiple thresholds. These methods are not restricted to one-bit quantization and can be generalized to multi-bit cases, though typically at the expense of reduced estimation accuracy. The studied model can also be applied to quantized structured covariance modeling, which focuses on recovering high-dimensional covariance matrices from quantized measurements by exploiting inherent geometric structures—such as Toeplitz (Liu & Lin, 2021; Eamaz et al., 2022), sparse, or low-rank (Saha et al., 2023) properties (Maly et al., 2022).

### 3.6. Quantized Graphical Modeling

Quantized graphical modeling studies the recovery and inference of conditional dependence graphs when the observed variables are discrete or arise from quantization of latent continuous processes. Model construction typically proceeds either by directly specifying discrete Markov random fields (Fang & Li, 2010), or by introducing latent Gaussian variables combined with quantization (Lai, 2016; Tavassolipour et al., 2018). Besides, several studies focus on the robust design of graphs (Nobre & Frossard, 2019; Saad et al., 2021).

## 4. The ECM Algorithm

The optimization problem in (6) is challenging due to the multiple integrals in the density function (5). Nevertheless, the ECM framework can be applied by leveraging the models in (1) and (3), where $\boldsymbol{x}$ and $z$ can be naturally treated as latent variables. In the following, we introduce an ECM procedure for (6).

### 4.1. E-step

In the E-step, we derive a surrogate function for the log-likelihood function $L(\boldsymbol{\theta})$. Based on (1), the observed signal $\boldsymbol{y}$ is conditioned on the hidden variable $\boldsymbol{x}$. Hence, given $[\boldsymbol{y}_1 \ldots, \boldsymbol{y}_n]^\top$ and the corresponding hidden variables $[\boldsymbol{x}_1 \ldots, \boldsymbol{x}_n]^\top$, we have[2]

$$
\begin{aligned}
L(\boldsymbol{\theta}) &= \sum_{t=1}^{n} \log \int_{\mathbb{R}} p(\boldsymbol{y}_t \mid \boldsymbol{x}_t; \boldsymbol{\theta}) p(\boldsymbol{x}_t; \boldsymbol{\theta}) \mathrm{d}\boldsymbol{x}_t \\
&\geq \sum_{t=1}^{n} \mathsf{E}_{\boldsymbol{x}_t|\boldsymbol{y}_t;\underline{\boldsymbol{\theta}}} \log p(\boldsymbol{y}_t, \boldsymbol{x}_t; \boldsymbol{\theta}) + \text{const.},
\end{aligned}
\tag{10}
$$

[2]Throughout this paper, underlined variables denote those whose values are given as constants.

where the inequality is based on Jensen's inequality and const. is some constant term independent of $\boldsymbol{\theta}$. Under the models specified in (1), the joint density function is given by

$$
p(\boldsymbol{y}_t, \boldsymbol{x}_t; \boldsymbol{\theta}) = p(\boldsymbol{x}_t; \boldsymbol{\theta})\delta\left(\mathcal{Q}\left(\boldsymbol{x}_t\right) - \boldsymbol{y}_t\right), \tag{11}
$$

where $\delta(\cdot)$ denotes the multivariate Dirac delta function, defined as $\delta\left(\boldsymbol{x}_t\right) = \begin{cases} 0, & \text{if } \boldsymbol{x}_t \neq \boldsymbol{0} \\ +\infty, & \text{if } \boldsymbol{x}_t = \boldsymbol{0} \end{cases}$. Since the term $\delta\left(\mathcal{Q}\left(\boldsymbol{x}_t\right) - \boldsymbol{y}_t\right)$ is independent of $\boldsymbol{\theta}$, we further obtain

$$
\begin{aligned}
L(\boldsymbol{\theta}) &\geq \sum_{t=1}^{n} \mathsf{E}_{\boldsymbol{x}_t|\boldsymbol{y}_t;\underline{\boldsymbol{\theta}}} \log p(\boldsymbol{x}_t; \boldsymbol{\theta}) + \text{const.} \\
&= \sum_{t=1}^{n} \mathsf{E}_{\boldsymbol{x}_t|\boldsymbol{y}_t;\underline{\boldsymbol{\theta}}} \log \int_0^\infty p(\boldsymbol{x}_t, z_t; \boldsymbol{\theta}) \mathrm{d}z_t + \text{const.}
\end{aligned}
$$

Regarding $z$ as a hidden variable, applying Jensen's inequality leads to

$$
L(\boldsymbol{\theta}) \geq \sum_{t=1}^{n} \mathsf{E}_{\boldsymbol{x}_t|\boldsymbol{y}_t;\underline{\boldsymbol{\theta}}} \left[ \mathsf{E}_{z_t|\boldsymbol{x}_t;\underline{\boldsymbol{\theta}}} \log p(\boldsymbol{x}_t, z_t; \boldsymbol{\theta}) \right] + \text{const..}
\tag{12}
$$

Since $z \mid \boldsymbol{x}$ is independent of $\boldsymbol{x} \mid \boldsymbol{y}$, we have $p(\boldsymbol{x} \mid \boldsymbol{y}; \boldsymbol{\theta})p(z \mid \boldsymbol{x}; \boldsymbol{\theta}) = p(\boldsymbol{x}, z \mid \boldsymbol{y}; \boldsymbol{\theta})$. Given the $p(\boldsymbol{x} \mid z; \boldsymbol{\theta})$ in (4), the surrogate function $S(\boldsymbol{\theta}; \underline{\boldsymbol{\theta}})$ in (12) can be further expressed as follows:

$$
\begin{aligned}
S(\boldsymbol{\theta}; \underline{\boldsymbol{\theta}}) = \sum_{t=1}^{n} \Bigg[ & \mathsf{E}_{z_t|\boldsymbol{y}_t;\underline{\boldsymbol{\theta}}} \left[\log p(z_t)\right] - \frac{1}{2}\log\det\boldsymbol{\Sigma} \\
& -\frac{1}{2}\mathrm{Tr}\left( \boldsymbol{U}_t - 2\boldsymbol{v}_t\boldsymbol{\mu}^\top - 2\boldsymbol{w}_t\boldsymbol{\xi}^\top + 2\boldsymbol{\mu}\boldsymbol{\xi}^\top \right. \\
& \left. +\iota_t\boldsymbol{\mu}\boldsymbol{\mu}^\top + \zeta_t\boldsymbol{\xi}\boldsymbol{\xi}^\top\right) \boldsymbol{\Sigma}^{-1} \Bigg] + \text{const.}
\end{aligned}
\tag{13}
$$

where

$$
\boldsymbol{U}_t = \mathsf{E}_{\boldsymbol{x}_t,z_t|\boldsymbol{y}_t;\underline{\boldsymbol{\theta}}}[z_t^{-1}\boldsymbol{x}\boldsymbol{x}^\top], \quad \boldsymbol{v}_t = \mathsf{E}_{\boldsymbol{x}_t,z_t|\boldsymbol{y}_t;\underline{\boldsymbol{\theta}}}[z_t^{-1}\boldsymbol{x}_t],
$$
$$
\boldsymbol{w}_t = \mathsf{E}_{\boldsymbol{x}_t|\boldsymbol{y}_t;\underline{\boldsymbol{\theta}}}[\boldsymbol{x}_t], \iota_t = \mathsf{E}_{z_t|\boldsymbol{y}_t;\underline{\boldsymbol{\theta}}}[z_t^{-1}], \zeta_t = \mathsf{E}_{z_t|\boldsymbol{y}_t;\underline{\boldsymbol{\theta}}}[z_t].
$$

The details for computing these expectations are given in Appendix A. In the term $\mathsf{E}_{z_t|\boldsymbol{y}_t;\underline{\boldsymbol{\theta}}}[\log p(z_t)]$, some scalar parameters (such as the shape parameter $\nu$ in $t$ distribution) are contained. In the practical implementation (Galarza et al., 2021), they are typically treated as given or estimated through the one-dimensional search.

### 4.2. CM-step

Based on the surrogate function (13), in the CM-step, we solve the following optimization problem:

$$
\max_{\boldsymbol{\theta}} \ S(\boldsymbol{\theta}; \underline{\boldsymbol{\theta}}), \tag{14}
$$

where parameters $\boldsymbol{\mu}$, $\boldsymbol{\xi}$, and $\boldsymbol{\Sigma}$ can be solved with closed-form solutions:

$$\boldsymbol{\mu} = \frac{\sum_{t=1}^{n}(\boldsymbol{v}_t - \boldsymbol{\xi})}{\sum_{t=1}^{n}\iota_t}, \qquad \boldsymbol{\xi} = \frac{\sum_{t=1}^{n}(\boldsymbol{w}_t - \boldsymbol{\mu})}{\sum_{t=1}^{n}\zeta_t},$$

$$\boldsymbol{\Sigma} = \frac{1}{n}\sum_{t=1}^{n}\Big(\boldsymbol{U}_t - 2\boldsymbol{v}_t\boldsymbol{\mu}^{\top} - 2\boldsymbol{w}_t\boldsymbol{\xi}^{\top}$$
$$+ 2\boldsymbol{\mu}\boldsymbol{\xi}^{\top} + \iota_t\boldsymbol{\mu}\boldsymbol{\mu}^{\top} + \zeta_t\boldsymbol{\xi}\boldsymbol{\xi}^{\top}\Big). \tag{15}$$

**Remark 1.** *In the context of quantization model estimation, a prototypical scenario is the one-bit Gaussian case (i.e., $e = 2$ and $z$ is a constant). However, this model suffers from an inherent identifiability issue: for each dimension $i = 1, \ldots, d$, only the ratio $\frac{\mu_i}{\sigma_i^2}$ can be determined, rather than the individual parameters. Hence, the estimated values of $\mu_i$ and $\sigma_i^2$ differ from their desirable values by an arbitrary scaling factor. To resolve this, existing estimation methods in the one-bit Gaussian setting typically fix one parameter (either the location vector or the scatter matrix) to identify the other. Moreover, the same ambiguity persists under one-bit quantization whenever $\boldsymbol{x}$ follows any elliptical distribution.*

## 5. Convergence Analysis

In this section, we analyze the convergence properties of the proposed ECM algorithm. Based on (15), all parameters admit unique closed-form updates at each step. Consequently, starting from any feasible point, our ECM algorithm is guaranteed to converge to a stationary point in the feasible set (McLachlan & Krishnan, 2008). We first establish the convexity of each parameter individually.

**Theorem 2.** *The optimization problem (6) is strictly concave with respect to each of the parameters $\boldsymbol{\mu}$, $\boldsymbol{\xi}$, and $\boldsymbol{\Sigma}$.*

Based on Theorem 2, the individual convergence rates of the three parameters are given in the following.

**Proposition 3.** *Denote the parameters in the $k$-th iteration as $\boldsymbol{\mu}^{(k)}$, $\boldsymbol{\xi}^{(k)}$, and $\boldsymbol{\Sigma}^{(k)}$, and the stationary points as $\boldsymbol{\mu}^{\star}$, $\boldsymbol{\xi}^{\star}$, and $\boldsymbol{\Sigma}^{\star}$. One parameter is selected and iteratively updated via (15) until convergence, while the remaining two parameters are kept constant. For the convergent sequences of these three parameters, the following relationship holds for any $k$:*

$$\|\boldsymbol{\mu}^{(k+1)} - \boldsymbol{\mu}^{\star}\|_2 \le c_\mu \|\boldsymbol{\mu}^{(k)} - \boldsymbol{\mu}^{\star}\|_2,$$
$$\|\boldsymbol{\xi}^{(k+1)} - \boldsymbol{\xi}^{\star}\|_2 \le c_\xi \|\boldsymbol{\xi}^{(k)} - \boldsymbol{\xi}^{\star}\|_2,$$
$$\|\boldsymbol{\Sigma}^{(k+1)} - \boldsymbol{\Sigma}^{\star}\|_{\mathsf{F}} \le c_\Sigma \|\boldsymbol{\Sigma}^{(k)} - \boldsymbol{\Sigma}^{\star}\|_{\mathsf{F}},$$

*where*

$$c_\mu = \max_{\boldsymbol{\theta}} \left\| \frac{\sum_{t=1}^{n} \boldsymbol{\Sigma}^{-1}\mathsf{Cov}_{\boldsymbol{x}_t,z_t|\boldsymbol{y}_t;\boldsymbol{\theta}}\left[z_t^{-1}\boldsymbol{x}_t\right]}{\sum_{t=1}^{n} \mathsf{E}_{z|\boldsymbol{y}_t;\boldsymbol{\theta}}[z_t^{-1}]} \right\|_2 \in (0,1),$$

$$c_\xi = \max_{\boldsymbol{\theta}} \left\| \frac{\sum_{t=1}^{n} \boldsymbol{\Sigma}^{-1}\mathsf{Cov}_{\boldsymbol{x}_t,z_t|\boldsymbol{y}_t;\boldsymbol{\theta}}\left[\boldsymbol{x}_t - z_t\boldsymbol{\xi}\right]}{\sum_{t=1}^{n} \mathsf{E}_{z_t|\boldsymbol{y}_t;\boldsymbol{\theta}}[z_t]} \right\|_2 \in (0,1),$$

$$c_\Sigma = \max_{\boldsymbol{\theta}} \left\| \frac{1}{2n}\sum_{t=1}^{n}\left(\boldsymbol{\Sigma}^{-1} \otimes \boldsymbol{\Sigma}^{-1}\right) \right.$$
$$\left. \mathsf{Cov}_{\boldsymbol{x}_t,z_t|\boldsymbol{y}_t;\boldsymbol{\theta}}\left[z_t^{-1}\mathrm{vec}\left(\boldsymbol{x}_t\boldsymbol{x}_t^{\top}\right)\right] \right\|_2 \in (0,1).$$

Based on the result in Proposition 3, it follows that quotient linear convergence rates are achieved for each parameter when updated independently. For the global convergence rate, we introduce a result.

Denote $\boldsymbol{\theta}^{\star} = \left[\boldsymbol{\mu}^{\star\top}, \boldsymbol{\xi}^{\star\top}, \mathrm{vec}\left(\boldsymbol{\Sigma}^{\star}\right)^{\top}\right]^{\top}$ as a stationary point of the optimization problem in (6). According to (Meng & Rubin, 1994), the global convergence rate of an ECM algorithm is the maximum of the component-wise rates of convergence. Then, we establish that the proposed ECM algorithm converges globally at a linear rate, as detailed in the following result.

**Theorem 4.** *Denote the sequence generated by the ECM algorithm as $\left\{\boldsymbol{\theta}^{(k)}\right\}$. Given an initial point $\boldsymbol{\theta}^{(0)}$, there exists a constant $c_0 > 0$ such that, for any $k \ge 0$, we have*

$$\|\boldsymbol{\theta}^{(k)} - \boldsymbol{\theta}^{\star}\|_2 \le c^k \|\boldsymbol{\theta}^{(0)} - \boldsymbol{\theta}^{\star}\|_2, \tag{16}$$

*where $c = \max\{c_\mu, c_\xi, c_\Sigma\} \in (0, 1 - c_0]$.*

## 6. Experiments

In the previous sections, an algorithm for the ML estimation problem was proposed to model quantized data using a normal variance-mean mixture model. In this section, we present the practical applications of the proposed algorithm to quantized models and provide a comparative analysis between the proposed method and existing algorithms in these scenarios.

### 6.1. Convergence Analysis

Figure 1 presents a comparative analysis of the convergence rates of the algorithm across two statistical distributions: Gaussian and GH. All proposed methods achieve linear convergence rates.

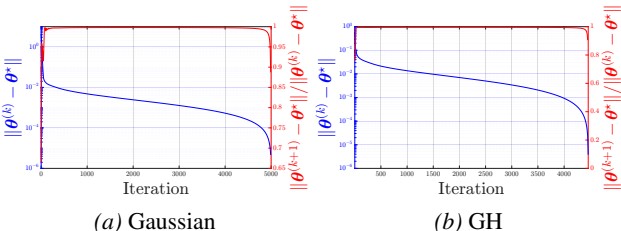

*(a)* Gaussian      *(b)* GH

*Figure 1.* Convergence curve versus iterations

## 6.2. Quantized Linear Regression

Based on model (7), the corresponding optimization problem is formulated as follows:

$$\max_{\boldsymbol{\beta}} \quad \sum_{t=1}^{n} \log p\left(y_t \mid \boldsymbol{\beta}\right). \tag{17}$$

Since the linear term $\boldsymbol{\beta}^{\top}\boldsymbol{x}_t$ in optimization problem (17) plays a role equivalent to the location parameter in the previously introduced ECM algorithm, the closed-form solution for the location parameter can be directly adapted to the linear regression framework. Specifically, the closed-form update for $\boldsymbol{\beta}$ at each ECM iteration is derived as:

$$\boldsymbol{\beta} = \left(\sum_{t=1}^{n} \iota_t \boldsymbol{x}_t \boldsymbol{x}_t^{\top}\right)^{-1} \left(\sum_{t=1}^{n} \left(v_t - \left(1 + \iota_t \mathsf{E}[z]\right)\xi\right)\boldsymbol{x}_t\right).$$

The proposed method is evaluated against existing algorithms using synthetic data. The dimension of the parameters to be estimated is set to 10, and the number of samples is 1000. Three types of noise are considered: Gaussian noise, $t$ noise (a heavy-tailed distribution with $\nu = 3$), and log-normal noise (a heavy-tailed and asymmetric distribution). The baseline algorithms for comparison include ordinary least squares (OLS) and the Gaussian EM algorithm; for OLS, each quantized observation is replaced by the representative value of its corresponding quantization interval. As illustrated in Figure 2, OLS and the Gaussian EM algorithm perform poorly under heavy-tailed noise, whereas the GHST and GH ECM algorithms exhibit significantly better performance in the presence of asymmetric noise.

## 6.3. Quantized Probabilistic Matrix Completion

Given (8) and $\boldsymbol{Y} = \mathcal{Q}(\boldsymbol{X})$, the ML estimation problem for $\boldsymbol{M}$ is given by

$$\max_{\boldsymbol{M}} \quad \sum_{(i,j)\in\mathcal{O}} \log p\left(y_{ij} \mid \mu_{ij}\right) \tag{18}$$
$$\text{s.t.} \quad \text{rank}(\boldsymbol{M}) \leq r.$$

By applying a low-rank factorization to the matrix $\boldsymbol{M}$, we express it as $\boldsymbol{M} = \boldsymbol{A}\boldsymbol{B}^{\top}$, where $\boldsymbol{A} \in \mathbb{R}^{d_1 \times r}$ and

$\boldsymbol{B} \in \mathbb{R}^{d_2 \times r}$. Through the E-step in our proposed ECM algorithm, we have the surrogate function of (18) as

$$\sum_{(i,j)\in\mathcal{O}} \left(\boldsymbol{a}_i \boldsymbol{b}_j^{\top} - e_{ij}\right)^2, \tag{19}$$

where $\boldsymbol{a}_i$ and $\boldsymbol{b}_i$ are $i$-th row of the matrix $\boldsymbol{A}$ and $\boldsymbol{B}$, respectively, and $e_{ij} = \frac{v_{ij} - \xi}{\iota_{ij}}$. The details of obtaining the surrogate (19) are given in the Appendix E.1. Then we can use the ECM algorithm to alternately update $\boldsymbol{A}$ and $\boldsymbol{B}$.

For the experimental design, the MovieLens 1M dataset (Harper & Konstan, 2015) is employed, which contains 1,000,000 movie ratings provided by 6040 users for 3952 movies, with each rating ranging from 1 to 5. All ratings are randomly partitioned into training and test sets, accounting for 40% and 60% of the data, respectively. The training set is used to predict the completed rating matrix, and the entries overlapping between the completed rating matrix and the test set are then compared to evaluate prediction accuracy and root mean square error (RMSE), which are defined in Appendix F.

Existing methods encompass classic machine learning approaches for matrix completion, including the singular value decomposition (SVD (Sarwar et al., 2000)) model, the $\ell_2$-regularized matrix factorization ($\ell_2$-regularized (Paterek, 2007)) model, and the nuclear norm minimization (nuclear norm (Cai et al., 2010)) model. Furthermore, several approaches are established upon random variable model assumptions. These include: directly modeling the Gaussian without quantization (Gaussian (Candes & Plan, 2010)); 1-bit quantization with a Gaussian distribution (1-bit Gaussian (Davenport et al., 2014)); and multi-bit quantization under a Gaussian assumption (multi-bit Gaussian (Bhaskar, 2016)). Our proposed method is regarded as an extension of the multi-bit Gaussian approach by extending the Gaussian assumption to a normal variance-mean mixture model, comprising $t$, GHST, and symmetric GH distributions.

The results are demonstrated in Table 1. As shown in the table, the proposed method consistently achieves superior performance in terms of both accuracy and RMSE. Among the evaluated models, the approach based on the symmetric GH distribution attains the best results, attributable to its flexibility and robustness in modeling skewness and heavy tails. Details regarding the experimental parameter settings are provided in the Appendix F.

## 6.4. Quantized Compressive Sensing

Given (9) and the $\ell_1$ regularization model from (Zymnis et al., 2009), we have the optimization problem

$$\max_{\boldsymbol{\vartheta}} \quad \log p\left(\boldsymbol{y} \mid \boldsymbol{\vartheta}\right) + \eta\|\boldsymbol{\vartheta}\|_1. \tag{20}$$

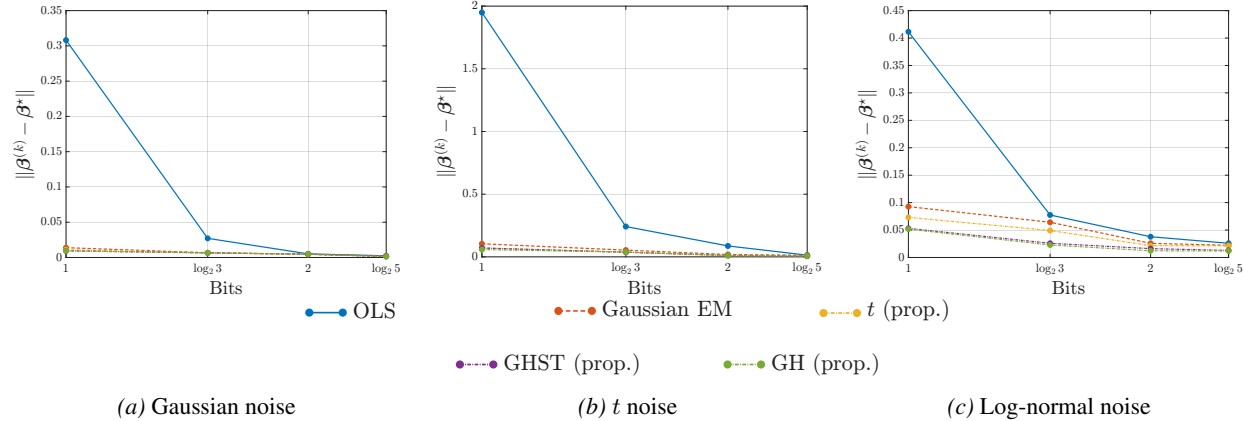

*(a)* Gaussian noise        *(b)* $t$ noise        *(c)* Log-normal noise

*Figure 2.* MSE comparisons for linear regression averaged over 10 Monte Carlo simulations.

*Table 1.* Accuracy and RMSE comparisons of matrix completion on MovieLens 1M

| Method | Accuracy | RMSE | Time (s) |
|---|---|---|---|
| SVD | 0.4261 | 0.9148 | 364.36 |
| L2 regularization | 0.4388 | 0.9355 | 165.08 |
| Nuclear norm | 0.3802 | 1.1662 | 867.39 |
| Gaussian | 0.4363 | 0.9486 | 25.26 |
| 1-bit Gaussian | 0.4217 | 0.9659 | 110.44 |
| Multi-bit Gaussian | 0.4453 | 0.9151 | 90.84 |
| Multi-bit $t$ (prop.) | 0.4464 | 0.9091 | 232.16 |
| Multi-bit GH (prop.) | **0.4503** | **0.8908** | 272.18 |

With the E-step in our proposed ECM algorithm, we have the surrogate function of (20) as

$$\frac{1}{2}\|\boldsymbol{\Phi}\boldsymbol{\vartheta} - \boldsymbol{e}\|_2^2 + \eta\|\boldsymbol{\vartheta}\|_1,$$

where the details of deriving the surrogate function are given in the Appendix E.2. We can use the FISTA algorithm (Beck & Teboulle, 2009) to solve the surrogate. Performance comparisons are conducted using synthetic data, where the measurement dimension is denoted as $d_1 = 3000$ and the 30-sparse signal dimension as $d_2 = 1000$. To evaluate algorithmic robustness, an additive noise term exhibiting both skewed and heavy-tailed characteristics is introduced to the original measurements. The existing methods included in our comparison consist of restricted step shrinkage (RSS) (Laska et al., 2011), adaptive outlier pursuit (AOP) (Yan et al., 2012), and 1-bit Gaussian MLE (Zymnis et al., 2009). For each signal-to-noise ratio (SNR) level, experiments are repeated 10 times, and the average cosine similarity (Cos Sim) and computational time are reported. The results are summarized in Table 2, which demonstrates that the 1-bit GH ECM algorithm achieves the largest average cosine similarity. The definitions of SNR and cosine similarity are given in Appendix F.

## 6.5. Quantized Correlation Estimation

In this section, we address the problem of quantized covariance estimation. Existing approaches can be broadly categorized as follows:

- non-dithered method
  - arcsine law method (Van Vleck & Middleton, 1966) (Zero Threshold),
- non-zero threshold methods
  - one-bit autocorrelation estimation (Liu & Lin, 2021) (One-bit Autocorrelation);
  - one-bit Hermite law (Liu & Chou, 2025) (One-bit Hermite Law);
  - One-bit MLE (Xiao et al., 2023) (One-bit MLE);
- dithered methods
  - uniform dithering signal (Dirksen et al., 2022) (Dithering Threshold);
  - Gaussian dithering signal (Eamaz et al., 2022) (One-bit Time-varying);
  - adaptive dithering threshold (Dirksen & Maly, 2024) (Adaptive Dithering);
- dithered methods with multi-bit problem
  - multi-bit covariance estimator (Chen et al., 2024) (Multi-bit Estimator);
  - multi-bit parameter-free estimator (Chen & Ng, 2025) (Parameter-free Estimator).

We begin by comparing the estimation accuracy of the correlation matrix among our method and existing approaches, since some previous work cannot estimate the diagonal entries of one-bit data with zero mean (Van Vleck & Middleton, 1966). We generate samples from a $t$ distribution with shape parameter $\nu = 3$ and estimate the correlation matrix using the $t$ ECM and the aforementioned methods. Figure 3 illustrates the Frobenius norm error between the estimated and true correlation matrices across varying sample sizes, under the configuration $e = 4$, $\tau = 0.3$, and $d = 3$. The proposed ECM algorithm achieves a lower MSE compared

*Table 2.* Cosine similarity comparisons of 1-bit compressive sensing under different SNR levels.

| Method | SNR = 0 dB | | SNR = -5 dB | | SNR = -10 dB | |
|---|---|---|---|---|---|---|
| | Cos Sim | Time (s) | Cos Sim | Time (s) | Cos Sim | Time (s) |
| RSS | 0.7872 | 0.2639 | 0.6598 | 0.2651 | 0.3027 | 0.2546 |
| AOP | 0.7052 | 0.2689 | 0.4041 | 0.2951 | 0.2209 | 0.2738 |
| 1-bit Gaussian | 0.8220 | 2.7704 | 0.5487 | 3.0875 | 0.2896 | 2.7202 |
| 1-bit $t$ (prop.) | 0.8396 | 2.8573 | 0.5678 | 3.1643 | 0.3031 | 3.3174 |
| 1-bit GH (prop.) | **0.8906** | 2.7252 | **0.7705** | 3.2215 | **0.5386** | 3.5325 |

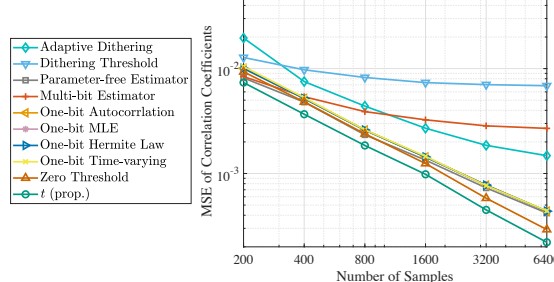

*Figure 3.* Performance comparison under the $t$ distribution.

to the existing methods. Notably, when comparing the use of different sample sizes for estimating the autocorrelation, the difference between One-bit Autocorrelation, One-bit MLE, One-bit Hermite Law, and One-bit Time-varying is quite small in the MSE.

## 7. Conclusion and Discussion

In this paper, we have proposed an ECM-based algorithm for parameter estimation in quantized models. The proposed method has exhibited broad applicability, handling problems from one-bit to multi-bit quantization and accommodating distributions ranging from Gaussian to the broader class of normal variance-mean mixtures. Experimental results have shown that our approach yields more accurate estimates than existing methods in certain cases (e.g., the one-bit Gaussian setting), while maintaining high accuracy across a wide range of extended scenarios. Furthermore, the proposed method has demonstrated strong potential in typical machine learning tasks, including quantized linear regression, quantized matrix completion, and quantized compressive sensing. An interesting future direction is to explore the statistical estimation properties of the proposed ECM algorithm for quantized maximum likelihood estimation problems.

## Impact Statement

This paper presents work whose goal is to advance the field of machine learning. There are many potential societal consequences of our work, none of which we feel must be specifically highlighted here.

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

## A. Expectation Evaluations on Specific Distribution Assumptions

From Section 4, we have established that when the random variable $x$ in model (3) belongs to the normal variance-mean mixture family, the surrogate function (13) can be derived, along with its closed-form solutions with respect to $\mu$, $\xi$, and $\Sigma$. Nevertheless, these closed-form solutions (15) depend on the expected values $U_t$, $v_t$, $w_t$, $\iota_t$, and $\zeta_t$, which are determined by the distribution of the random variable $x$. Therefore, in the following, we analyze the integrals corresponding to these expected values under various distributions of $x$, and examine the convergence properties and statistical characteristics of the proposed algorithm under specific distributional assumptions. We first consider the fundamental case of the Gaussian distribution (i.e., $z = 1$ and $\xi = 0$). Let $x \sim \mathcal{N}(\mu, \Sigma)$. In this case, the first- and second-order moments are given by (Ho et al., 2012) as

$$\mathsf{E}_{x|y;\theta}\left[x\right] = \mu + p^{-1}(y;\theta)\,\Sigma q, \tag{21}$$

$$\mathsf{E}_{x|y;\theta}\left[xx^\top\right] = \mu\mu^\top + \Sigma + p^{-1}(y;\theta)\left(2\left(\Sigma q\right)\mu^\top + \Sigma\left(H + D\right)\Sigma\right). \tag{22}$$

Let the interval $\mathcal{Q}^{-1}(y_i)$ be $[l_i, u_i]$. The $i$-th elements of vector $q$ is given by

$$q_i = p(x_i = l_i, y_{\backslash i}; \theta) - p(x_i = u_i, y_{\backslash i}; \theta), \tag{23}$$

The matrix $H$ is a matrix with all diagonal entries being zero and the $(i, j)$-th off-diagonal element being

$$H_{ij} = h(l_i, l_j) - h(l_i, u_j) - h(u_i, l_j) + h(u_i, u_j), \tag{24}$$

with

$$h(c_i, c_j) = p(x_i = c_i, x_j = c_j, y_{\backslash i,j}; \theta)).$$

The matrix $D$ is a diagonal matrix with the diagonal entries:

$$D_{ii} = \frac{l_i - \mu_i}{\sigma_i^2}p(x_i = l_i, y_{\backslash i}; \theta) - \frac{u_i - \mu_i}{\sigma_i^2}p(x_i = u_i, y_{\backslash i}; \theta) - \frac{[\Sigma H]_{ii}}{\sigma_i^2}. \tag{25}$$

Now we consider the general normal variance-mean mixture with $x = \mu + z\xi + (z\Sigma)^{\frac{1}{2}}\epsilon$. In this case, the first and second moments satisfy

$$\mathsf{E}_{x,z|y;\theta}\left[x\right] = \mathsf{E}_{z|y;\theta}\left[\mathsf{E}_{x|z,y;\theta}[x]\right],$$

and

$$\mathsf{E}_{x,z|y;\theta}\left[xx^\top\right] = \mathsf{E}_{z|y;\theta}\left[\mathsf{E}_{x|z,y;\theta}[xx^\top]\right].$$

Based on $x \mid z \sim \mathcal{N}(\mu + z\xi, z\Sigma)$ and the moments in (21) and (22), we have

$$\mathsf{E}_{x|z,y;\theta}\left[x\right] = \mu + z\xi + p^{-1}(y \mid z; \theta)z\Sigma q_z, \tag{26}$$

$$\begin{aligned}\mathsf{E}_{x|z,y;\theta}\left[xx^\top\right] = {}& \mu\mu^\top + z^2\xi\xi^\top + 2z\mu\xi^\top + z\Sigma + p^{-1}(y \mid z; \theta)\left(2z\left(\Sigma q_z\right)\mu^\top\right.\\ & \left. + 2z^2\left(\Sigma q_z\right)\xi^\top + z^2\Sigma\left(H_z + D_z\right)\Sigma\right),\end{aligned} \tag{27}$$

where the $i$-th elements of vector $q_z$ is denoted as

$$q_{z,i} = p(x_i = l_i, y_{\backslash i} \mid z; \theta) - p(x_i = u_i, y_{\backslash i} \mid z; \theta), \tag{28}$$

the matrix $H_z$ is a matrix with all diagonal entries being zero and the $(i, j)$-th off-diagonal element being

$$H_{z,ij} = h_z(l_i, l_j) - h_z(l_i, u_j) - h_z(u_i, l_j) + h_z(u_i, u_j), \tag{29}$$

with

$$h_z(c_i, c_j) = p(x_i = c_i, x_j = c_j, y_{\backslash i,j} \mid z; \theta)),$$

and the matrix $D_z$ is a diagonal matrix with the diagonal entries:

$$D_{z,ii} = \frac{l_i - \mu_i - z\xi_i}{\sigma_i^2}p(x_i = l_i, y_{\backslash i} \mid z; \theta) - \frac{u_i - \mu_i - z\xi_i}{\sigma_i^2}p(x_i = u_i, y_{\backslash i} \mid z; \theta) - \frac{[\Sigma H_z]_{ii}}{\sigma_i^2}. \tag{30}$$

Based on (26) and (27), we can obtain

$$\mathsf{E}_{\boldsymbol{x},z|\boldsymbol{y};\boldsymbol{\theta}}\left[\boldsymbol{x}\right] = \boldsymbol{\mu} + \mathsf{E}_{z|\boldsymbol{y};\boldsymbol{\theta}}\left[z\right]\boldsymbol{\xi} + \mathsf{E}_{z|\boldsymbol{y};\boldsymbol{\theta}}\left[zp^{-1}(\boldsymbol{y}\mid z;\boldsymbol{\theta})\boldsymbol{\Sigma}\boldsymbol{q}_z\right], \tag{31}$$

$$\mathsf{E}_{\boldsymbol{x},z|\boldsymbol{y};\boldsymbol{\theta}}\left[\boldsymbol{x}\boldsymbol{x}^{\top}\right] = \boldsymbol{\mu}\boldsymbol{\mu}^{\top} + \mathsf{E}_{z|\boldsymbol{y};\boldsymbol{\theta}}\left[z^2\right]\boldsymbol{\xi}\boldsymbol{\xi}^{\top} + 2\,\mathsf{E}_{z|\boldsymbol{y};\boldsymbol{\theta}}\left[z\right]\boldsymbol{\mu}\boldsymbol{\xi}^{\top} + \mathsf{E}_{z|\boldsymbol{y};\boldsymbol{\theta}}\left[z\right]\boldsymbol{\Sigma} + \mathsf{E}_{z|\boldsymbol{y};\boldsymbol{\theta}}\left[p^{-1}(\boldsymbol{y}\mid z;\boldsymbol{\theta})\right.$$
$$\left.\left(2z\left(\boldsymbol{\Sigma}\boldsymbol{q}_z\right)\boldsymbol{\mu}^{\top} + 2z^2\left(\boldsymbol{\Sigma}\boldsymbol{q}_z\right)\boldsymbol{\xi}^{\top} + z^2\boldsymbol{\Sigma}\left(\boldsymbol{H}_z + \boldsymbol{D}_z\right)\boldsymbol{\Sigma}\right)\right] \tag{32}$$

We first compute $\mathsf{E}_{z|\boldsymbol{y};\boldsymbol{\theta}}\left[z^k\right]$ as follows:

$$\mathsf{E}_{z|\boldsymbol{y};\boldsymbol{\theta}}\left[z^k\right] = \int_0^{+\infty} p(z\mid\boldsymbol{y};\boldsymbol{\theta})z^k\mathrm{d}z = \int_0^{+\infty}\frac{p(\boldsymbol{y}\mid z;\boldsymbol{\theta})p(z)}{p(\boldsymbol{y};\boldsymbol{\theta})}z^k\mathrm{d}z.$$

Here, we introduce the size-biased distribution of order $k$ of the positive random variable $z$, which has the density function $p_k(z) = \frac{z^k p(z)}{\mathsf{E}_z[z^k]}$. Hence, we have

$$\mathsf{E}_{z|\boldsymbol{y};\boldsymbol{\theta}}\left[z^k\right] = \mathsf{E}_z[z^k]\frac{\int_0^{+\infty} p(\boldsymbol{y}\mid z;\boldsymbol{\theta})p_k(z)\mathrm{d}z}{p(\boldsymbol{y};\boldsymbol{\theta})} = \mathsf{E}_z[z^k]\frac{p_k(\boldsymbol{y};\boldsymbol{\theta})}{p(\boldsymbol{y};\boldsymbol{\theta})}, \tag{33}$$

where we denote $p_k(\boldsymbol{y};\boldsymbol{\theta}) = \int_0^{+\infty} p(\boldsymbol{y}\mid z;\boldsymbol{\theta})p_k(z)\mathrm{d}z$.

Similarly, we can obtain

$$\mathsf{E}_{z|\boldsymbol{y};\boldsymbol{\theta}}\left[z^k p^{-1}(\boldsymbol{y}\mid z;\boldsymbol{\theta})\boldsymbol{q}_z\right] = \mathsf{E}_z[z^k]p^{-1}(\boldsymbol{y};\boldsymbol{\theta})\boldsymbol{q}_k, \tag{34}$$

where $\boldsymbol{q}_k = \int_0^{+\infty}\boldsymbol{q}_z p_k(z)\mathrm{d}z$ with the $i$-th element

$$q_{k,i} = p_k(x_i = l_i, \boldsymbol{y}_{\setminus i};\boldsymbol{\theta}) - p_k(x_i = u_i, \boldsymbol{y}_{\setminus i};\boldsymbol{\theta}). \tag{35}$$

For the expectation of $z^k\boldsymbol{H}_z$ we have

$$\mathsf{E}_{z|\boldsymbol{y};\boldsymbol{\theta}}\left[z^k p^{-1}(\boldsymbol{y}\mid z;\boldsymbol{\theta})\left(\boldsymbol{H}_z + \boldsymbol{D}_z\right)\right] = p^{-1}(\boldsymbol{y};\boldsymbol{\theta})\left(\boldsymbol{H}_k + \boldsymbol{D}_k\right), \tag{36}$$

with the $(i,j)$ entry of

$$\boldsymbol{H}_{k,ij} = \mathsf{E}_z[z^k]\left(h_k(l_i,l_j) - h_k(l_i,u_j) - h_k(u_i,l_j) + h_k(u_i,u_j)\right),$$

$$h_k(c_i,c_j) = p_k(x_i = c_i, x_j = c_j, \boldsymbol{y}_{\setminus i,j};\boldsymbol{\theta})),$$

and the matrix $\boldsymbol{D}_k$ is a diagonal matrix with the diagonal entries:

$$\boldsymbol{D}_{k,ii} = \mathsf{E}_z[z^{k-1}]\frac{l_i - \mu_i}{\sigma_i^2}p_{k-1}(x_i = l_i, \boldsymbol{y}_{\setminus i}\mid z;\boldsymbol{\theta}) - \mathsf{E}_z[z^{k-1}]\frac{u_i - \mu_i}{\sigma_i^2}p_{k-1}(x_i = u_i, \boldsymbol{y}_{\setminus i}\mid z;\boldsymbol{\theta})$$
$$- \mathsf{E}_z[z^k]\frac{\xi_i}{\sigma_i^2}q_{k,i} - \frac{[\boldsymbol{\Sigma}\boldsymbol{H}_k]_{ii}}{\sigma_i^2}. \tag{37}$$

Therefore, the expectations in (31) and (32) become

$$\mathsf{E}_{\boldsymbol{x},z|\boldsymbol{y};\boldsymbol{\theta}}\left[\boldsymbol{x}\right] = \boldsymbol{\mu} + \mathsf{E}_{z|\boldsymbol{y};\boldsymbol{\theta}}\left[z\right]\boldsymbol{\xi} + \mathsf{E}_z\left[z\right]p^{-1}(\boldsymbol{y};\boldsymbol{\theta})\boldsymbol{\Sigma}\boldsymbol{q}_1, \tag{38}$$

$$\mathsf{E}_{\boldsymbol{x},z|\boldsymbol{y};\boldsymbol{\theta}}\left[\boldsymbol{x}\boldsymbol{x}^{\top}\right] = \boldsymbol{\mu}\boldsymbol{\mu}^{\top} + \mathsf{E}_{z|\boldsymbol{y};\boldsymbol{\theta}}\left[z^2\right]\boldsymbol{\xi}\boldsymbol{\xi}^{\top} + 2\mathsf{E}_{z|\boldsymbol{y};\boldsymbol{\theta}}\left[z\right]\boldsymbol{\mu}\boldsymbol{\xi}^{\top} + \mathsf{E}_{z|\boldsymbol{y};\boldsymbol{\theta}}\left[z\right]\boldsymbol{\Sigma} + p^{-1}(\boldsymbol{y};\boldsymbol{\theta})$$
$$\times\left(2\mathsf{E}_z\left[z\right]\left(\boldsymbol{\Sigma}\boldsymbol{q}_1\right)\boldsymbol{\mu}^{\top} + 2\mathsf{E}_z\left[z^2\right]\left(\boldsymbol{\Sigma}\boldsymbol{q}_2\right)\boldsymbol{\xi}^{\top} + \boldsymbol{\Sigma}\left(\boldsymbol{H}_2 + \boldsymbol{D}_2\right)\boldsymbol{\Sigma}\right). \tag{39}$$

Meanwhile, we can also obtain

$$\mathsf{E}_{\boldsymbol{x},z|\boldsymbol{y};\boldsymbol{\theta}}\left[z^{-1}\boldsymbol{x}\right] = \mathsf{E}_{z|\boldsymbol{y};\boldsymbol{\theta}}\left[\mathsf{E}_{\boldsymbol{x}|z,\boldsymbol{y};\boldsymbol{\theta}}[z^{-1}\boldsymbol{x}]\right] = \mathsf{E}_{z|\boldsymbol{y};\boldsymbol{\theta}}\left[z^{-1}\right]\boldsymbol{\mu} + \boldsymbol{\xi} + p^{-1}(\boldsymbol{y};\boldsymbol{\theta})\boldsymbol{\Sigma}\boldsymbol{q}, \tag{40}$$

$$\tag{41}$$

$$\mathsf{E}_{\boldsymbol{x},z|\boldsymbol{y};\boldsymbol{\theta}}\left[z^{-1}\boldsymbol{x}\boldsymbol{x}^{\top}\right] = \mathsf{E}_{z|\boldsymbol{y};\boldsymbol{\theta}}\left[\mathsf{E}_{\boldsymbol{x}|z,\boldsymbol{y};\boldsymbol{\theta}}[z^{-1}\boldsymbol{x}\boldsymbol{x}^{\top}]\right] = \mathsf{E}_{z|\boldsymbol{y};\boldsymbol{\theta}}\left[z^{-1}\right]\boldsymbol{\mu}\boldsymbol{\mu}^{\top} + \mathsf{E}_{z|\boldsymbol{y};\boldsymbol{\theta}}\left[z\right]\boldsymbol{\xi}\boldsymbol{\xi}^{\top} + 2\boldsymbol{\mu}\boldsymbol{\xi}^{\top} + \boldsymbol{\Sigma} + p^{-1}(\boldsymbol{y};\boldsymbol{\theta})\times$$
$$\left(2\left(\boldsymbol{\Sigma}\boldsymbol{q}\right)\boldsymbol{\mu}^{\top} + 2\mathsf{E}_z\left[z\right]\left(\boldsymbol{\Sigma}\boldsymbol{q}_1\right)\boldsymbol{\xi}^{\top} + \boldsymbol{\Sigma}\left(\boldsymbol{H}_1 + \boldsymbol{D}_1\right)\boldsymbol{\Sigma}\right). \tag{42}$$

# B. The Proof of Theorem 2

Given the log-likelihood function as

$$L(\boldsymbol{\theta}) = \sum_{t=1}^{n} \log p(\boldsymbol{y}_t; \boldsymbol{\theta}).$$

in the following, we prove that $L(\boldsymbol{\theta})$ is component-wise concave with respect to $\boldsymbol{\mu}$, $\boldsymbol{\xi}$, and $\boldsymbol{\Sigma}$.

## B.1. Concavity of Location Parameter

The second order derivative of $\log p(\boldsymbol{y}; \boldsymbol{\theta})$ with respect to $\boldsymbol{\mu}$ is given as

$$\nabla_{\boldsymbol{\mu}}^2 \log p(\boldsymbol{y}; \boldsymbol{\theta}) = \frac{\nabla_{\boldsymbol{\mu}}^2 p(\boldsymbol{y}; \boldsymbol{\theta}) \cdot p(\boldsymbol{y}; \boldsymbol{\theta}) - \nabla_{\boldsymbol{\mu}} p(\boldsymbol{y}; \boldsymbol{\theta}) \nabla_{\boldsymbol{\mu}}^{\top} p(\boldsymbol{y}; \boldsymbol{\theta})}{p^2(\boldsymbol{y}; \boldsymbol{\theta})}. \tag{43}$$

To analyze the above expression, we should compute the first and second order derivatives of $p(\boldsymbol{y}; \boldsymbol{\theta})$ with respect to $\boldsymbol{\mu}$ at first. The first order one is given as follows:

$$\nabla_{\boldsymbol{\mu}} p(\boldsymbol{y}; \boldsymbol{\theta}) = \nabla_{\boldsymbol{\mu}} \int_{\mathcal{Q}^{-1}(\boldsymbol{y})} p(\boldsymbol{x}; \boldsymbol{\theta}) \mathrm{d}\boldsymbol{x} = \int_{\mathcal{Q}^{-1}(\boldsymbol{y})} \nabla_{\boldsymbol{\mu}} p(\boldsymbol{x}; \boldsymbol{\theta}) \mathrm{d}\boldsymbol{x}. \tag{44}$$

Since $p(\boldsymbol{x}; \boldsymbol{\theta}) = \int_0^{+\infty} p(z)p(\boldsymbol{x} \mid z; \boldsymbol{\theta})\mathrm{d}z$, (44) becomes

$$\nabla_{\boldsymbol{\mu}} p(\boldsymbol{y}; \boldsymbol{\theta}) = \int_{\mathcal{Q}^{-1}(\boldsymbol{y})} \int_0^{+\infty} p(z) \nabla_{\boldsymbol{\mu}} p(\boldsymbol{x} \mid z; \boldsymbol{\theta}) \mathrm{d}z \mathrm{d}\boldsymbol{x} = \int_{\mathcal{Q}^{-1}(\boldsymbol{y})} \int_0^{+\infty} \frac{p(z)}{z} p(\boldsymbol{x} \mid z; \boldsymbol{\theta}) \boldsymbol{\Sigma}^{-1}(\boldsymbol{x} - \boldsymbol{\mu} - z\boldsymbol{\xi}) \mathrm{d}z \mathrm{d}\boldsymbol{x}.$$

Then the second order one can be further computed as

$$\begin{aligned} \nabla_{\boldsymbol{\mu}}^2 p(\boldsymbol{y}; \boldsymbol{\theta}) &= \int_{\mathcal{Q}^{-1}(\boldsymbol{y})} \int_0^{+\infty} \frac{p(z)}{z} \nabla_{\boldsymbol{\mu}} \left( p(\boldsymbol{x} \mid z; \boldsymbol{\theta}) \boldsymbol{\Sigma}^{-1}(\boldsymbol{x} - \boldsymbol{\mu} - z\boldsymbol{\xi}) \right) \mathrm{d}z \mathrm{d}\boldsymbol{x} \\ &= \int_{\mathcal{Q}^{-1}(\boldsymbol{y})} \int_0^{+\infty} \boldsymbol{\Sigma}^{-1} p(\boldsymbol{x} \mid z; \boldsymbol{\theta}) p(z) \left( -z^{-1}\boldsymbol{I} + z^{-2}(\boldsymbol{x} - \boldsymbol{\mu} - z\boldsymbol{\xi})(\boldsymbol{x} - \boldsymbol{\mu} - z\boldsymbol{\xi})^{\top} \boldsymbol{\Sigma}^{-1} \right) \mathrm{d}z \mathrm{d}\boldsymbol{x}. \end{aligned} \tag{45}$$

**Lemma 5.** *For a function $f(\boldsymbol{x}, z)$, if $\int_{\mathcal{Q}^{-1}(\boldsymbol{y})} p(\boldsymbol{x} \mid z; \boldsymbol{\theta}) f(\boldsymbol{x}, z)\mathrm{d}\boldsymbol{x}$ is integrable, we have*

$$\int_{\mathbb{R}^d} \int_0^{+\infty} p(z)p(\boldsymbol{x}, \boldsymbol{y} \mid z; \boldsymbol{\theta}) f(\boldsymbol{x}, z)\mathrm{d}z\mathrm{d}\boldsymbol{x} = p(\boldsymbol{y}; \boldsymbol{\theta}) \mathsf{E}_{\boldsymbol{x}, z \mid \boldsymbol{y}; \boldsymbol{\theta}} \left[ f(\boldsymbol{x}, z) \right].$$

*Proof.* Given that

$$\int_{\mathcal{Q}^{-1}(\boldsymbol{y})} p(\boldsymbol{x} \mid z; \boldsymbol{\theta}) f(\boldsymbol{x}, z)\mathrm{d}\boldsymbol{x} = \int_{\mathbb{R}^d} p(\boldsymbol{x}, \boldsymbol{y} \mid z; \boldsymbol{\theta}) f(\boldsymbol{x}, z)\mathrm{d}\boldsymbol{x},$$

we can obtain that

$$\int_{\mathbb{R}^d} \int_0^{+\infty} p(z)p(\boldsymbol{x}, \boldsymbol{y} \mid z; \boldsymbol{\theta}) f(\boldsymbol{x}, z)\mathrm{d}z\mathrm{d}\boldsymbol{x} = p(\boldsymbol{y}; \boldsymbol{\theta}) \int_{\mathbb{R}^d} \int_0^{+\infty} p(\boldsymbol{x}, z \mid \boldsymbol{y}; \boldsymbol{\theta}) f(\boldsymbol{x}, z)\mathrm{d}z\mathrm{d}\boldsymbol{x} = p(\boldsymbol{y}; \boldsymbol{\theta}) \mathsf{E}_{\boldsymbol{x}, z \mid \boldsymbol{y}; \boldsymbol{\theta}} \left[ f(\boldsymbol{x}, z) \right].$$

$$\square$$

Based on Lemma 5, the first and second order derivatives of $p(\boldsymbol{y}; \boldsymbol{\theta})$ becomes

$$\nabla_{\boldsymbol{\mu}} p(\boldsymbol{y}; \boldsymbol{\theta}) = p(\boldsymbol{y}; \boldsymbol{\theta}) \boldsymbol{\Sigma}^{-1} \mathsf{E}_{\boldsymbol{x}, z \mid \boldsymbol{y}; \boldsymbol{\theta}} \left[ z^{-1}(\boldsymbol{x} - \boldsymbol{\mu} - z\boldsymbol{\xi}) \right], \tag{46}$$

and

$$\nabla_{\boldsymbol{\mu}}^2 p(\boldsymbol{y}; \boldsymbol{\theta}) = p(\boldsymbol{y}; \boldsymbol{\theta}) \boldsymbol{\Sigma}^{-1} \left( -\mathsf{E}_{z \mid \boldsymbol{y}; \boldsymbol{\theta}} \left[ z^{-1} \right] \boldsymbol{I} + \mathsf{E}_{\boldsymbol{x}, z \mid \boldsymbol{y}; \boldsymbol{\theta}} \left[ z^{-2}(\boldsymbol{x} - \boldsymbol{\mu} - z\boldsymbol{\xi})(\boldsymbol{x} - \boldsymbol{\mu} - z\boldsymbol{\xi})^{\top} \right] \boldsymbol{\Sigma}^{-1} \right). \tag{47}$$

Substituting (46) and (47) into the second order derivative of $\log p(\boldsymbol{y}; \boldsymbol{\theta})$ in (43) becomes

$$\nabla_{\boldsymbol{\mu}}^2 \log p(\boldsymbol{y}; \boldsymbol{\theta}) = \boldsymbol{\Sigma}^{-1} \left( -\mathsf{E}_{z|\boldsymbol{y};\boldsymbol{\theta}} \left[ z^{-1} \right] \boldsymbol{I} + \mathsf{E}_{\boldsymbol{x},z|\boldsymbol{y};\boldsymbol{\theta}} \left[ z^{-2}(\boldsymbol{x} - \boldsymbol{\mu} - z\boldsymbol{\xi})(\boldsymbol{x} - \boldsymbol{\mu} - z\boldsymbol{\xi})^\top \right] \boldsymbol{\Sigma}^{-1} \right.$$
$$\left. - \boldsymbol{\Sigma}^{-1} \mathsf{E}_{\boldsymbol{x},z|\boldsymbol{y};\boldsymbol{\theta}} \left[ z^{-1}(\boldsymbol{x} - \boldsymbol{\mu} - z\boldsymbol{\xi}) \right] \mathsf{E}_{\boldsymbol{x},z|\boldsymbol{y};\boldsymbol{\theta}}^\top \left[ z^{-1}(\boldsymbol{x} - \boldsymbol{\mu} - z\boldsymbol{\xi}) \right] \boldsymbol{\Sigma}^{-1} \right) \tag{48}$$
$$= \boldsymbol{\Sigma}^{-1} \left( -\mathsf{E}_{z|\boldsymbol{y};\boldsymbol{\theta}} \left[ z^{-1} \right] \boldsymbol{\Sigma} + \mathsf{Cov}_{\boldsymbol{x},z|\boldsymbol{y};\boldsymbol{\theta}} \left[ z^{-1}(\boldsymbol{x} - \boldsymbol{\mu} - z\boldsymbol{\xi}) \right] \right) \boldsymbol{\Sigma}^{-1},$$

Since $\boldsymbol{\Sigma}^{-1}$ on the both sides of the parentheses is positive definite, the definiteness of $\nabla_{\boldsymbol{\mu}}^2 \log p(\boldsymbol{y}; \boldsymbol{\theta})$ is determined by the term in the parentheses. For the covariance term, based on law of total variance, we have

$$\mathsf{Cov}_{\boldsymbol{x},z|\boldsymbol{y};\boldsymbol{\theta}} \left[ z^{-1}(\boldsymbol{x} - \boldsymbol{\mu}) \right] = \mathsf{E}_{\boldsymbol{x},z|\boldsymbol{y};\boldsymbol{\theta}} \left[ \mathsf{Cov}_{\boldsymbol{x}|z,\boldsymbol{y};\boldsymbol{\theta}} \left[ z^{-1}(\boldsymbol{x} - \boldsymbol{\mu} - z\boldsymbol{\xi}) \right] \right] + \mathsf{Cov}_{\boldsymbol{x},z|\boldsymbol{y};\boldsymbol{\theta}} \left[ \mathsf{E}_{\boldsymbol{x}|z,\boldsymbol{y};\boldsymbol{\theta}} \left[ z^{-1}(\boldsymbol{x} - \boldsymbol{\mu} - z\boldsymbol{\xi}) \right] \right].$$
$$\tag{49}$$

Due to $\mathsf{E}_{\boldsymbol{x}|z,\boldsymbol{y};\boldsymbol{\theta}} \left[ \boldsymbol{x} - \boldsymbol{\mu} - z\boldsymbol{\xi} \right] = \boldsymbol{0}$, we only consider the first term in (49). To describe the relations between the expectation term and the covariance term, we introduce the Brascamp–Lieb inequality.

**Lemma 6** ((Brascamp & Lieb, 1976)). *Consider a probability density function $p(\boldsymbol{X} \mid \boldsymbol{Y})$ which is log-concave to $\boldsymbol{X}$. The Brascamp–Lieb inequality is given by*

$$\mathsf{Cov}_{\boldsymbol{X}|\boldsymbol{Y}}(f(\boldsymbol{X})) \preceq \mathsf{E}_{\boldsymbol{X}|\boldsymbol{Y}} \left[ \nabla_{\boldsymbol{X}}^\top f(\boldsymbol{X})[\nabla_{\boldsymbol{X}}^2(-\log p(\boldsymbol{X} \mid \boldsymbol{Y}))]^{-1} \nabla_{\boldsymbol{X}} f(\boldsymbol{X}) \right],$$

*where the equality is obtained when $\log p(\boldsymbol{X} \mid \boldsymbol{Y})$ is linear with respect to $\boldsymbol{X}$.*

Let $\boldsymbol{X} = \boldsymbol{x}$, $\boldsymbol{X} \mid \boldsymbol{Y} = \boldsymbol{x} \mid z, \boldsymbol{y}$, and $f(\boldsymbol{X}) = z^{-1}\boldsymbol{x}$. Given that

$$\nabla_{\boldsymbol{x}}^2 \log p(\boldsymbol{x} \mid z, \boldsymbol{y}; \boldsymbol{\theta}) = \nabla_{\boldsymbol{x}}^2 \log \left( \frac{p(\boldsymbol{x}, z; \boldsymbol{\theta})p(\boldsymbol{y} \mid \boldsymbol{x}; \boldsymbol{\theta})}{p(\boldsymbol{y}, z; \boldsymbol{\theta})} \right) = -\frac{\boldsymbol{\Sigma}^{-1}}{z} \prec \boldsymbol{0},$$

and $\nabla_{\boldsymbol{x}} z^{-1}(\boldsymbol{x} - \boldsymbol{\mu}) = z^{-1}\boldsymbol{1}$, based on the Brascamp–Lieb inequality in Lemma 6, we have

$$\mathsf{Cov}_{\boldsymbol{x}|z,\boldsymbol{y};\boldsymbol{\theta}} \left[ z^{-1}(\boldsymbol{x} - \boldsymbol{\mu}) \right] \preceq \mathsf{E}_{\boldsymbol{x}|z,\boldsymbol{y};\boldsymbol{\theta}} \left[ z^{-1} z \boldsymbol{\Sigma} z^{-1} \right] = z^{-1} \boldsymbol{\Sigma},$$

Since $p(\boldsymbol{x} \mid z, \boldsymbol{y})$ is not linear to $\boldsymbol{x}$, the above inequality is strict. Hence we have

$$\mathsf{Cov}_{\boldsymbol{x},z|\boldsymbol{y};\boldsymbol{\theta}} \left[ z^{-1}(\boldsymbol{x} - \boldsymbol{\mu}) \right] \prec \mathsf{E}_{\boldsymbol{x},z|\boldsymbol{y};\boldsymbol{\theta}} \left[ z^{-1} \right] \boldsymbol{\Sigma},$$

i.e., $\nabla_{\boldsymbol{\mu}}^2 \log p(\boldsymbol{y}; \boldsymbol{\theta})$ is negative definite.

## B.2. Concavity of Skewness Parameter

The second order derivative of $\log p(\boldsymbol{y}; \boldsymbol{\theta})$ with respect to $\boldsymbol{\xi}$ is given as

$$\nabla_{\boldsymbol{\xi}}^2 \log p(\boldsymbol{y}; \boldsymbol{\theta}) = \frac{\nabla_{\boldsymbol{\xi}}^2 p(\boldsymbol{y}; \boldsymbol{\theta}) \cdot p(\boldsymbol{y}; \boldsymbol{\theta}) - \nabla_{\boldsymbol{\xi}} p(\boldsymbol{y}; \boldsymbol{\theta}) \nabla_{\boldsymbol{\xi}}^\top p(\boldsymbol{y}; \boldsymbol{\theta})}{p^2(\boldsymbol{y}; \boldsymbol{\theta})}. \tag{50}$$

To analyze the above expression, we should compute the first and second order derivatives of $p(\boldsymbol{y}; \boldsymbol{\theta})$ with respect to $\boldsymbol{\xi}$ at first. The first order one is given as follows:

$$\nabla_{\boldsymbol{\xi}} p(\boldsymbol{y}; \boldsymbol{\theta}) = \nabla_{\boldsymbol{\xi}} \int_{\mathcal{Q}^{-1}(\boldsymbol{y})} p(\boldsymbol{x}; \boldsymbol{\theta}) \mathrm{d}\boldsymbol{x} = \int_{\mathcal{Q}^{-1}(\boldsymbol{y})} \nabla_{\boldsymbol{\xi}} p(\boldsymbol{x}; \boldsymbol{\theta}) \mathrm{d}\boldsymbol{x}. \tag{51}$$

Since $p(\boldsymbol{x}; \boldsymbol{\theta}) = \int_0^{+\infty} p(z) p(\boldsymbol{x} \mid z; \boldsymbol{\theta}) \mathrm{d}z$, (51) becomes

$$\nabla_{\boldsymbol{\xi}} p(\boldsymbol{y}; \boldsymbol{\theta}) = \int_{\mathcal{Q}^{-1}(\boldsymbol{y})} \int_0^{+\infty} p(z) \nabla_{\boldsymbol{\xi}} p(\boldsymbol{x} \mid z; \boldsymbol{\theta}) \mathrm{d}z \mathrm{d}\boldsymbol{x} = \int_{\mathcal{Q}^{-1}(\boldsymbol{y})} \int_0^{+\infty} \left( p(z) p(\boldsymbol{x} \mid z; \boldsymbol{\theta}) \boldsymbol{\Sigma}^{-1}(\boldsymbol{x} - \boldsymbol{\mu} - z\boldsymbol{\xi}) \right) \mathrm{d}z \mathrm{d}\boldsymbol{x}.$$

Then the second order derivative can be further computed as

$$\nabla_{\boldsymbol{\xi}}^2 p(\boldsymbol{y}; \boldsymbol{\theta}) = \int_{\mathcal{Q}^{-1}(\boldsymbol{y})} \int_0^{+\infty} p(z) \nabla_{\boldsymbol{\xi}} \left( p(\boldsymbol{x} \mid z; \boldsymbol{\theta}) \boldsymbol{\Sigma}^{-1}(\boldsymbol{x} - \boldsymbol{\mu} - z\boldsymbol{\xi}) \right) \mathrm{d}z \mathrm{d}\boldsymbol{x}$$
$$= \int_{\mathcal{Q}^{-1}(\boldsymbol{y})} \int_0^{+\infty} \boldsymbol{\Sigma}^{-1} p(\boldsymbol{x} \mid z; \boldsymbol{\theta}) p(z) \left( -z\boldsymbol{I} + (\boldsymbol{x} - \boldsymbol{\mu} - z\boldsymbol{\xi})(\boldsymbol{x} - \boldsymbol{\mu} - z\boldsymbol{\xi})^\top \boldsymbol{\Sigma}^{-1} \right) \mathrm{d}z \mathrm{d}\boldsymbol{x}. \tag{52}$$

Based on Lemma 5, the derivatives of $p(\boldsymbol{y}; \boldsymbol{\theta})$ becomes

$$\nabla_{\boldsymbol{\xi}} p(\boldsymbol{y}; \boldsymbol{\theta}) = p(\boldsymbol{y}; \boldsymbol{\theta}) \boldsymbol{\Sigma}^{-1} \mathsf{E}_{\boldsymbol{x}, z | \boldsymbol{y}; \boldsymbol{\theta}} \left[ \boldsymbol{x} - \boldsymbol{\mu} - z \boldsymbol{\xi} \right], \tag{53}$$

and

$$\nabla_{\boldsymbol{\xi}}^2 p(\boldsymbol{y}; \boldsymbol{\theta}) = p(\boldsymbol{y}; \boldsymbol{\theta}) \boldsymbol{\Sigma}^{-1} \left( -\mathsf{E}_{z | \boldsymbol{y}; \boldsymbol{\theta}}[z] \boldsymbol{I} + \mathsf{E}_{\boldsymbol{x}, z | \boldsymbol{y}; \boldsymbol{\theta}} \left[ (\boldsymbol{x} - \boldsymbol{\mu} - z \boldsymbol{\xi})(\boldsymbol{x} - \boldsymbol{\mu} - z \boldsymbol{\xi})^{\top} \right] \boldsymbol{\Sigma}^{-1} \right). \tag{54}$$

Substituting (53) and (54) into $\log p(\boldsymbol{y}; \boldsymbol{\theta})$ in (50), we have that

$$\nabla_{\boldsymbol{\xi}}^2 \log p(\boldsymbol{y}; \boldsymbol{\theta}) = \boldsymbol{\Sigma}^{-1} \left( -\mathsf{E}_{z | \boldsymbol{y}; \boldsymbol{\theta}}[z] \boldsymbol{\Sigma} + \mathsf{Cov}_{\boldsymbol{x}, z | \boldsymbol{y}; \boldsymbol{\theta}}[\boldsymbol{x} - \boldsymbol{\mu} - z \boldsymbol{\xi}] \right) \boldsymbol{\Sigma}^{-1}. \tag{55}$$

Based on the law of total variance, the covariance term becomes

$$\mathsf{Cov}_{\boldsymbol{x}, z | \boldsymbol{y}; \boldsymbol{\theta}} \left[ \boldsymbol{x} - \boldsymbol{\mu} - z \boldsymbol{\xi} \right] = \mathsf{E}_{\boldsymbol{x}, z | \boldsymbol{y}; \boldsymbol{\theta}} \left[ \mathsf{Cov}_{\boldsymbol{x} | z, \boldsymbol{y}; \boldsymbol{\theta}} \left[ \boldsymbol{x} - \boldsymbol{\mu} - z \boldsymbol{\xi} \right] \right] + \mathsf{Cov}_{\boldsymbol{x}, z | \boldsymbol{y}; \boldsymbol{\theta}} \left[ \mathsf{E}_{\boldsymbol{x} | z, \boldsymbol{y}; \boldsymbol{\theta}} \left[ \boldsymbol{x} - \boldsymbol{\mu} - z \boldsymbol{\xi} \right] \right]. \tag{56}$$

where the second term satisfies $\mathsf{Cov}_{\boldsymbol{x}, z | \boldsymbol{y}; \boldsymbol{\theta}} \left[ \mathsf{E}_{\boldsymbol{x} | z, \boldsymbol{y}; \boldsymbol{\theta}} \left[ \boldsymbol{x} - \boldsymbol{\mu} - z \boldsymbol{\xi} \right] \right] = \boldsymbol{0}$.

Setting $\boldsymbol{X} = \boldsymbol{x}$, $\boldsymbol{X} = \boldsymbol{x} \mid z, \boldsymbol{y}$, and $f(\boldsymbol{X}) = \boldsymbol{x} - \boldsymbol{\mu} - z \boldsymbol{\xi}$, since $p(\boldsymbol{x}, z \mid \boldsymbol{y}; \boldsymbol{\theta})$ is a log-concave function with respect to $\boldsymbol{x}$. Based on the Brascamp–Lieb inequality in Lemma 6, we have

$$\mathsf{Cov}_{\boldsymbol{x} | z, \boldsymbol{y}; \boldsymbol{\theta}} \left[ \boldsymbol{x} - \boldsymbol{\mu} - z \boldsymbol{\xi} \right] \prec \mathsf{E}_{\boldsymbol{x}, z | \boldsymbol{y}; \boldsymbol{\theta}} \left[ z \right] \boldsymbol{\Sigma},$$

and hence $\nabla_{\boldsymbol{\xi}}^2 p(\boldsymbol{y}; \boldsymbol{\theta})$ is negative definite.

## B.3. Concavity of Scatter Matrix

Since $\log p(\boldsymbol{x} \mid z; \boldsymbol{\theta})$ is linear with respect to $\boldsymbol{\Sigma}^{-1}$, the second order derivative of $\log p(\boldsymbol{y}; \boldsymbol{\theta})$ with respect to $\boldsymbol{\Sigma}^{-1}$ is easier to be obtained. We first compute $\frac{\partial \mathrm{vec} \left( \nabla_{\boldsymbol{\Sigma}^{-1}} \log p(\boldsymbol{y}_t; \boldsymbol{\theta}) \right)}{\partial \mathrm{vec} \left( \boldsymbol{\Sigma}^{-1} \right)}$ and transform it later.

The derivative term $\frac{\partial \mathrm{vec} \left( \nabla_{\boldsymbol{\Sigma}^{-1}} \log p(\boldsymbol{y}_t; \boldsymbol{\theta}) \right)}{\partial \mathrm{vec} \left( \boldsymbol{\Sigma}^{-1} \right)}$ can be computed as

$$\frac{\partial \mathrm{vec} \left( \nabla_{\boldsymbol{\Sigma}^{-1}} \log p(\boldsymbol{y}; \boldsymbol{\theta}) \right)}{\partial \mathrm{vec} \left( \boldsymbol{\Sigma}^{-1} \right)} = \frac{\frac{\partial \mathrm{vec} \left( \nabla_{\boldsymbol{\Sigma}^{-1}} p(\boldsymbol{y}_t; \boldsymbol{\theta}) \right)}{\partial \mathrm{vec} \left( \boldsymbol{\Sigma}^{-1} \right)} \cdot p(\boldsymbol{y}; \boldsymbol{\theta}) - \nabla_{\boldsymbol{\Sigma}^{-1}} p(\boldsymbol{y}; \boldsymbol{\theta}) \otimes \nabla_{\boldsymbol{\Sigma}^{-1}} p(\boldsymbol{y}; \boldsymbol{\theta})}{p^2(\boldsymbol{y}; \boldsymbol{\theta})}. \tag{57}$$

Since the first and second order derivatives of $p(\boldsymbol{y}; \boldsymbol{\theta})$ with respect to $\boldsymbol{\Sigma}^{-1}$ satisfy

$$\nabla_{\boldsymbol{\Sigma}^{-1}} p(\boldsymbol{y}; \boldsymbol{\theta}) = \int_{\mathcal{Q}^{-1}(\boldsymbol{y})} \int_0^{+\infty} \nabla_{\boldsymbol{\Sigma}^{-1}} p(\boldsymbol{x}, z; \boldsymbol{\theta}) \mathrm{d}z \mathrm{d}\boldsymbol{x},$$

and

$$\frac{\partial \mathrm{vec} \left( \nabla_{\boldsymbol{\Sigma}^{-1}} p(\boldsymbol{y}; \boldsymbol{\theta}) \right)}{\partial \mathrm{vec} \left( \boldsymbol{\Sigma}^{-1} \right)} = \int_{\mathcal{Q}^{-1}(\boldsymbol{y})} \int_0^{+\infty} \frac{\partial \mathrm{vec} \left( \nabla_{\boldsymbol{\Sigma}^{-1}} p(\boldsymbol{x}, z; \boldsymbol{\theta}) \right)}{\partial \mathrm{vec} \left( \boldsymbol{\Sigma}^{-1} \right)} \mathrm{d}z \mathrm{d}\boldsymbol{x},$$

based on Lemma 5, the expression (57) becomes

$$\frac{\frac{\partial \mathrm{vec} \left( \nabla_{\boldsymbol{\Sigma}^{-1}} p(\boldsymbol{y}; \boldsymbol{\theta}) \right)}{\partial \mathrm{vec} \left( \boldsymbol{\Sigma}^{-1} \right)} \cdot p(\boldsymbol{y}; \boldsymbol{\theta}) - \nabla_{\boldsymbol{\Sigma}^{-1}} p(\boldsymbol{y}; \boldsymbol{\theta}) \otimes \nabla_{\boldsymbol{\Sigma}^{-1}} p(\boldsymbol{y}; \boldsymbol{\theta})}{p^2(\boldsymbol{y}; \boldsymbol{\theta})}$$

$$= \frac{\int_{\mathcal{Q}^{-1}(\boldsymbol{y})} \int_0^{+\infty} \frac{\partial \mathrm{vec} \left( \nabla_{\boldsymbol{\Sigma}^{-1}} p(\boldsymbol{x}, z; \boldsymbol{\theta}) \right)}{\partial \mathrm{vec} \left( \boldsymbol{\Sigma}^{-1} \right)} \mathrm{d}z \mathrm{d}\boldsymbol{x}}{p(\boldsymbol{y}; \boldsymbol{\theta})} - \frac{\int_{\mathcal{Q}^{-1}(\boldsymbol{y})} \int_0^{+\infty} \nabla_{\boldsymbol{\Sigma}^{-1}} p(\boldsymbol{x}, z; \boldsymbol{\theta}) \mathrm{d}z \mathrm{d}\boldsymbol{x}}{p(\boldsymbol{y}; \boldsymbol{\theta})} \otimes \frac{\int_{\mathcal{Q}^{-1}(\boldsymbol{y})} \int_0^{+\infty} \nabla_{\boldsymbol{\Sigma}^{-1}} p(\boldsymbol{x}, z; \boldsymbol{\theta}) \mathrm{d}z \mathrm{d}\boldsymbol{x}}{p(\boldsymbol{y}; \boldsymbol{\theta})}$$

$$= \mathsf{E}_{\boldsymbol{x}, z | \boldsymbol{y}; \boldsymbol{\theta}} \left[ \frac{\frac{\partial \mathrm{vec} \left( \nabla_{\boldsymbol{\Sigma}^{-1}} p(\boldsymbol{x}, z; \boldsymbol{\theta}) \right)}{\partial \mathrm{vec} \left( \boldsymbol{\Sigma}^{-1} \right)}}{p(\boldsymbol{x}, z; \boldsymbol{\theta})} \right] - \mathsf{E}_{\boldsymbol{x}, z | \boldsymbol{y}; \boldsymbol{\theta}} \left[ \frac{\nabla_{\boldsymbol{\Sigma}^{-1}} p(\boldsymbol{x}, z; \boldsymbol{\theta})}{p(\boldsymbol{x}, z; \boldsymbol{\theta})} \right] \otimes \mathsf{E}_{\boldsymbol{x}, z | \boldsymbol{y}; \boldsymbol{\theta}} \left[ \frac{\nabla_{\boldsymbol{\Sigma}^{-1}} p(\boldsymbol{x}, z; \boldsymbol{\theta})}{p(\boldsymbol{x}, z; \boldsymbol{\theta})} \right]. \tag{58}$$

Since we have

$$\nabla_{\boldsymbol{\Sigma}^{-1}} p(\boldsymbol{x}, z; \boldsymbol{\theta}) = \frac{p(\boldsymbol{x}, z; \boldsymbol{\theta})}{2} \left( \boldsymbol{\Sigma} - z^{-1}(\boldsymbol{x} - \boldsymbol{\mu} - z\boldsymbol{\xi})(\boldsymbol{x} - \boldsymbol{\mu} - z\boldsymbol{\xi})^{\top} \right),$$

and

$$\frac{\partial \mathrm{vec}\left( \nabla_{\boldsymbol{\Sigma}^{-1}} p(\boldsymbol{x}, z; \boldsymbol{\theta}) \right)}{\partial \mathrm{vec}\left( \boldsymbol{\Sigma}^{-1} \right)} = \frac{p(\boldsymbol{x}, z; \boldsymbol{\theta})}{4} \left( \boldsymbol{\Sigma} - z^{-1}(\boldsymbol{x} - \boldsymbol{\mu} - z\boldsymbol{\xi})(\boldsymbol{x} - \boldsymbol{\mu} - z\boldsymbol{\xi})^{\top} \right)$$

$$\otimes \left( \boldsymbol{\Sigma} - z^{-1}(\boldsymbol{x} - \boldsymbol{\mu} - z\boldsymbol{\xi})(\boldsymbol{x} - \boldsymbol{\mu} - z\boldsymbol{\xi})^{\top} \right) - \frac{p(\boldsymbol{x}, z; \boldsymbol{\theta})}{2} \boldsymbol{\Sigma} \otimes \boldsymbol{\Sigma},$$

the expression (58) can further be derived as

$$
\begin{aligned}
& \mathsf{E}_{\boldsymbol{x}, z | \boldsymbol{y}; \boldsymbol{\theta}} \left[ \frac{\frac{\partial \mathrm{vec}\left( \nabla_{\boldsymbol{\Sigma}^{-1}} p(\boldsymbol{x}, z; \boldsymbol{\theta}) \right)}{\partial \mathrm{vec}\left( \boldsymbol{\Sigma}^{-1} \right)}}{p(\boldsymbol{x}, z; \boldsymbol{\theta})} \right] - \mathsf{E}_{\boldsymbol{x}, z | \boldsymbol{y}; \boldsymbol{\theta}} \left[ \frac{\nabla_{\boldsymbol{\Sigma}^{-1}} p(\boldsymbol{x}, z; \boldsymbol{\theta})}{p(\boldsymbol{x}, z; \boldsymbol{\theta})} \right] \otimes \mathsf{E}_{\boldsymbol{x}, z | \boldsymbol{y}; \boldsymbol{\theta}} \left[ \frac{\nabla_{\boldsymbol{\Sigma}^{-1}} p(\boldsymbol{x}, z; \boldsymbol{\theta})}{p(\boldsymbol{x}, z; \boldsymbol{\theta})} \right] \\
&= \frac{1}{4} \mathsf{E}_{\boldsymbol{x}, z | \boldsymbol{y}; \boldsymbol{\theta}} \left[ \mathrm{vec}\left( \boldsymbol{\Sigma} - z^{-1}(\boldsymbol{x} - \boldsymbol{\mu} - z\boldsymbol{\xi})(\boldsymbol{x} - \boldsymbol{\mu} - z\boldsymbol{\xi})^{\top} \right) \mathrm{vec}\left( \boldsymbol{\Sigma} - z^{-1}(\boldsymbol{x} - \boldsymbol{\mu} - z\boldsymbol{\xi})(\boldsymbol{x} - \boldsymbol{\mu} - z\boldsymbol{\xi})^{\top} \right)^{\top} \right] \\
& \quad - \frac{1}{2} \boldsymbol{\Sigma} \otimes \boldsymbol{\Sigma} - \frac{1}{4} \mathsf{E}_{\boldsymbol{x}, z | \boldsymbol{y}; \boldsymbol{\theta}} \left[ \mathrm{vec}\left( \boldsymbol{\Sigma} - z^{-1}(\boldsymbol{x} - \boldsymbol{\mu} - z\boldsymbol{\xi})(\boldsymbol{x} - \boldsymbol{\mu} - z\boldsymbol{\xi})^{\top} \right) \right] \\
& \quad \times \mathsf{E}_{\boldsymbol{x}, z | \boldsymbol{y}; \boldsymbol{\theta}} \left[ \mathrm{vec}\left( \boldsymbol{\Sigma} - z^{-1}(\boldsymbol{x} - \boldsymbol{\mu} - z\boldsymbol{\xi})(\boldsymbol{x} - \boldsymbol{\mu} - z\boldsymbol{\xi})^{\top} \right)^{\top} \right] \\
&= \frac{1}{4} \mathsf{Cov}_{\boldsymbol{x}, z | \boldsymbol{y}; \boldsymbol{\theta}} \left[ \mathrm{vec}\left( \boldsymbol{\Sigma} - z^{-1}(\boldsymbol{x} - \boldsymbol{\mu} - z\boldsymbol{\xi})(\boldsymbol{x} - \boldsymbol{\mu} - z\boldsymbol{\xi})^{\top} \right) \right] - \frac{1}{2} \boldsymbol{\Sigma} \otimes \boldsymbol{\Sigma}.
\end{aligned}
\tag{59}
$$

For the covariance term, by the law of total variance, we have

$$
\begin{aligned}
& \mathsf{Cov}_{\boldsymbol{x}, z | \boldsymbol{y}; \boldsymbol{\theta}} \left[ \mathrm{vec}\left( \boldsymbol{\Sigma} - z^{-1}(\boldsymbol{x} - \boldsymbol{\mu} - z\boldsymbol{\xi})(\boldsymbol{x} - \boldsymbol{\mu} - z\boldsymbol{\xi})^{\top} \right) \right] \\
&= \mathsf{Cov}_{\boldsymbol{x}, z | \boldsymbol{y}; \boldsymbol{\theta}} \left[ \mathrm{vec}\left( z^{-1}(\boldsymbol{x} - \boldsymbol{\mu} - z\boldsymbol{\xi})(\boldsymbol{x} - \boldsymbol{\mu} - z\boldsymbol{\xi})^{\top} \right) \right] \\
&= \mathsf{E}_{\boldsymbol{x}, z | \boldsymbol{y}; \boldsymbol{\theta}} \left[ \mathsf{Cov}_{\boldsymbol{x} | z, \boldsymbol{y}; \boldsymbol{\theta}} \left[ \mathrm{vec}\left( z^{-1}(\boldsymbol{x} - \boldsymbol{\mu} - z\boldsymbol{\xi})(\boldsymbol{x} - \boldsymbol{\mu} - z\boldsymbol{\xi})^{\top} \right) \right] \right] \\
& \quad + \mathsf{Cov}_{\boldsymbol{x}, z | \boldsymbol{y}; \boldsymbol{\theta}} \left[ \mathsf{E}_{\boldsymbol{x} | z, \boldsymbol{y}; \boldsymbol{\theta}} \left[ \mathrm{vec}\left( z^{-1}(\boldsymbol{x} - \boldsymbol{\mu} - z\boldsymbol{\xi})(\boldsymbol{x} - \boldsymbol{\mu} - z\boldsymbol{\xi})^{\top} \right) \right] \right].
\end{aligned}
$$

The second term is equal to a zero matrix based on $\mathsf{E}_{\boldsymbol{x} | z, \boldsymbol{y}; \boldsymbol{\theta}} \left[ \boldsymbol{x} - \boldsymbol{\mu} - z\boldsymbol{\xi} \right] = \boldsymbol{0}$. Setting $\boldsymbol{X} = \boldsymbol{x} - \boldsymbol{\mu} - z\boldsymbol{\xi}$, since $\nabla^2_{\boldsymbol{x} - \boldsymbol{\mu} - z\boldsymbol{\xi}} \log p(\boldsymbol{x}, z \mid \boldsymbol{y}; \boldsymbol{\theta}) = -\frac{\boldsymbol{\Sigma}^{-1}}{z}$, $p(\boldsymbol{x}, z \mid \boldsymbol{y}; \boldsymbol{\theta})$ is a log-concave function with respect to $\boldsymbol{x} - \boldsymbol{\mu} - z\boldsymbol{\xi}$. Based on the Brascamp–Lieb inequality in Lemma 6, we have

$$
\begin{aligned}
\mathsf{Cov}_{\boldsymbol{x} | z, \boldsymbol{y}; \boldsymbol{\theta}} \left( z^{-1} \mathrm{vec}\left( (\boldsymbol{x} - \boldsymbol{\mu} - z\boldsymbol{\xi})(\boldsymbol{x} - \boldsymbol{\mu} - z\boldsymbol{\xi})^{\top} \right) \right) &\prec \mathsf{E}_{\boldsymbol{x} | z, \boldsymbol{y}; \boldsymbol{\theta}} \left[ z^{-1} \mathrm{vec}\left( (\boldsymbol{x} - \boldsymbol{\mu} - z\boldsymbol{\xi})(\boldsymbol{x} - \boldsymbol{\mu} - z\boldsymbol{\xi})^{\top} \right) \right] \otimes \boldsymbol{\Sigma} \\
&= (\boldsymbol{I}_{d \times d} + \boldsymbol{K}_{d \times d}) \boldsymbol{\Sigma} \otimes \boldsymbol{\Sigma},
\end{aligned}
$$

where $\boldsymbol{K}_{d \times d}$ is the commutation matrix. Hence, the expression (59) can further becomes

$$\frac{\partial \mathrm{vec}\left( \nabla_{\boldsymbol{\Sigma}^{-1}} p(\boldsymbol{x}, z; \boldsymbol{\theta}) \right)}{\partial \mathrm{vec}\left( \boldsymbol{\Sigma}^{-1} \right)} \prec (\boldsymbol{K}_{d \times d} - \boldsymbol{I}_{d \times d}) \boldsymbol{\Sigma} \otimes \boldsymbol{\Sigma},$$

which is negative definite.

## C. The Proof of Proposition 3

In this section, we prove the global linear convergence of our proposed ECM algorithm with respect to the three parameters: $\boldsymbol{\mu}$, $\boldsymbol{\Sigma}$, and $\boldsymbol{\xi}$. Since the ECM algorithm can always converge to the stationary points (McLachlan & Krishnan, 2008), we define $\boldsymbol{\mu}^{\star}$, $\boldsymbol{\Sigma}^{\star}$, and $\boldsymbol{\xi}^{\star}$ as the stationary points of the optimization problem (6). In the following, we give the convergence analysis of these three parameters with the convergence rate, respectively.

## C.1. Convergence Rate of Location Parameter

Given the update rule of $\boldsymbol{\mu}$ [3] in (15), we have

$$\boldsymbol{\mu} = \frac{\sum_{t=1}^{n}(\boldsymbol{v}_t - \underline{\boldsymbol{\xi}})}{\sum_{t=1}^{n}\iota_t} = \frac{\sum_{t=1}^{n}(\mathsf{E}_{\boldsymbol{x}_t, z_t | \boldsymbol{y}_t; \underline{\boldsymbol{\theta}}}[z^{-1}\boldsymbol{x}] - \underline{\boldsymbol{\xi}})}{\sum_{t=1}^{n}\mathsf{E}_{z_t | \boldsymbol{y}_t; \underline{\boldsymbol{\theta}}}[z^{-1}]}. \tag{60}$$

Based on (33) and (40), the update rule of $\boldsymbol{\mu}$ in (60) becomes

$$\boldsymbol{\mu} = \underline{\boldsymbol{\mu}} + \left(\sum_{i=1}^{n}\frac{p_{-1}(\boldsymbol{y}_i; \underline{\boldsymbol{\theta}})}{p(\boldsymbol{y}_i; \underline{\boldsymbol{\theta}})}\right)^{-1}\sum_{t=1}^{n}p^{-1}(\boldsymbol{y}_t; \underline{\boldsymbol{\theta}})\underline{\boldsymbol{\Sigma}}\boldsymbol{q}_t \tag{61}$$

Hence, the distance between $\boldsymbol{\mu}$ at the current iteration and $\boldsymbol{\mu}^{\star}$ is given by

$$\|\boldsymbol{\mu} - \boldsymbol{\mu}^{\star}\|_2 = \left\|\underline{\boldsymbol{\mu}} - \boldsymbol{\mu}^{\star} + \left(\sum_{i=1}^{n}\frac{p_{-1}(\boldsymbol{y}_i; \underline{\boldsymbol{\theta}})}{p(\boldsymbol{y}_i; \underline{\boldsymbol{\theta}})}\right)^{-1}\sum_{t=1}^{n}p^{-1}(\boldsymbol{y}_t; \underline{\boldsymbol{\theta}})\underline{\boldsymbol{\Sigma}}\boldsymbol{q}_t\right\|_2. \tag{62}$$

To further analyze the term on the right side in (62), we first give a result of $\boldsymbol{q}$.

**Lemma 7.** *When $\boldsymbol{x}$ follows a normal variance-mean mixture and $\boldsymbol{q}$ is defined in (23), we have*

$$\boldsymbol{q} = \nabla_{\boldsymbol{\mu}} p(\boldsymbol{y}; \boldsymbol{\theta}).$$

*Proof.* Consider the partial derivative of $p(\boldsymbol{y}; \boldsymbol{\theta})$ with respect to the $i$-th element of $\boldsymbol{\mu}$. We have that

$$\nabla_{\mu_i} p(\boldsymbol{y}; \boldsymbol{\theta}) = \nabla_{\mu_i}\int_{\mathcal{Q}^{-1}(\boldsymbol{y})} p(\boldsymbol{x}; \boldsymbol{\theta})\mathrm{d}\boldsymbol{x} = \int_{\mathcal{Q}^{-1}(\boldsymbol{y})}\nabla_{\mu_i}p(\boldsymbol{x}; \boldsymbol{\theta})\mathrm{d}\boldsymbol{x},$$

where the interchange of the integration and differentiation operations in the second equal sign is valid as the Leibniz integral rule (Casella & Berger, 2024). Since the partial derivatives of $p(\boldsymbol{x}; \boldsymbol{\theta})$

$$\nabla_{\mu_i} p(\boldsymbol{x} \mid z; \boldsymbol{\theta}) = -\nabla_{x_i}p(\boldsymbol{x} \mid z; \boldsymbol{\theta}),$$

we can obtain that

$$\begin{aligned}\nabla_{\mu_i} p(\boldsymbol{y}; \boldsymbol{\theta}) &= -\int_0^{+\infty}\int_{\mathcal{Q}^{-1}(\boldsymbol{y}_{\backslash i})}\int_{l_i}^{u_i}\nabla_{x_i}p(\boldsymbol{x} \mid z; \boldsymbol{\theta})p(z)\mathrm{d}x_i\mathrm{d}\boldsymbol{x}_{\backslash i}\mathrm{d}z\\ &= \int_0^{+\infty}\int_{\mathcal{Q}^{-1}(\boldsymbol{y}_{\backslash i})}p(z)\left(p(\boldsymbol{x}_{\backslash i}, x_i = l_i \mid z; \boldsymbol{\theta}) - p(\boldsymbol{x}_{\backslash i}, x_i = u_i \mid z; \boldsymbol{\theta})\right)\mathrm{d}\boldsymbol{x}_{\backslash i}\mathrm{d}z\\ &= p(x_i = l_i, \boldsymbol{y}_{\backslash i}, \boldsymbol{\theta}) - p(x_i = u_i, \boldsymbol{y}_{\backslash i}, \boldsymbol{\theta}),\end{aligned}$$

which is equivalent to the definition of the $i$-th element of $\boldsymbol{q}$ in (23).

Based on Lemma 7, we have $p^{-1}(\boldsymbol{y}; \boldsymbol{\theta})\boldsymbol{q} = \nabla_{\boldsymbol{\mu}}\log p(\boldsymbol{y}; \boldsymbol{\theta})$ and $\boldsymbol{q}^{\star} = \boldsymbol{0}$. Then the distance (62) becomes

$$\begin{aligned}\|\boldsymbol{\mu} - \boldsymbol{\mu}^{\star}\|_2 &= \left\|\underline{\boldsymbol{\mu}} - \boldsymbol{\mu}^{\star} + \left(\sum_{i=1}^{n}\frac{p_{-1}(\boldsymbol{y}_i; \underline{\boldsymbol{\theta}})}{p(\boldsymbol{y}_i; \underline{\boldsymbol{\theta}})}\right)^{-1}\sum_{t=1}^{n}\underline{\boldsymbol{\Sigma}}\nabla_{\underline{\boldsymbol{\mu}}}\log p(\boldsymbol{y}_t; \underline{\boldsymbol{\theta}})\right.\\ &\qquad\left. - \left(\sum_{i=1}^{n}\frac{p_{-1}(\boldsymbol{y}_i; \boldsymbol{\theta}^{\star})}{p(\boldsymbol{y}_i; \boldsymbol{\theta}^{\star})}\right)^{-1}\sum_{t=1}^{n}\underline{\boldsymbol{\Sigma}}\nabla_{\boldsymbol{\mu}^{\star}}\log p(\boldsymbol{y}_t; \boldsymbol{\theta}^{\star})\right\|_2\\ &= \left\|\underline{\boldsymbol{\mu}} - \boldsymbol{\mu}^{\star} + \int_0^1\left(\sum_{i=1}^{n}\frac{p_{-1}(\boldsymbol{y}_i; \tilde{\boldsymbol{\theta}}_{\mu})}{p(\boldsymbol{y}_i; \tilde{\boldsymbol{\theta}}_{\mu})}\right)^{-1}\sum_{t=1}^{n}\underline{\boldsymbol{\Sigma}}\nabla_{\tilde{\boldsymbol{\mu}}}^2\log p(\boldsymbol{y}_t; \tilde{\boldsymbol{\theta}}_{\mu})(\tilde{\boldsymbol{\mu}} - \boldsymbol{\mu}^{\star})\mathrm{d}\beta\right\|_2\\ &\leq \sup_{\beta \in (0,1]}\left\|\boldsymbol{I} + \left(\sum_{i=1}^{n}\frac{p_{-1}(\boldsymbol{y}_i; \tilde{\boldsymbol{\theta}}_{\mu})}{p(\boldsymbol{y}_i; \tilde{\boldsymbol{\theta}}_{\mu})}\right)^{-1}\sum_{t=1}^{n}\underline{\boldsymbol{\Sigma}}\nabla_{\tilde{\boldsymbol{\mu}}}^2\log p(\boldsymbol{y}_t; \tilde{\boldsymbol{\theta}}_{\mu})\right\|_2\|\underline{\boldsymbol{\mu}} - \boldsymbol{\mu}^{\star}\|_2,\end{aligned}$$

---

[3]For simplicity, we use $\underline{\boldsymbol{\mu}}$, $\underline{\boldsymbol{\xi}}$, and $\underline{\boldsymbol{\Sigma}}$ to denote the parameters $\boldsymbol{\mu}^{(k)}$, $\boldsymbol{\xi}^{(k)}$, and $\boldsymbol{\Sigma}^{(k)}$ before the $k$-th update, and use $\boldsymbol{\mu}$, $\boldsymbol{\xi}$, and $\boldsymbol{\Sigma}$ to denote the updated parameters $\boldsymbol{\mu}^{(k+1)}$, $\boldsymbol{\xi}^{(k+1)}$, and $\boldsymbol{\Sigma}^{(k+1)}$.

where the second equation is given by the mean value theorem of integrals and $\tilde{\boldsymbol{\theta}}_\mu = \{\tilde{\boldsymbol{\mu}}, \underline{\boldsymbol{\Sigma}}, \underline{\boldsymbol{\xi}}\}$ with $\tilde{\boldsymbol{\mu}} = \beta\underline{\boldsymbol{\mu}} + (1-\beta)\boldsymbol{\mu}^\star$ and $\beta \in (0, 1]$.

To bound the term $\sup_{\beta \in (0,1]} \left\| \boldsymbol{I} + \left(\sum_{i=1}^n \frac{p_{-1}(\boldsymbol{y}_i; \tilde{\boldsymbol{\theta}}_\mu)}{p(\boldsymbol{y}_i; \tilde{\boldsymbol{\theta}}_\mu)}\right)^{-1} \sum_{t=1}^n \boldsymbol{\Sigma} \nabla_{\tilde{\boldsymbol{\mu}}}^2 \log p(\boldsymbol{y}_t; \tilde{\boldsymbol{\theta}}_\mu) \right\|_2$, we have a result.

**Lemma 8.** *The second order derivative of $\log p(\boldsymbol{y}; \boldsymbol{\theta})$ with respect to $\boldsymbol{\mu}$, i.e., $\nabla_{\boldsymbol{\mu}}^2 \log p(\boldsymbol{y}; \boldsymbol{\theta})$, satisfy that $\boldsymbol{I} + \left(\sum_{i=1}^n \frac{p_{-1}(\boldsymbol{y}_i; \boldsymbol{\theta})}{p(\boldsymbol{y}_i; \boldsymbol{\theta})}\right)^{-1} \sum_{t=1}^n \boldsymbol{\Sigma} \nabla_{\boldsymbol{\mu}}^2 \log p(\boldsymbol{y}_t; \boldsymbol{\theta})$ is a positive definite matrix, for all $\boldsymbol{\theta}$ in the feasible set.*

*Proof.* Substituting $\nabla_{\boldsymbol{\mu}}^2 \log p(\boldsymbol{y}; \boldsymbol{\theta})$ in (48) into $\boldsymbol{I} + \left(\sum_{i=1}^n \frac{p_{-1}(\boldsymbol{y}_i; \boldsymbol{\theta})}{p(\boldsymbol{y}_i; \boldsymbol{\theta})}\right)^{-1} \sum_{t=1}^n \boldsymbol{\Sigma} \nabla_{\boldsymbol{\mu}}^2 \log p(\boldsymbol{y}_t; \boldsymbol{\theta})$, we have

$$\boldsymbol{I} + \left(\sum_{i=1}^n \frac{p_{-1}(\boldsymbol{y}_i; \boldsymbol{\theta})}{p(\boldsymbol{y}_i; \boldsymbol{\theta})}\right)^{-1} \sum_{t=1}^n \boldsymbol{\Sigma} \nabla_{\boldsymbol{\mu}}^2 \log p(\boldsymbol{y}_t; \boldsymbol{\theta}) = \frac{\sum_{t=1}^n \boldsymbol{\Sigma}^{-1} \mathsf{Cov}_{\boldsymbol{x}, z | \boldsymbol{y}_t; \boldsymbol{\theta}} \left[z^{-1} \boldsymbol{x}\right]}{\sum_{t=1}^n \mathsf{E}_{z | \boldsymbol{y}_t; \boldsymbol{\theta}} [z^{-1}]}.$$

Since both $\boldsymbol{\Sigma}$ and $\mathsf{Cov}_{\boldsymbol{x}, z | \boldsymbol{y}; \boldsymbol{\theta}} \left[z^{-1} \boldsymbol{x}_t\right]$ are positive definite matrices, $\boldsymbol{I} + \left(\sum_{i=1}^n \frac{p_{-1}(\boldsymbol{y}_i; \boldsymbol{\theta})}{p(\boldsymbol{y}_i; \boldsymbol{\theta})}\right)^{-1} \sum_{t=1}^n \boldsymbol{\Sigma} \nabla_{\boldsymbol{\mu}}^2 \log p(\boldsymbol{y}_t; \boldsymbol{\theta})$ is positive definite. $\square$

Based on Theorem 2 and Lemma 8, we have that all the eigenvalues of matrix $\boldsymbol{I} + \boldsymbol{\Sigma} \nabla_{\boldsymbol{\mu}}^2 \log p(\boldsymbol{y}; \boldsymbol{\theta})$ for any $\boldsymbol{\theta}$ are belongs to $(0, 1)$, hence we have

$$\sup_{\beta \in (0,1]} \left\| \boldsymbol{I} + \left(\sum_{i=1}^n \frac{p_{-1}(\boldsymbol{y}_i; \tilde{\boldsymbol{\theta}}_\mu)}{p(\boldsymbol{y}_i; \tilde{\boldsymbol{\theta}}_\mu)}\right)^{-1} \sum_{t=1}^n \boldsymbol{\Sigma} \nabla_{\tilde{\boldsymbol{\mu}}}^2 \log p(\boldsymbol{y}_t; \tilde{\boldsymbol{\theta}}_\mu) \right\|_2 = \sup_{\beta \in (0,1]} \left\| \frac{\sum_{t=1}^n \boldsymbol{\Sigma}^{-1} \mathsf{Cov}_{\boldsymbol{x}, z | \boldsymbol{y}_t; \tilde{\boldsymbol{\theta}}_\mu} \left[z^{-1} \boldsymbol{x}\right]}{\sum_{t=1}^n \mathsf{E}_{z | \boldsymbol{y}_t; \tilde{\boldsymbol{\theta}}_\mu} [z^{-1}]} \right\|_2$$

$$\leq \max_{\boldsymbol{\theta}} \left\| \frac{\sum_{t=1}^n \boldsymbol{\Sigma}^{-1} \mathsf{Cov}_{\boldsymbol{x}, z | \boldsymbol{y}_t; \boldsymbol{\theta}} \left[z^{-1} \boldsymbol{x}\right]}{\sum_{t=1}^n \mathsf{E}_{z | \boldsymbol{y}_t; \boldsymbol{\theta}} [z^{-1}]} \right\|_2$$

$$\triangleq c_\mu \in (0, 1).$$

$\square$

### C.2. Convergence Rate of skewness parameter

Given the update rule of $\boldsymbol{\xi}$ in (15), we have

$$\boldsymbol{\xi} = \frac{\sum_{t=1}^n (\boldsymbol{w}_t - \boldsymbol{\mu})}{\sum_{t=1}^n \zeta_t} = \frac{\sum_{t=1}^n (\mathsf{E}_{\boldsymbol{x} | \boldsymbol{y}_t; \boldsymbol{\theta}}[\boldsymbol{x}] - \boldsymbol{\mu})}{\sum_{t=1}^n \mathsf{E}_{z_t | \boldsymbol{y}_t; \underline{\boldsymbol{\theta}}}[z]}. \tag{63}$$

Based on (38), we have

$$\mathsf{E}_{\boldsymbol{x} | \boldsymbol{y}; \boldsymbol{\theta}}[\boldsymbol{x}] - \boldsymbol{\mu} = \mathsf{E}_{z | \boldsymbol{y}; \boldsymbol{\theta}}[z] \boldsymbol{\xi} + p^{-1}(\boldsymbol{y}; \boldsymbol{\theta}) \boldsymbol{\Sigma} \boldsymbol{q}_1.$$

Hence, the update rule of $\boldsymbol{\xi}$ in (63) becomes

$$\boldsymbol{\xi} = \underline{\boldsymbol{\xi}} + \left(\sum_{i=1}^n \frac{p_1(\boldsymbol{y}_i; \underline{\boldsymbol{\theta}})}{p(\boldsymbol{y}_i; \underline{\boldsymbol{\theta}})}\right)^{-1} \sum_{t=1}^n p^{-1}(\boldsymbol{y}_t; \underline{\boldsymbol{\theta}}) \boldsymbol{\Sigma} \boldsymbol{q}_{1,t} \tag{64}$$

Hence, the distance between $\boldsymbol{\xi}$ at the current iteration and $\boldsymbol{\xi}^\star$ is given by

$$\|\boldsymbol{\xi} - \boldsymbol{\xi}^\star\|_2 = \left\| \underline{\boldsymbol{\xi}} - \boldsymbol{\xi}^\star + \left(\sum_{i=1}^n \frac{p_1(\boldsymbol{y}_i; \underline{\boldsymbol{\theta}})}{p(\boldsymbol{y}_i; \underline{\boldsymbol{\theta}})}\right)^{-1} \sum_{t=1}^n p^{-1}(\boldsymbol{y}_t; \underline{\boldsymbol{\theta}}) \boldsymbol{\Sigma} \boldsymbol{q}_{1,t} \right\|_2. \tag{65}$$

To further analyze the term on the right side in (65), we first give a result of $\boldsymbol{q}_1$.

**Lemma 9.** *When $\boldsymbol{x}$ follows a normal variance-mean mixture and $\boldsymbol{q}_1$ is defined in (23), we have*

$$\boldsymbol{q}_1 = \nabla_{\boldsymbol{\xi}} p(\boldsymbol{y}; \boldsymbol{\theta}).$$

*Proof.* Consider the partial derivative of $p(\boldsymbol{y}; \boldsymbol{\theta})$ with respect to the $i$-th element of $\boldsymbol{\xi}$. We have that

$$\nabla_{\xi_i} p(\boldsymbol{y}; \boldsymbol{\theta}) = \nabla_{\xi_i} \int_{\mathcal{Q}^{-1}(\boldsymbol{y})} p(\boldsymbol{x}; \boldsymbol{\theta}) \mathrm{d}\boldsymbol{x} = \int_{\mathcal{Q}^{-1}(\boldsymbol{y})} \nabla_{\xi_i} p(\boldsymbol{x}; \boldsymbol{\theta}) \mathrm{d}\boldsymbol{x}.$$

Since the partial derivatives of $p(\boldsymbol{x}; \boldsymbol{\theta})$

$$\nabla_{\xi_i} p(\boldsymbol{x} \mid z; \boldsymbol{\theta}) = -z \nabla_{x_i} p(\boldsymbol{x} \mid z; \boldsymbol{\theta}),$$

we can obtain that

$$\nabla_{\xi_i} p(\boldsymbol{y}; \boldsymbol{\theta}) = -\int_0^{+\infty} \int_{\mathcal{Q}^{-1}(\boldsymbol{y}_{\setminus i})} \int_{l_i}^{u_i} z \nabla_{x_i} p(\boldsymbol{x} \mid z; \boldsymbol{\theta}) p(z) \mathrm{d}x_i \mathrm{d}\boldsymbol{x}_{\setminus i} \mathrm{d}z$$

$$= \int_0^{+\infty} \int_{\mathcal{Q}^{-1}(\boldsymbol{y}_{\setminus i})} \frac{p(z)}{z} z \left( p(\boldsymbol{x}_{\setminus i}, x_i = l_i \mid z; \boldsymbol{\theta}) - p(\boldsymbol{x}_{\setminus i}, x_i = u_i \mid z; \boldsymbol{\theta}) \right) \mathrm{d}\boldsymbol{x}_{\setminus i} \mathrm{d}z$$

$$= p(x_i = l_i, \boldsymbol{y}_{\setminus i} \mid z, \boldsymbol{\theta}) - p(x_i = u_i, \boldsymbol{y}_{\setminus i} \mid z, \boldsymbol{\theta}).$$

which is equivalent to the definition of the $i$-th element of $\boldsymbol{q}_1$ from (35).

Based on Lemma 9, we have $p^{-1}(\boldsymbol{y}; \boldsymbol{\theta}) \boldsymbol{q}_1 = \nabla_{\boldsymbol{\xi}} \log p(\boldsymbol{y}; \boldsymbol{\theta})$ and $\boldsymbol{q}_1^{\star} = \boldsymbol{0}$. Then the distance (62) becomes

$$\|\boldsymbol{\xi} - \boldsymbol{\xi}^{\star}\|_2 = \left\| \underline{\boldsymbol{\xi}} - \boldsymbol{\xi}^{\star} + \left( \sum_{i=1}^n \frac{p_1(\boldsymbol{y}_i; \underline{\boldsymbol{\theta}})}{p(\boldsymbol{y}_i; \underline{\boldsymbol{\theta}})} \right)^{-1} \sum_{t=1}^n \underline{\boldsymbol{\Sigma}} \nabla_{\underline{\boldsymbol{\xi}}} \log p(\boldsymbol{y}; \underline{\boldsymbol{\theta}}) \right.$$

$$\left. - \left( \sum_{i=1}^n \frac{p_1(\boldsymbol{y}_i; \boldsymbol{\theta}^{\star})}{p(\boldsymbol{y}_i; \boldsymbol{\theta}^{\star})} \right)^{-1} \sum_{t=1}^n \boldsymbol{\Sigma}^{\star} \nabla_{\boldsymbol{\xi}^{\star}} \log p(\boldsymbol{y}; \boldsymbol{\theta}^{\star}) \right\|_2$$

$$= \left\| \underline{\boldsymbol{\xi}} - \boldsymbol{\xi}^{\star} + \int_0^1 \left( \sum_{i=1}^n \frac{p_1(\boldsymbol{y}_i; \tilde{\boldsymbol{\theta}}_{\xi})}{p(\boldsymbol{y}_i; \tilde{\boldsymbol{\theta}}_{\xi})} \right)^{-1} \sum_{t=1}^n \tilde{\boldsymbol{\Sigma}} \nabla_{\tilde{\boldsymbol{\xi}}}^2 \log p(\boldsymbol{y}_t; \tilde{\boldsymbol{\theta}}_{\xi})(\tilde{\boldsymbol{\xi}} - \boldsymbol{\xi}^{\star}) \mathrm{d}\beta \right\|_2$$

$$\leq \sup_{\beta \in (0,1]} \left\| \boldsymbol{I} + \left( \sum_{i=1}^n \frac{p_1(\boldsymbol{y}_i; \tilde{\boldsymbol{\theta}}_{\xi})}{p(\boldsymbol{y}_i; \tilde{\boldsymbol{\theta}}_{\xi})} \right)^{-1} \sum_{t=1}^n \tilde{\boldsymbol{\Sigma}} \nabla_{\tilde{\boldsymbol{\xi}}}^2 \log p(\boldsymbol{y}_t; \tilde{\boldsymbol{\theta}}_{\xi}) \right\|_2 \|\underline{\boldsymbol{\xi}} - \boldsymbol{\xi}^{\star}\|_2,$$

where the second equation is given by the mean value theorem of integrals and $\tilde{\boldsymbol{\theta}}_{\xi} = \left\{ \boldsymbol{\mu}, \boldsymbol{\Sigma}, \tilde{\boldsymbol{\xi}} \right\}$ with $\tilde{\boldsymbol{\xi}} = \beta \underline{\boldsymbol{\xi}} + (1 - \beta) \boldsymbol{\xi}^{\star}$, and $\beta \in (0, 1]$.

To bound the term $\sup_{\beta \in (0,1]} \left\| \boldsymbol{I} + \left( \sum_{i=1}^n \frac{p_{-1}(\boldsymbol{y}_i; \tilde{\boldsymbol{\theta}}_{\xi})}{p(\boldsymbol{y}_i; \tilde{\boldsymbol{\theta}}_{\xi})} \right)^{-1} \sum_{t=1}^n \tilde{\boldsymbol{\Sigma}} \nabla_{\tilde{\boldsymbol{\xi}}}^2 \log p(\boldsymbol{y}_t; \tilde{\boldsymbol{\theta}}_{\xi}) \right\|_2$, we need some results for the second order derivative of $\log p(\boldsymbol{y}_t; \boldsymbol{\theta})$ at first.

**Lemma 10.** *The second order derivative of $\log p(\boldsymbol{y}; \boldsymbol{\theta})$ with respect to $\boldsymbol{\xi}$, i.e., $\nabla_{\boldsymbol{\xi}}^2 \log p(\boldsymbol{y}; \boldsymbol{\theta})$, satisfy that $\boldsymbol{I} + \left( \sum_{i=1}^n \frac{p_{-1}(\boldsymbol{y}_i; \boldsymbol{\theta})}{p(\boldsymbol{y}_i; \boldsymbol{\theta})} \right)^{-1} \sum_{t=1}^n \boldsymbol{\Sigma} \nabla_{\boldsymbol{\xi}}^2 \log p(\boldsymbol{y}_t; \boldsymbol{\theta})$ is a positive definite matrix.*

*Proof.* Substituting $\nabla_{\boldsymbol{\xi}}^2 \log p(\boldsymbol{y}; \boldsymbol{\theta})$ in (55) into $\boldsymbol{I} + \left( \sum_{i=1}^n \frac{p_{-1}(\boldsymbol{y}_i; \boldsymbol{\theta})}{p(\boldsymbol{y}_i; \boldsymbol{\theta})} \right)^{-1} \sum_{t=1}^n \boldsymbol{\Sigma} \nabla_{\boldsymbol{\xi}}^2 \log p(\boldsymbol{y}_t; \boldsymbol{\theta})$, we have

$$\boldsymbol{I} + \left( \sum_{i=1}^n \frac{p_1(\boldsymbol{y}_i; \boldsymbol{\theta})}{p(\boldsymbol{y}_i; \boldsymbol{\theta})} \right)^{-1} \sum_{t=1}^n \boldsymbol{\Sigma} \nabla_{\boldsymbol{\xi}}^2 \log p(\boldsymbol{y}_t; \boldsymbol{\theta}) = \frac{\sum_{t=1}^n \boldsymbol{\Sigma}^{-1} \mathsf{Cov}_{\boldsymbol{x}, z \mid \boldsymbol{y}_t; \boldsymbol{\theta}} [\boldsymbol{x} - z \boldsymbol{\xi}]}{\sum_{t=1}^n \mathsf{E}_{z \mid \boldsymbol{y}_t; \boldsymbol{\theta}} [z]}.$$

Since both $\boldsymbol{\Sigma}$ and $\mathsf{Cov}_{\boldsymbol{x}, z \mid \boldsymbol{y}; \boldsymbol{\theta}} [\boldsymbol{x}_t - z \boldsymbol{\xi}]$ are positive definite matrices, $\boldsymbol{I} + \left( \sum_{i=1}^n \frac{p_{-1}(\boldsymbol{y}_i; \boldsymbol{\theta})}{p(\boldsymbol{y}_i; \boldsymbol{\theta})} \right)^{-1} \sum_{t=1}^n \boldsymbol{\Sigma} \nabla_{\boldsymbol{\xi}}^2 \log p(\boldsymbol{y}_t; \boldsymbol{\theta})$ is positive definite. $\square$

Based on Theorem 2 and Lemma 10, we have that all the eigenvalues of matrix $\boldsymbol{I} + \boldsymbol{\Sigma}\nabla_{\tilde{\boldsymbol{\xi}}}^2 \log p(\boldsymbol{y}; \boldsymbol{\theta})$ for any $\boldsymbol{\theta}$ are belong to $(0,1)$, hence we have

$$
\sup_{\beta \in (0,1]} \left\| \boldsymbol{I} + \left( \sum_{i=1}^{n} \frac{p_1(\boldsymbol{y}_i; \tilde{\boldsymbol{\theta}}_\xi)}{p(\boldsymbol{y}_i; \tilde{\boldsymbol{\theta}}_\xi)} \right)^{-1} \sum_{t=1}^{n} \underline{\boldsymbol{\Sigma}} \nabla_{\tilde{\boldsymbol{\xi}}}^2 \log p(\boldsymbol{y}_t; \tilde{\boldsymbol{\theta}}_\xi) \right\|_2 = \sup_{\beta \in (0,1]} \left\| \frac{\sum_{t=1}^{n} \boldsymbol{\Sigma}^{-1} \mathsf{Cov}_{\boldsymbol{x},z|\boldsymbol{y}_t; \tilde{\boldsymbol{\theta}}_\xi}[\boldsymbol{x} - z\boldsymbol{\xi}]}{\sum_{t=1}^{n} \mathsf{E}_{z|\boldsymbol{y}_t; \tilde{\boldsymbol{\theta}}_\xi}[z]} \right\|_2
$$

$$
\leq \max_{\boldsymbol{\theta}} \left\| \frac{\sum_{t=1}^{n} \boldsymbol{\Sigma}^{-1} \mathsf{Cov}_{\boldsymbol{x},z|\boldsymbol{y}_t; \boldsymbol{\theta}}[\boldsymbol{x} - z\boldsymbol{\xi}]}{\sum_{t=1}^{n} \mathsf{E}_{z|\boldsymbol{y}_t; \boldsymbol{\theta}}[z]} \right\|_2
$$

$$
\triangleq c_\xi \in (0,1).
$$

□

### C.3. Convergence Rate of Scatter Matrix

Given the update rule of $\boldsymbol{\Sigma}$ in (15), we have

$$
\boldsymbol{\Sigma} = \frac{1}{n} \sum_{t=1}^{n} \left( \left( \boldsymbol{U}_t - 2\boldsymbol{v}_t\underline{\boldsymbol{\mu}}^\top + \iota_t\underline{\boldsymbol{\mu}}\underline{\boldsymbol{\mu}}^\top \right) - 2(\boldsymbol{w}_t - \underline{\boldsymbol{\mu}})\boldsymbol{\xi}^\top + \zeta_t\underline{\boldsymbol{\xi}}\underline{\boldsymbol{\xi}}^\top \right)
$$

$$
= \frac{1}{n} \sum_{t=1}^{n} \left( \mathsf{E}_{\boldsymbol{x}_t,z_t|\boldsymbol{y}_t; \underline{\boldsymbol{\theta}}}[z^{-1}\boldsymbol{x}\boldsymbol{x}^\top] - 2\mathsf{E}_{\boldsymbol{x}_t,z_t|\boldsymbol{y}_t; \underline{\boldsymbol{\theta}}}[z^{-1}\boldsymbol{x}_t]\underline{\boldsymbol{\mu}}^\top + \mathsf{E}_{z_t|\boldsymbol{y}_t; \underline{\boldsymbol{\theta}}}[z^{-1}]\underline{\boldsymbol{\mu}}\underline{\boldsymbol{\mu}}^\top - 2(\mathsf{E}_{\boldsymbol{x}|\boldsymbol{y}_t; \underline{\boldsymbol{\theta}}}[\boldsymbol{x}] - \underline{\boldsymbol{\mu}})\underline{\boldsymbol{\xi}}^\top + \mathsf{E}_{z_t|\boldsymbol{y}_t; \underline{\boldsymbol{\theta}}}[z]\underline{\boldsymbol{\xi}}\underline{\boldsymbol{\xi}}^\top \right).
$$

(66)

In the following, we derive all five terms in the summation in (66).

**Term 1**: Based on (42), we have

$$
\mathsf{E}_{\boldsymbol{x},z|\boldsymbol{y}; \underline{\boldsymbol{\theta}}}\left[ z^{-1}\boldsymbol{x}\boldsymbol{x}^\top \right] = \mathsf{E}_{z|\boldsymbol{y}; \underline{\boldsymbol{\theta}}}\left[ z^{-1} \right] \underline{\boldsymbol{\mu}}\underline{\boldsymbol{\mu}}^\top + \mathsf{E}_{z|\boldsymbol{y}; \underline{\boldsymbol{\theta}}}[z] \underline{\boldsymbol{\xi}}\underline{\boldsymbol{\xi}}^\top + 2\underline{\boldsymbol{\mu}}\underline{\boldsymbol{\xi}}^\top + \underline{\boldsymbol{\Sigma}}
$$

$$
+ p^{-1}(\boldsymbol{y}; \underline{\boldsymbol{\theta}}) \left( 2 \left( \underline{\boldsymbol{\Sigma}}\boldsymbol{q} \right) \underline{\boldsymbol{\mu}}^\top + 2\mathsf{E}_z\left[ z \right] \left( \underline{\boldsymbol{\Sigma}}\boldsymbol{q}_1 \right) \underline{\boldsymbol{\xi}}^\top + \underline{\boldsymbol{\Sigma}} \left( \boldsymbol{H}_1 + \boldsymbol{D}_1 \right) \underline{\boldsymbol{\Sigma}} \right),
$$

where $\boldsymbol{q}, \boldsymbol{q}_1, \boldsymbol{H}_1$ and $\boldsymbol{D}_1$ are defined in (23), (34), (36) and (37), respectively.

**Term 2**: Based on (40), we can obtain

$$
-2\mathsf{E}_{\boldsymbol{x}_t,z_t|\boldsymbol{y}_t; \underline{\boldsymbol{\theta}}}[z_t^{-1}\boldsymbol{x}_t]\underline{\boldsymbol{\mu}}^\top = -2 \left( \mathsf{E}_{z|\boldsymbol{y}; \underline{\boldsymbol{\theta}}}\left[ z^{-1} \right] \underline{\boldsymbol{\mu}}\underline{\boldsymbol{\mu}}^\top + \boldsymbol{\xi} + p^{-1}(\boldsymbol{y}; \underline{\boldsymbol{\theta}})\underline{\boldsymbol{\Sigma}}\boldsymbol{q} \right) \underline{\boldsymbol{\mu}}^\top.
$$

**Term 4**: Based on (38), we have

$$
-2(\mathsf{E}_{\boldsymbol{x}_t|\boldsymbol{y}_t; \underline{\boldsymbol{\theta}}}[\boldsymbol{x}_t] - \underline{\boldsymbol{\mu}})\underline{\boldsymbol{\xi}}^\top = -2\mathsf{E}_{z|\boldsymbol{y}; \underline{\boldsymbol{\theta}}}\left[ z \right] \underline{\boldsymbol{\xi}}\underline{\boldsymbol{\xi}}^\top - 2p^{-1}(\boldsymbol{y}; \boldsymbol{\theta}) \left( \underline{\boldsymbol{\Sigma}}\boldsymbol{q}_1 \right) \underline{\boldsymbol{\xi}}^\top
$$

Upon cancellation of the opposing terms, the update rule in (66) reduces to:

$$
\boldsymbol{\Sigma} = \underline{\boldsymbol{\Sigma}} + \frac{1}{n} \sum_{t=1}^{n} p^{-1}(\boldsymbol{y}_t; \underline{\boldsymbol{\theta}})\underline{\boldsymbol{\Sigma}}(\boldsymbol{H}_{1,t} + \boldsymbol{D}_{1,t})\underline{\boldsymbol{\Sigma}},
$$

(67)

Hence, the distance between $\boldsymbol{\Sigma}$ at the current iteration and $\boldsymbol{\Sigma}^\star$ is given by

$$
\| \boldsymbol{\Sigma} - \boldsymbol{\Sigma}^\star \|_{\mathsf{F}} = \left\| \underline{\boldsymbol{\Sigma}} - \boldsymbol{\Sigma}^\star + \frac{1}{n} \sum_{t=1}^{n} p^{-1}(\boldsymbol{y}_t; \underline{\boldsymbol{\theta}})\underline{\boldsymbol{\Sigma}}(\boldsymbol{H}_{1,t} + \boldsymbol{D}_{1,t})\underline{\boldsymbol{\Sigma}} \right\|_{\mathsf{F}}.
$$

(68)

To further analyze the term on the right side, we first give a result of $\boldsymbol{H}_{1,t} + \boldsymbol{D}_{1,t}$.

**Lemma 11.** *When $\boldsymbol{x}$ follows a normal mean variance mixture and $\boldsymbol{H}_1$ and $\boldsymbol{D}_1$ are defined in (36) and (37), respectively, we have*

$$
\frac{1}{2}(\boldsymbol{H}_1 + \boldsymbol{D}_1) = \nabla_{\boldsymbol{\Sigma}}p(\boldsymbol{y}; \boldsymbol{\theta}).
$$

*Proof.* Considering the partial derivative of $p(\boldsymbol{y}; \boldsymbol{\theta})$ with respect to the $(i, j)$-th element of $\boldsymbol{\Sigma}$, we have that

$$\nabla_{\boldsymbol{\Sigma}_{ij}} p(\boldsymbol{y}; \boldsymbol{\theta}) = \nabla_{\boldsymbol{\Sigma}_{ij}} \int_{\mathcal{Q}^{-1}(\boldsymbol{y})} p(\boldsymbol{x}; \boldsymbol{\theta}) \mathrm{d}\boldsymbol{x} = \int_{\mathcal{Q}^{-1}(\boldsymbol{y})} \nabla_{\boldsymbol{\Sigma}_{ij}} p(\boldsymbol{x}; \boldsymbol{\theta}) \mathrm{d}\boldsymbol{x}.$$

The partial derivatives of $p(\boldsymbol{x}; \boldsymbol{\theta})$ satisfy the following equation:

$$\nabla_{\boldsymbol{\Sigma}_{ij}} p(\boldsymbol{x} \mid z; \boldsymbol{\theta}) = \frac{z}{2} \nabla^2_{x_i x_j} p(\boldsymbol{x} \mid z; \boldsymbol{\theta}), \tag{69}$$

Hence, for $i \neq j$, we can obtain that

$$\begin{aligned}
\nabla_{\boldsymbol{\Sigma}_{ij}} p(\boldsymbol{y}; \boldsymbol{\theta}) &= \int_{\mathcal{Q}^{-1}(\boldsymbol{y}_{\backslash i,j})} \int_{l_i}^{u_i} \int_{l_j}^{u_j} \int_0^{+\infty} \frac{z}{2} \nabla^2_{x_i x_j} p(\boldsymbol{x} \mid z; \boldsymbol{\theta}) p(z) \mathrm{d}z \mathrm{d}x_i \mathrm{d}x_j \mathrm{d}\boldsymbol{x}_{\backslash i,j} \\
&= \int_{\mathcal{Q}^{-1}(\boldsymbol{y}_{\backslash i,j})} \int_0^{+\infty} \frac{zp(z)}{2} \left( p(\boldsymbol{x}_{\backslash i,j}, x_i = l_i, x_j = l_j \mid z; \boldsymbol{\theta}) - p(\boldsymbol{x}_{\backslash i,j}, x_i = l_i, x_j = u_j \mid z; \boldsymbol{\theta}) \right. \\
&\qquad \left. - p(\boldsymbol{x}_{\backslash i,j}, x_i = u_i, x_j = l_j \mid z; \boldsymbol{\theta}) + p(\boldsymbol{x}_{\backslash i,j}, x_i = u_i, x_j = u_j \mid z; \boldsymbol{\theta}) \right) \mathrm{d}z \mathrm{d}\boldsymbol{x}_{\backslash i,j} \\
&= \frac{1}{2} \mathsf{E}_z(z) \left( p_1(\boldsymbol{y}_{\backslash i,j}, x_i = l_i, x_j = l_j; \boldsymbol{\theta}) - p_1(\boldsymbol{y}_{\backslash i,j}, x_i = l_i, x_j = u_j; \boldsymbol{\theta}) \right. \\
&\qquad \left. - p_1(\boldsymbol{y}_{\backslash i,j}, x_i = u_i, x_j = l_j; \boldsymbol{\theta}) + p_1(\boldsymbol{y}_{\backslash i,j}, x_i = u_i, x_j = u_j; \boldsymbol{\theta}) \right) \\
&= \frac{1}{2} \boldsymbol{H}_{1,ij}.
\end{aligned}$$

For the case $i = j$, we first introduce a result.

**Lemma 12.** *When $\boldsymbol{x}$ follows a normal variance-mean mixture, we have the following equation:*

$$\sum_{k=1}^d \boldsymbol{\Sigma}_{ik} \nabla^2_{x_k x_i} p_1(\boldsymbol{x} \mid z; \boldsymbol{\theta}) = \nabla_{x_i} \left( -\frac{p_1(\boldsymbol{x} \mid z; \boldsymbol{\theta})}{z} (x_i - \mu_i - z\xi_i) \right). \tag{70}$$

*Proof.* We begin with

$$\sum_{k=1}^d \boldsymbol{\Sigma}_{ik} \nabla^2_{x_k x_i} p_1(\boldsymbol{x} \mid z; \boldsymbol{\theta}) = \nabla_{x_i} \left( \sum_{k=1}^d \boldsymbol{\Sigma}_{ik} \nabla_{x_k} p_1(\boldsymbol{x} \mid z; \boldsymbol{\theta}) \right)$$

For the term $\sum_{k=1}^d \boldsymbol{\Sigma}_{ik} \nabla_{x_k} p(\boldsymbol{x} \mid z; \boldsymbol{\theta})$, we have

$$\sum_{k=1}^d \boldsymbol{\Sigma}_{ik} \nabla_{x_k} p_1(\boldsymbol{x} \mid z; \boldsymbol{\theta}) = \sum_{k=1}^d \boldsymbol{\Sigma}_{ik} \left( -\frac{p_1(\boldsymbol{x} \mid z; \boldsymbol{\theta})}{z} \sum_{l=1}^d \boldsymbol{\Sigma}^{-1}_{kl}(x_l - \mu_l - z\xi_l) \right) = -\frac{p_1(\boldsymbol{x} \mid z; \boldsymbol{\theta})}{z} \sum_{l=1}^d \left( \sum_{k=1}^d \boldsymbol{\Sigma}_{ik} \boldsymbol{\Sigma}^{-1}_{kl}(x_l - \mu_l - z\xi_l) \right).$$

Since $\sum_{k=1}^d \boldsymbol{\Sigma}_{ik} \boldsymbol{\Sigma}^{-1}_{kl}$ is equivalent to the $(i, l)$-entry of the matrix $\boldsymbol{\Sigma} \boldsymbol{\Sigma}^{-1}$, we can obtain that

$$\sum_{k=1}^d \boldsymbol{\Sigma}_{ik} \boldsymbol{\Sigma}^{-1}_{kl} = \begin{cases} 0, & i \neq l, \\ 1, & i = l. \end{cases}$$

Hence, we have

$$-\frac{p_1(\boldsymbol{x} \mid z; \boldsymbol{\theta})}{z} \sum_{k=1}^d \left( \sum_{l=1}^d \boldsymbol{\Sigma}_{ik} \boldsymbol{\Sigma}^{-1}_{kl}(x_l - \mu_l - z\xi_l) \right) = -\frac{p_1(\boldsymbol{x} \mid z; \boldsymbol{\theta})}{z} (x_i - \mu_i - z\xi_i).$$

Therefore, we prove the equation (70). $\qquad \square$

Computing the integral of the left side of (70), we have

$$
\int_{\mathcal{Q}^{-1}(\boldsymbol{y})} \int_0^{+\infty} \sum_{k=1}^d \boldsymbol{\Sigma}_{ik} \nabla^2_{x_k x_i} p_1(\boldsymbol{x} \mid z; \boldsymbol{\theta}) p(z) \mathrm{d}z \mathrm{d}\boldsymbol{x}
$$
$$
= \int_0^{+\infty} \int_{\mathcal{Q}^{-1}(\boldsymbol{y})} \left( \sigma_i \nabla^2_{x_i} p_1(\boldsymbol{x} \mid z; \boldsymbol{\theta}) + \sum_{k \neq i} \boldsymbol{\Sigma}_{ik} \nabla^2_{x_k x_i} p_1(\boldsymbol{x} \mid z; \boldsymbol{\theta}) \right) p(z) \mathrm{d}z \mathrm{d}\boldsymbol{x} \tag{71}
$$
$$
= 2\mathsf{E}_z[z^{-1}] \sigma_i \nabla_{\sigma_i} p(\boldsymbol{y}; \boldsymbol{\theta}) + [\boldsymbol{\Sigma} \boldsymbol{H}_1]_{ii}.
$$

Then the integral of the right side of (70) is given by

$$
\int_{\mathcal{Q}^{-1}(\boldsymbol{y})} \int_0^{+\infty} \nabla_{x_i} \left( -\frac{p_1(\boldsymbol{x}; \boldsymbol{\theta})}{z} (x_i - \mu_i - z\xi_i) \right) \mathrm{d}z \mathrm{d}\boldsymbol{x} = \int_{\mathcal{Q}^{-1}(\boldsymbol{y}_{\backslash i})} \int_{l_i}^{u_i} \int_0^{+\infty} \nabla_{x_i} \left( -\frac{p_1(\boldsymbol{x}; \boldsymbol{\theta})}{z} (x_i - \mu_i - z\xi_i) \right) \mathrm{d}z \mathrm{d}x_i \mathrm{d}\boldsymbol{x}_{\backslash i}
$$
$$
= \mathsf{E}_z(z^{-1}) \Big( (l_i - \mu_i) p(x_i = l_i, \boldsymbol{y}_{\backslash i}; \boldsymbol{\theta}) - \xi_i p_1(x_i = l_i, \boldsymbol{y}_{\backslash i}; \boldsymbol{\theta})
$$
$$
- (u_i - \mu_i) p(x_i = u_i, \boldsymbol{y}_{\backslash i}; \boldsymbol{\theta}) + \xi_i p_1(x_i = u_i, \boldsymbol{y}_{\backslash i}; \boldsymbol{\theta}) \Big). \tag{72}
$$

Since (71) and (72) are equivalent, we have

$$
\nabla_{\sigma_i} p(\boldsymbol{y}; \boldsymbol{\theta}) = \frac{1}{2\sigma_i} \Big( (l_i - \mu_i) p(x_i = l_i, \boldsymbol{y}_{\backslash i}; \boldsymbol{\theta}) - \xi_i p_1(x_i = l_i, \boldsymbol{y}_{\backslash i}; \boldsymbol{\theta})
$$
$$
- (u_i - \mu_i) p(x_i = u_i, \boldsymbol{y}_{\backslash i}; \boldsymbol{\theta}) + \xi_i p_1(x_i = u_i, \boldsymbol{y}_{\backslash i}; \boldsymbol{\theta}) - \mathsf{E}_z[z][\boldsymbol{\Sigma} \boldsymbol{H}_1]_{ii} \Big) = \frac{1}{2} \boldsymbol{D}_{1,ii}.
$$

Therefore, $\frac{1}{2}(\boldsymbol{H}_1 + \boldsymbol{D}_1) = \nabla_{\boldsymbol{\Sigma}} p(\boldsymbol{y}; \boldsymbol{\theta})$ is valid. $\qquad\square$

Based on Lemma 11, we have that

$$
p^{-1}(\boldsymbol{y}; \boldsymbol{\theta}) \boldsymbol{\Sigma} (\boldsymbol{H}_1 + \boldsymbol{D}_1) \boldsymbol{\Sigma} = -2 \nabla_{\boldsymbol{\Sigma}^{-1}} \log p(\boldsymbol{y}; \boldsymbol{\theta}).
$$

Hence, the distance (68) becomes

$$
\| \boldsymbol{\Sigma} - \boldsymbol{\Sigma}^\star \|_{\mathsf{F}} = \left\| \underline{\boldsymbol{\Sigma}} - \boldsymbol{\Sigma}^\star - \frac{2}{n} \sum_{t=1}^n \nabla_{\underline{\boldsymbol{\Sigma}}^{-1}} \log p(\boldsymbol{y}; \underline{\boldsymbol{\theta}}) + \frac{2}{n} \sum_{t=1}^n \nabla_{\boldsymbol{\Sigma}^{\star-1}} \log p(\boldsymbol{y}; \boldsymbol{\theta}^\star) \right\|_{\mathsf{F}}
$$
$$
= \left\| \underline{\boldsymbol{\Sigma}} - \boldsymbol{\Sigma}^\star - \frac{2}{n} \sum_{t=1}^n \int_0^1 \nabla^2_{\tilde{\boldsymbol{\Sigma}}^{-1}, \tilde{\boldsymbol{\Sigma}}} \log p(\boldsymbol{y}_t; \tilde{\boldsymbol{\theta}}_{\Sigma})(\tilde{\boldsymbol{\Sigma}} - \boldsymbol{\Sigma}^\star) \mathrm{d}\beta \right\|_{\mathsf{F}} \tag{73}
$$
$$
\leq \sup_{\beta \in (0,1]} \left\| \boldsymbol{I}_{d \times d} - \frac{2}{n} \sum_{t=1}^n \frac{\partial \mathrm{vec}\left( \nabla_{\tilde{\boldsymbol{\Sigma}}^{-1}} \log p(\boldsymbol{y}_t; \tilde{\boldsymbol{\theta}}_{\Sigma}) \right)}{\partial \mathrm{vec}\left( \tilde{\boldsymbol{\Sigma}} \right)} \right\|_2 \| \underline{\boldsymbol{\Sigma}} - \boldsymbol{\Sigma}^\star \|_{\mathsf{F}}.
$$

where the second equation is given by the mean value theorem of integrals and $\tilde{\boldsymbol{\theta}}_{\Sigma} = \left\{ \underline{\boldsymbol{\mu}}, \tilde{\boldsymbol{\Sigma}}, \underline{\boldsymbol{\xi}} \right\}$ with $\tilde{\boldsymbol{\Sigma}} = \beta \underline{\boldsymbol{\Sigma}} + (1 - \beta) \boldsymbol{\Sigma}^\star$ and $\beta \in (0, 1]$.

To bound the term $\sup_{\beta \in (0,1]} \left\| \boldsymbol{I}_{d \times d} - \frac{2}{n} \sum_{t=1}^n \frac{\partial \mathrm{vec}(\nabla_{\tilde{\boldsymbol{\Sigma}}^{-1}} \log p(\boldsymbol{y}_t; \tilde{\boldsymbol{\theta}}_{\Sigma}))}{\partial \mathrm{vec}(\tilde{\boldsymbol{\Sigma}})} \right\|_2$, we need some results for the term $\frac{\partial \mathrm{vec}(\nabla_{\boldsymbol{\Sigma}^{-1}} \log p(\boldsymbol{y}_t; \boldsymbol{\theta}))}{\partial \mathrm{vec}(\boldsymbol{\Sigma})}$ at first.

**Lemma 13.** *The term $\frac{\partial \mathrm{vec}(\nabla_{\boldsymbol{\Sigma}^{-1}} \log p(\boldsymbol{y}; \boldsymbol{\theta}))}{\partial \mathrm{vec}(\boldsymbol{\Sigma})}$ satisfy that $\boldsymbol{I}_{d \times d} - \frac{2}{n} \sum_{t=1}^n \frac{\partial \mathrm{vec}(\nabla_{\boldsymbol{\Sigma}^{-1}} \log p(\boldsymbol{y}_t; \boldsymbol{\theta}))}{\partial \mathrm{vec}(\boldsymbol{\Sigma})}$ is a positive definite matrix.*

*Proof.* Based on

$$
\frac{\partial \mathrm{vec}\left( \nabla_{\boldsymbol{\Sigma}^{-1}} p(\boldsymbol{x}, z; \boldsymbol{\theta}) \right)}{\partial \mathrm{vec}\left( \boldsymbol{\Sigma} \right)} = -\frac{\partial \mathrm{vec}\left( \nabla_{\boldsymbol{\Sigma}^{-1}} p(\boldsymbol{x}, z; \boldsymbol{\theta}) \right)}{\partial \mathrm{vec}\left( \boldsymbol{\Sigma}^{-1} \right)} \left( \boldsymbol{\Sigma}^{-1} \otimes \boldsymbol{\Sigma}^{-1} \right),
$$

we can obtain that $\frac{\partial \mathrm{vec}\left(\nabla_{\boldsymbol{\Sigma}^{-1}} p(\boldsymbol{x},z;\boldsymbol{\theta})\right)}{\partial \mathrm{vec}(\boldsymbol{\Sigma})}$ is positive definite and

$$
\boldsymbol{I}_{d \times d} - \frac{2}{n} \sum_{t=1}^{n} \frac{\partial \mathrm{vec}\left(\nabla_{\boldsymbol{\Sigma}^{-1}} p(\boldsymbol{y}_t; \boldsymbol{\theta})\right)}{\partial \mathrm{vec}\left(\boldsymbol{\Sigma}\right)}
$$

$$
= \frac{1}{2n} \sum_{t=1}^{n} \left(\boldsymbol{\Sigma}^{-1} \otimes \boldsymbol{\Sigma}^{-1}\right) \mathsf{Cov}_{\boldsymbol{x},z|\boldsymbol{y};\boldsymbol{\theta}}\left[\mathrm{vec}\left(z^{-1}(\boldsymbol{x}-\boldsymbol{\mu}-z\boldsymbol{\xi})(\boldsymbol{x}-\boldsymbol{\mu}-z\boldsymbol{\xi})^{\top}\right)\right] \succ \boldsymbol{0}.
$$

$\square$

Based on Theorem 2 and Lemma 13, we have

$$
\sup_{\beta \in (0,1]} \left\| \boldsymbol{I}_{d \times d} - \sum_{t=1}^{n} \frac{\partial \mathrm{vec}\left(\nabla_{\tilde{\boldsymbol{\Sigma}}^{-1}} \log p(\boldsymbol{y}_t; \tilde{\boldsymbol{\theta}}_{\Sigma})\right)}{\partial \mathrm{vec}\left(\tilde{\boldsymbol{\Sigma}}\right)} \right\|_2 = \sup_{\beta \in (0,1]} \left\| \frac{1}{2n} \sum_{t=1}^{n} \left(\tilde{\boldsymbol{\Sigma}}^{-1} \otimes \tilde{\boldsymbol{\Sigma}}^{-1}\right) \mathsf{Cov}_{\boldsymbol{x},z|\boldsymbol{y}_t; \tilde{\boldsymbol{\theta}}_{\Sigma}}\left[\mathrm{vec}\left(\boldsymbol{x}\boldsymbol{x}^{\top}\right)\right] \right\|_2
$$

$$
= \max_{\boldsymbol{\theta}} \left\| \frac{1}{2n} \sum_{t=1}^{n} \left(\boldsymbol{\Sigma}^{-1} \otimes \boldsymbol{\Sigma}^{-1}\right) \mathsf{Cov}_{\boldsymbol{x},z|\boldsymbol{y}_t; \boldsymbol{\theta}}\left(\mathrm{vec}\left(\boldsymbol{x}\boldsymbol{x}^{\top}\right)\right) \right\|_2
$$

$$
\triangleq c_{\Sigma} \in (0,1).
$$

# D. The Proof of Theorem 4

The component-wise convergence rates $c_\mu$, $c_\Sigma$, and $c_\xi$ given in Proposition 3 are the upper bound for all iterates. Since all of these rates are in $(0,1)$, there exists a constant $c_0 > 0$, we have

$$
c_\mu \in (0, 1-c_0], \ c_\xi \in (0, 1-c_0], \ \text{and} \ c_\Sigma \in (0, 1-c_0].
$$

# E. Derivations of Surrogate Functions in Quantized Matrix Completion and Compressive Sensing

## E.1. Surrogate Function Derivation in Quantized Matrix Completion

The goal of the low-rank matrix completion problem is to recover an unknown low-rank matrix $\boldsymbol{M} \in \mathbb{R}^{d_1 \times d_2}$ from an observed, yet incomplete, matrix. Let $\boldsymbol{X}$ denote a matrix whose entries are drawn from a normal variance-mean mixture distribution. We use $\mu_{ij}$ and $x_{ij}$ to represent the $(i,j)$-th entries of $\boldsymbol{M}$ and $\boldsymbol{X}$, respectively. Based on the normal variance-mean model, the relationship between $\mu_{ij}$ and $x_{ij}$ is given by

$$
x_{ij} = \mu_{ij} + z^{\frac{1}{2}} \sigma \epsilon,
$$

where $\mu_{ij}$ serves as the location parameter of $x_{ij}$ and both $\xi$ and $\sigma$ are given constants. In the quantization scenario, we have $\boldsymbol{Y} = \mathcal{Q}(\boldsymbol{X})$. The entire matrix $\boldsymbol{Y}$ is not available for observation. Let $\mathcal{O}$ denote the index set of the observed entries. Our goal is to recover $\boldsymbol{M}$ from incomplete $\boldsymbol{Y}$, which is equivalent to estimating all the $\mu_{ij}$ in the matrix $\boldsymbol{M}$.

Since $x_{ij}$ follows a univariate normal variance-mean model and $\boldsymbol{Y} = \mathcal{Q}(\boldsymbol{X})$, the density function of $y_{ij}$ is identical with (5) under the univariate case. Hence, the negative log-likelihood function for $y_{ij}$ with $ij \in \mathcal{O}$ is

$$
\sum_{(i,j) \in \mathcal{O}} \log p\left(y_{ij} \mid \boldsymbol{M}\right),
$$

which is the objective function of (18). Applying Jensen's inequality as in (12), we have the surrogate function

$$
S(\boldsymbol{M}; \underline{\boldsymbol{M}}) = \sum_{(i,j) \in \mathcal{O}} \Bigg[ \mathsf{E}_{z_t|\boldsymbol{y}_t;\underline{\boldsymbol{\theta}}}\left[\log p(z_t)\right] - \frac{1}{2} \log \det \sigma^2 - \frac{1}{2\sigma^2}\big(u_{ij} - 2v_{ij}(\mu_{ij} - \mathsf{E}(z)\xi)
$$

$$
+ \iota_{ij}(\mu_{ij} - \mathsf{E}(z)\xi)^2 - 2(w_{ij} - (\mu_{ij} - \mathsf{E}(z)\xi))\xi + \zeta_{ij}\xi^2\big) \Bigg] + \text{const.},
$$

where $u_{ij}$, $v_{ij}$, and $w_{ij}$ are the univariate versions of $\boldsymbol{U}_{ij}$, $\boldsymbol{v}_{ij}$, and $\boldsymbol{w}_{ij}$, respectively. Ignoring the terms which are independent with $\mu_{ij}$, the surrogate function becomes

$$S(\boldsymbol{M}; \underline{\boldsymbol{M}}) = \frac{1}{2\sigma^2} \sum_{(i,j)\in\mathcal{O}} \left(2v_{ij}\mu_{ij} - \iota_{ij}\mu_{ij}^2 - 2\mu_{ij}\xi\right) + \text{const.}$$

Making a low-rank factorization to the matrix $\boldsymbol{M}$ as $\boldsymbol{M} = \boldsymbol{A}\boldsymbol{B}^\top$, we have $\mu_{ij} = \boldsymbol{a}_i\boldsymbol{b}_j^\top$. Setting $e_{ij} = \frac{v_{ij}-\xi}{\iota_{ij}}$, we have that maximizing $S(\boldsymbol{M}; \underline{\boldsymbol{M}})$ is equivalent to maximize

$$\sum_{(i,j)\in\mathcal{O}} \left(\boldsymbol{a}_i\boldsymbol{b}_j^\top - e_{ij}\right)^2,$$

which is identical with (19).

### E.2. Surrogate Function Derivation in Quantized Compressive Sensing

In quantized compressive sensing, the base model with normal variance-mean mixture noise is given by

$$\boldsymbol{y} = \mathcal{Q}(\boldsymbol{x}), \quad \boldsymbol{x} = \boldsymbol{\Phi}\boldsymbol{\vartheta} + z\boldsymbol{\xi} + z^{\frac{1}{2}}\sigma\boldsymbol{\epsilon}.$$

In the above model, the term $\boldsymbol{\Phi}\boldsymbol{\vartheta}$ can be regarded as the location parameter of $\boldsymbol{x}$. We can use the ML estimation method to recover the sparse signal $\boldsymbol{\vartheta}$, which has the optimization problem

$$\max_{\boldsymbol{\vartheta}} \quad \log p\left(\boldsymbol{y} \mid \boldsymbol{\vartheta}\right).$$

Following the suggestions from (Zymnis et al., 2009), we add a $\ell_1$-regularization term to force the solution of $\boldsymbol{\vartheta}$ to be sparse. Hence, we obtain the optimization problem

$$\max_{\boldsymbol{\vartheta}} \quad \log p\left(\boldsymbol{y} \mid \boldsymbol{\vartheta}\right) + \eta\|\boldsymbol{\vartheta}\|_1.$$

To solve the above optimization problem through ECM algorithm, we can apply the E-step as in (12) to obtain the surrogate function

$$\begin{aligned}
S(\boldsymbol{\vartheta}; \underline{\boldsymbol{\vartheta}}) = {}& \mathsf{E}_{z_t|\boldsymbol{y}_t;\underline{\boldsymbol{\theta}}}\left[\log p(z_t)\right] - \frac{1}{2}\log\det\sigma^2 \\
& - \frac{1}{2\sigma^2}\left(\left(\boldsymbol{u} - 2\boldsymbol{v}^\top\boldsymbol{A}\boldsymbol{\vartheta} + \iota\boldsymbol{\vartheta}^\top\boldsymbol{A}^\top\boldsymbol{A}\boldsymbol{\vartheta}\right) - 2(\boldsymbol{w} - \boldsymbol{A}\boldsymbol{\vartheta})^\top\boldsymbol{\xi} + \zeta\boldsymbol{\xi}^\top\boldsymbol{\xi}\right) + \eta\|\boldsymbol{\vartheta}\|_1 + \text{const.}
\end{aligned}$$

Ignoring the terms which are independent with $\boldsymbol{\vartheta}$, the surrogate function becomes

$$S(\boldsymbol{\vartheta}; \underline{\boldsymbol{\vartheta}}) = \frac{1}{2\sigma^2}\left(2\boldsymbol{v}^\top\boldsymbol{A}\boldsymbol{\vartheta} - \iota\boldsymbol{\vartheta}^\top\boldsymbol{A}^\top\boldsymbol{A}\boldsymbol{\vartheta} - 2\boldsymbol{\vartheta}^\top\boldsymbol{A}^\top\boldsymbol{\xi}\right) + \eta\|\boldsymbol{\vartheta}\|_1 + \text{const.}$$

Setting $\boldsymbol{e} = \frac{\boldsymbol{v}-\boldsymbol{\xi}}{\iota}$, we have that maximizing $S(\boldsymbol{\vartheta}; \underline{\boldsymbol{\vartheta}})$ is equivalent to maximize

$$\|\boldsymbol{A}\boldsymbol{\vartheta} - \boldsymbol{e}\|_2^2 + \eta\|\boldsymbol{\vartheta}\|_1,$$

which is identical with the surrogate function in quantized compressive sensing.

## F. Experiment Details

### F.1. Benchmark Settings

**Quantized matrix completion:** For all algorithms presented in Table 1, the complete matrix $\boldsymbol{M}$ is reconstructed from the training data. Denote by $\mathcal{O}_{\text{test}}$ the set of indices corresponding to entries in the test data. The definitions of the two benchmarks are given as follows.

1. Accuracy: $\frac{1}{|\mathcal{O}_{\text{test}}|} \sum_{(i,j) \in \mathcal{O}_{\text{test}}} I(Q(\mu_{ij}) = y_{ij})$.;

2. RMSE: $\sqrt{\frac{1}{n_{\text{test}}} \sum_{(i,j) \in \mathcal{O}_{\text{test}}} (\mu_{ij} - y_{ij})^2}$,

where $I(\cdot)$ is the indicator function.

**Quantized compressive sensing:** The signal-to-noise ratio (SNR) is defined as the ratio of the expectation of the original measurements to the variance of the noise term, which is

$$\text{SNR} = \frac{\mathsf{E}[\boldsymbol{A}\boldsymbol{\vartheta}]}{\text{Var}[\boldsymbol{x} - \boldsymbol{A}\boldsymbol{\vartheta}]}.$$

Denote the ground truth sparse signal as $\boldsymbol{\vartheta}_{\text{gd}}$, the estimated sparse signal as $\boldsymbol{\vartheta}_{\text{est}}$. The cosine similarity is defined as

$$\text{Cos Sim} = \frac{\boldsymbol{\vartheta}_{\text{gd}}^{\top} \boldsymbol{\vartheta}_{\text{est}}}{\|\boldsymbol{\vartheta}_{\text{gd}}\|_2 \|\boldsymbol{\vartheta}_{\text{est}}\|_2}.$$

