# OpenReview forum: "Inference from Quantized Data via Normal Variance-Mean Mixtures"
_ICML.cc/2026/Conference — ICML 2026 regular_

### Official Review · Reviewer_LpRB · 2026-03-04

**Soundness:** 4
**Presentation:** 4
**Significance:** 2
**Originality:** 2
**Overall Recommendation:** 3
**Confidence:** 4

**Summary:**

The paper considers the problem of estimation from scalar quantization of the measurement signal. The main contribution of the paper is generalization of previous works in terms of the statistical model and the number of quantization bits in each dimension. The paper proposes an expectation constrained maximization (ECM) type algorithm for this problem, establishes its convergence to a stationary point (with convergence rates, though in somewhat implicit form), and experiment with the algorithm in various settings.

**Compliance With Llm Reviewing Policy:**

Affirmed.

**Key Questions For Authors:**

1) What are the differences of the ECM algorithm developed here from the ECM algorithm of (Papadopoulos et al., 2001), (Finesso et al., 1999) and in the book (McLachlan and Krishnan 2008)? Comment on challenges and novelty.

2) Quantized Probabilistic Matrix Completion - How do you assure in your model that $X$ has low rank?

3) The skewness is one-dimensional, in the direction of $\xi$. Can the algorithm be easily generalized to multiple directions?

**Limitations:**

Yes

**Strengths And Weaknesses:**

## Strengths ##

1) The paper is technically sound and the presentation is very good.

2) The proposed algorithm is elegant, and rather general. The demonstrations in the experiments are interesting.

3) The proof of the concavity of the likelihood in terms of the parameters in the appendix is non-trivial (and requires sophisticated tools such as the Brascamp–Lieb inequality)

## Weaknesses ##

1) The paper cites a lot of reference but the problem of statistical estimation under communication constraints (i.e., quantization) has been studied thoroughly in statistics, machine learning and information theory. Some representative examples are:

* Optimality guarantees for distributed statistical estimation, Duchi, Jordan, Wainwright 2014

* Generalized linear models with 1-bit measurements: Asymptotics of the maximum likelihood estimator, Shah et al 2025

* Communication-efficient sparse regression, Lee et al, 2017

* A geometric characterization of fisher information from quantized samples with applications to distributed statistical estimation, Barnes et al 2018

* Gaussian approximation of quantization error for estimation from compressed data, Kipnis and Reeves 2021

* Mean estimation from one-bit measurements, Kipnis and Duchi 2022

but there are many more (e.g., papers that cite these papers and references therein).

2) The developed ECM algorithm is fairly standard, and given the model and the identification of latent variables, the derivations are pretty straightforward (even if there are some technical issues here and there). The convergence analysis also follow directly from known results (again, modulo some technical challenges for showing concavity mentioned above). So overall, the originality is not high.

3) Moreover, starting with the seminal paper “Statistical guarantees for the EM algorithm” by Balakrishnan et al (2013), much stronger convergence guarantees have been developed for the EM algorithm, such as global convergence, and explicit dependence of convergence rates and estimation errors on the problem parameters (as well as minimax optimality). While this is not expected in the general setting the paper considers here, perhaps this can be considered in a simple case of the general setting,

4) The quantizer is arbitrary and fixed, and there is no clue on how to optimize it.

---

> ### Author Rebuttal · Authors · 2026-03-31
>
> **Response to W1:** We sincerely thank the reviewer for highlighting these foundational papers. We agree they provide excellent theoretical background, establishing fundamental limits (e.g., MSE bounds, Fisher information, and asymptotics) for statistical estimation under communication constraints.
>
> However, our paper has a different, highly specific focus. While the suggested literature primarily addresses general theoretical benchmarks or 1-bit measurements, our work specifically tackles data following a normal mean-variance mixture distribution. Our main contribution lies in formulating the location parameter estimation for this flexible distribution from quantized data, and, crucially, developing a practical and computationally efficient algorithm to solve it. Therefore, our algorithmic approach is complementary to these theoretical works. In the final version, we will cite these references in motivating the problem background and add a discussion to better position our specific algorithmic contributions within this broader theoretical context.
>
> **Response to W2 and Q1:** The work (Papadopoulos et al., 2001) applied ECM to a univariate multi-bits case under Gaussian distribution, and the work Finesso (1999) applied ECM to a univariate 1-bit case under Gaussian distribution. Our work extend the univariate to the multivariate and extend the Gaussian distributional assumption to the general normal mean-variance mixture. The book by McLachlan and Krishnan~(2008) provides the fundamental framework for EM-type algorithms. To extend this framework to the entire class of normal mean-variance mixture models, a two-layer nested latent-variable structure is introduced, in which $\\boldsymbol{x}$ is treated as a latent variable of $\\boldsymbol{y}$, and $z$ is further treated as a latent variable of $x$. This structure yields a tractable surrogate function by taking expectations twice. Meanwhile, although the EM algorithm itself has a fixed structural form, its expectation step requires problem-specific analysis. In particular, since this paper involves taking two expectations over quantized variables, the derivation of closed-form expressions for these expectations is likewise non-trivial.
>
> **Response to W3:** We thank the reviewer for this helpful suggestion. We agree that stronger EM guarantees of the type developed in Balakrishnan et al. (2017) would be very interesting in our setting. We will clarify this limitation and highlight it as an important direction for future work.
>
> **Response to W4:** We thank the reviewer for this insightful comment. We agree that optimizing the quantizer is theoretically significant, as it minimizes information loss and yields better estimation bounds.
>
> However, our paper specifically targets real-world scenarios where the quantizer is externally imposed and strictly unalterable. An "arbitrary and fixed" quantizer is not a limitation of our work, but precisely our foundational problem setting. For instance, in legacy hardware systems or human-designed scoring mechanisms (e.g., 1-to-5 star ratings or grading scales), quantization rules are predefined and fundamentally cannot be modified. Since altering the quantizer is practically impossible in these contexts, our algorithm actively focuses on estimating model given any fixed rule.
>
> **Response to Q2:** We thank the reviewer for this question. We assure the low-rank property of the matrix $\\boldsymbol{X}$ to be estimated (denoted as $\\boldsymbol{M}$ in our paper) by explicitly employing low-rank factorization. Specifically, instead of optimizing the full matrix directly, we parameterize the continuous true score matrix as the product of two lower-dimensional latent matrices: $\\boldsymbol{M} = \\boldsymbol{A} \\boldsymbol{B}^\\top$, where $\\boldsymbol{A} \\in \\mathbb{R}^{d_ {1} \\times r}$ and $\\boldsymbol{B} \\in \\mathbb{R}^{d_ {2} \\times r}$. The rank $r$ is a predefined hyperparameter chosen to be strictly much smaller than the matrix dimensions ($r \\ll \\min(n, m)$).
>
> **Response to Q3:** Thank you for this insightful suggestion. Theoretically, yes. Multiple-dimensional skewness can be introduced by extending $z$ to a vector. In this case, the definition of the random variable $\\boldsymbol{x}$ becomes
>
> $$
> \\boldsymbol{x} = \\boldsymbol{\\mu} + \\sum_ i z_ i \\boldsymbol{\\xi}_ i + \\left( \\sum_ i z_ i \\boldsymbol{\\varSigma} \\right)^{0.5}\\boldsymbol{\\epsilon}.
> $$
>
> From a derivation perspective, it is possible to construct a surrogate objective for the multi-dimensional $z$ case by applying Jensen's inequality twice. However, the main difficulty arises in the actual optimization, because the expectation with respect to $z \\mid \\boldsymbol{y}$ no longer admits a closed-form solution as in this paper and instead requires numerical computation. This issue will be considered as a direction for future work.

---

> > ### Author Rebuttal · Reviewer_LpRB · 2026-04-02
> >
> > Re response to W2 and Q1: My personal impression is that the novelty is mild.
> >
> > Re response to W4: I agree that there are various practical situations in which the quantizer is imposed, and in these cases, your proposed method is directly applicable.  Stronger results will also allow, beyond that, some hint as how to optimize this quantizer (or, say, a principled way to choose the better one from a few possible legacy methods).
> >
> > Re response to Q3: Thank you for the explanation. So I found this, unfortunately, another limitation, though I agree that a minor one.

---

> > > ### Author Response · Authors · 2026-04-06
> > >
> > > **On the Novelty of Our Work.**
> > >
> > > We thank the reviewer for the careful reading and follow-up comments. We would like to clarify that the main contribution of this paper lies **not in the ECM template itself**, but in the **modeling idea** and the nontrivial use of ECM in a much broader setting. The distinction from prior ECM-based works is summarized below:
> > >
> > > | **Aspect** | **Finesso (1999)** | **Papadopoulos et al. (2001)** | **Our work** |
> > > |---|---|---|---|
> > > | Dimension setting | Univariate | Univariate | **Multivariate** |
> > > | Bits | One-bit | Multi-bits | **Multi-bits** |
> > > | Distribution family | Gaussian | Gaussian | **Normal mean-variance mixtures** |
> > > | Convergence guarantee | Not provided | Not provided | **Provided** |
> > > | ECM structure | Standard ECM | Standard ECM | **Two-layer latent variables** |
> > >
> > > Our adopted **normal mean-variance mixture family** is among the most general and widely used classes for asymmetric and heavy-tailed data, including the Gaussian, Student's $t$, generalized hyperbolic, Variance-Gamma, and Normal Inverse Gaussian distributions as special cases. Our method relies on a **two-layer latent-variable structure**, where $\boldsymbol{x}$ is treated as a latent variable of $\boldsymbol{y}$, and $z$ is treated as a latent variable of $\boldsymbol{x}$, resulting in a tractable surrogate objective through a **double-expectation procedure**. Deriving the closed-form solutions under this framework over multiple integrals (Appendix A) constitutes a major technical contribution, which is significantly more challenging than standard Gaussian ECM and is **not a routine specialization** of standard ECM as in McLachlan and Krishnan (2008).
> > >
> > > **On Quantizer Optimization.**
> > >
> > > We agree that **quantizer design** is an important problem in its own right. In
> > > general, quantizer design accesses to the raw data or to the probability model, then seeks to reduce distortion. However, our paper studies the setting where the quantizer is **fixed and known**, and the goal is to recover the latent distribution parameters from **already quantized observations**. In this setting, jointly optimizing the distribution and the quantizer from quantized data alone is fundamentally hindered by a **lack of identifiability**.
> > >
> > > Specifically, the **scale parameters** of the latent distribution and the **quantizer thresholds** are tightly coupled: scaling both by the same factor leaves the likelihood of the discrete observations unchanged. For example, in Scenario A, let $X_A \sim \mathcal{N}(0,1)$ with a uniform quantizer with thresholds $[-2,-1,0,1,2]$. In Scenario B, let $X_B \sim \mathcal{N}(0,100)$ and scale the quantizer accordingly, with the thresholds $[-20,-10,0,10,20]$. These two scenarios induce **exactly the same distribution over quantized outputs**, although both the latent scale and quantizer differ. Hence, without additional constraints or side information, joint estimation of the latent distribution and the quantizer is in general **not identifiable** from quantized observations alone.
> > >
> > > The reviewer's suggestion is constructive. Under additional assumptions, such a joint optimization problem may become well-defined. However, that would be a **different problem formulation** from the one studied here and is beyond the scope of this work.
> > >
> > > **On Multi-Dimensional Skewness.**
> > >
> > > We thank the reviewer for this comment and would like to clarify our previous response. The current paper focuses on the **one-dimensional skewness setting** within the normal mean-variance mixture family, which is already one of the broadest and most widely used frameworks for asymmetric and heavy-tailed data. It covers a rich class of distributions, including the Gaussian and many important non-Gaussian families.
> > > More general multi-directional skewness can, in principle, be introduced by extending the latent mixing variable from a scalar to a vector, as suggested by the reviewer. In that setting, the same overall ECM framework would still apply in principle; that is, the framework is **not fundamentally restricted** to scalar skewness.
> > > What changes is that the required conditional expectations would generally no longer admit the particularly tractable treatment used here and would instead require additional numerical integration. We do not view this as a fundamental limitation. Numerical evaluation in the E-step is standard in EM/ECM-type methods beyond a few especially tractable cases, such as the Student's $t$. In this sense, numerical integration is better viewed as the **common case for more general models**, rather than a substantive weakness of the approach.
> > >
> > > We again thank the reviewer for the careful reading and positive assessment of the technical soundness and presentation of the paper. We hope the above clarification makes the scope and contribution of the paper more precise, and if it has adequately addressed the concern, we would appreciate it being taken into account in the final evaluation.

---

### Official Review · Reviewer_QzPT · 2026-03-11

**Soundness:** 3
**Presentation:** 3
**Significance:** 3
**Originality:** 3
**Overall Recommendation:** 4
**Confidence:** 4

**Summary:**

The paper proposes a method for estimating statistical parameters of generating models from quantized data. They propose the expectation-conditional-maximization (ECM) algorithm. This is a two-step iterative algorithm; the first step is the so-called expectation step (E-step), where a surrogate to the maximum likelihood function is derived, parametrizing the interest variables x and z; the second step is called the conditional-maximization step (CM-step), where the surrogate function is maximized, which inherits an identifiability issue solved by iterating over the parameter estimations. The method is tested in four scenarios:
(i) a quantized linear regression,
(ii) a recommendation problem,
(iii) a compressive sensing problem, and
(iv) a quantized correlation estimation problem.
The results show that the proposed method surpasses the others in all scenarios, with the cost of additional processing time for a few of the competing methods.

**Compliance With Llm Reviewing Policy:**

Affirmed.

**Key Questions For Authors:**

1. What are the parameters in Equation (7)? Some new parameters are introduced, and their meaning should be clearer.
2. In subsection 3.3, what is the size of X, and are there limitations when compared to the size of M?
3. Many wireless communication scenarios use complex-valued signals. Is it possible at least to mention how to extend the proposed solution to a complex-valued problem?
4. The results discussion lacks a more detailed evaluation of the proposed method when compared to the other methods. What is the impact of the computational time on the method decision, and the computational complexity?

**Limitations:**

yes

**Strengths And Weaknesses:**

The paper is solid and well-written for the most part. It lacks a thorough revision for repeating phrases, such as “In many applications, the primary interest is not the signal itself, but its underlying statistical properties...” which appears twice in the third paragraph of the Introduction. The paper has a good presentation of the state-of-the-art, with not only classical references but also includes some
new references. The significance of the contribution is clearly stated; however, it focuses only on the strength of the methods, without a clear tradeoff when compared to other methods. The paper has originality, but it does not provide new insights or further understanding of the problem.

---

> ### Author Rebuttal · Authors · 2026-03-31
>
> **Response to Q1:** For the parameter $\\boldsymbol{\\beta}$ in Equation (7), this was a typographical error. The parameter $\\boldsymbol{\\beta}$ should strictly be $\\boldsymbol{\\vartheta}$. We thank the reviewer for catching this typo. For the other parameters $z$ and $\\xi$, they have been introduced in Section 2.
>
> **Response to Q2:** We sincerely thank the reviewer for pointing this out. For the size of $\\boldsymbol{X}$ and its limitations compared to $\\boldsymbol{M}$, there are no size limitations between them, as $\\boldsymbol{X}$ is simply $\\boldsymbol{M}$ plus entry-wise noise. The only limitation is the number of observed entries ($|\\boldsymbol{\\Omega}| \\ll d_ 1 \\times d_ 2$), rather than the matrix dimensions. We have updated subsection 3.3 to explicitly state "Let $\\boldsymbol{X} \\in \\mathbb{R}^{d_ 1 \\times d_ 2}$".
>
> **Response to Q3:** We thank the reviewer for this insightful suggestion. Yes, our proposed solution can be naturally extended to complex-valued scenarios. In the complex-valued quantization literature for wireless communication scenarios, such as [A] and [B], the real and imaginary parts are quantized separately. In this case, parameter estimation for a $d$-dimensional complex-valued variable can be equivalently regarded as parameter estimation for a $2d$-dimensional real-valued variable; therefore, the proposed method is also applicable to parameter estimation for complex-valued variables. To address this point, a brief discussion on this extension pathway for wireless communication applications will be added.
>
> [A] Approximate message passing with parameter estimation for heavily quantized measurements, Huang et al., 2022.
>
> [B] Channel estimation in broadband millimeter wave MIMO systems with few-bit ADCs, Mo et al., 2017.
>
> **Response to Weakness and Q4:** Thank you for the constructive feedback. The main trade-off of our method is that it achieves higher accuracy at the cost of time.
>
> Regarding the scalability, since $\\boldsymbol{y}$ is constrained to discrete values, parameter updates require statistics for only $n_ 0 = \\min\\{(e+1)^d, n\\}$ distinct samples. The per-iteration complexity is dominated by the numerical integration (using $K$-node Gauss-Legendre quadrature) and the multivariate normal CDF evaluation ($\\mathcal{O}(d^3)$).
>
> For the parameters $\\boldsymbol{\\mu}$ and $\\boldsymbol{\\xi}$, the update complexity is $\\mathcal{O}(n_ 0 K d^3)$. For $\\boldsymbol{\\varSigma}$, a naive computation of the Hessian $\\boldsymbol{H}_ 1 + \\boldsymbol{D}_ 1$ leads to an expensive $\\mathcal{O}(n_ 0 K d^5)$; however, by applying the numerical differentiation method provided in Lemma 10 (Appendix B.3), this is reduced to $\\mathcal{O}(n_ 0 d^3)$. Consequently, the total per-iteration complexity is $\\mathcal{O}(n_ 0 K d^3)$, which scales linearly with $n_ 0$. In scenarios with coarse quantization or low dimensionality, $n_ 0$ becomes a small constant independent of $n$, ensuring the algorithm remains highly efficient for large datasets.
>
> The choice of method depends on the application scenario. When the observed data exhibit pronounced asymmetry, skewed distributions are adopted for modeling; in particular, the GH distribution can be used when a more refined characterization of tail behavior is required, whereas the skew-$t$ distribution is preferred otherwise. By contrast, when the asymmetry of the observed data is not significant, Gaussian and Student's $t$ distributions can be used for modeling. In this case, the skewness parameter $\\boldsymbol{\\xi}$ does not need to be estimated, thereby reducing the computational complexity.
>
> We thank the reviewer again for the valuable comments. We hope our responses have addressed your concerns. If there are any further questions, we would be happy to provide additional clarification.

---

> > ### Author Rebuttal · Reviewer_QzPT · 2026-04-02
> >
> > The authors have clearly answer to my questions. I recommend to compare the proposed method to other methods if possible.

---

> > > ### Author Response · Authors · 2026-04-07
> > >
> > > Thank you again for the **positive assessment** of our paper and for the careful review. We also appreciate your suggestion to further compare the proposed method with other methods in terms of **computational efficiency** and **runtime**. In response, we carried out additional and more detailed comparisons in the experimental scenarios considered in the paper.
> > >
> > > For the **quantized matrix completion** experiment, the results are reported in **Table 1**. As the model becomes more expressive, from the Gaussian distribution to the Student's $t$ distribution and then to the symmetric generalized hyperbolic (SGH) distribution, the running time increases only moderately, while the estimation accuracy improves consistently. This reflects a **favorable tradeoff**: the richer statistical model provides a more accurate description of the data, and the additional computational cost remains well controlled. In particular, compared with non-maximum-likelihood baselines, the proposed method achieves **clearly better performance** with comparable, and in some cases even lower, running time.
> > >
> > > For the **quantized compressive sensing** experiment, the results are reported in **Table 2**. The proposed method achieves the **best overall performance**. Its running time is of the same order as the existing methods, while the gain in estimation accuracy is substantial. In particular, under high noise levels ($\mathrm{SNR} = -10 \mathrm{dB}$), the SGH-based model, which provides the strongest robustness, yields **significantly better performance** than the competing methods. This suggests that the richer modeling framework brings a **clear practical benefit** without introducing a prohibitive computational burden.
> > >
> > > Overall, these additional comparisons show that the proposed method remains **computationally competitive** while providing **more accurate** and **more robust** estimation. More importantly, the richer modeling framework itself is **new** in the quantized maximum-likelihood setting studied here, and the improved empirical performance is precisely the **practical value** brought by this additional flexibility.
> > >
> > > We sincerely thank the reviewer again for the positive evaluation and the constructive suggestion. We hope that this comparison helps further clarify the **strengths** and **practical value** of the proposed method. If the reviewer feels that it adequately addresses the remaining concern, we would be grateful if this could be taken into account in the final evaluation.
> > >
> > > **Table 1. Accuracy and RMSE comparisons of matrix completion on MovieLens 1M**
> > >
> > > | Method | Accuracy | RMSE | Time (s) |
> > > |---|---:|---:|---:|
> > > | SVD | 0.4261 | 0.9148 | 364.36 |
> > > | L2-regularization | 0.4388 | 0.9355 | 165.08 |
> > > | Nuclear Norm | 0.3802 | 1.1662 | 867.39 |
> > > | Gaussian | 0.4363 | 0.9486 | 25.26 |
> > > | 1-bit Gaussian | 0.4217 | 0.9659 | 110.44 |
> > > | Multi-bit Gaussian | 0.4453 | 0.9151 | 90.84 |
> > > | Multi-bit $t$ (prop.) | 0.4464 | 0.9091 | 232.16 |
> > > | Multi-bit SGH (prop.) | **0.4503** | **0.8908** | 272.18 |
> > >
> > > **Table 2. Cosine similarity comparisons of 1-bit compressive sensing under different SNR levels**
> > >
> > > | Method | SNR = 0 dB Cos Sim | SNR = 0 dB Time (s) | SNR = -5 dB Cos Sim | SNR = -5 dB Time (s) | SNR = -10 dB Cos Sim | SNR = -10 dB Time (s) |
> > > |---|---:|---:|---:|---:|---:|---:|
> > > | RSS | 0.7872 | 0.2639 | 0.6598 | 0.2651 | 0.3027 | 0.2546 |
> > > | AOP | 0.7052 | 0.2689 | 0.4041 | 0.2951 | 0.2209 | 0.2738 |
> > > | 1-bit Gaussian | 0.8220 | 2.7704 | 0.5487 | 3.0875 | 0.2896 | 2.7202 |
> > > | 1-bit Student's $t$ ECM (prop.) | 0.8396 | 2.8573 | 0.5678 | 3.1643 | 0.3031 | 3.3174 |
> > > | 1-bit SGH ECM (prop.) | **0.8906** | 2.7252 | **0.7705** | 3.2215 | **0.5386** | 3.5325 |

---

### Official Review · Reviewer_WZcp · 2026-03-11

**Soundness:** 3
**Presentation:** 4
**Significance:** 3
**Originality:** 3
**Overall Recommendation:** 5
**Confidence:** 4

**Summary:**

This works considers max-likelihood (ML) estimation from quantized samples of a mean-variance Gaussian mixture model. The work elaborate the dimensionality challenge in the direct approach and then develops an iterative scheme based on Expectation Maximization to address this computationally hard task. Linear convergence has been shown. The experiments consider various applications including linear regression and quantized compressive sensing.

**Compliance With Llm Reviewing Policy:**

Affirmed.

**Final Justification:**

Following the initial review and the discussion with the authors, I would suggest acceptance. The paper has provided sufficient contribution to a well-imposed problem. While the scope of novelty is limited, the contribution is still appreciated.

**Key Questions For Authors:**

- Is there any reason that the model is restricted to a single-dimensional latent z? Can't we readily extend the algorithm (and maybe analysis) to the multi-dimensional latent, I mean
$$
x = \mu + \sum_i z_i \xi_i + (\sum_i z_i \Sigma_i)^{0.5} \epsilon
$$
Wouldn't that be more general? To what extent is that hard to extend the result? And if not very hard, why not present the results in this case?

- Could you please clarify the notation in the paper? My understanding is that bold lower case is 1D array (vector), bold upper is 2D array (Matrix), lower case normal is scalar, etc. Maybe refer it at the beginning. Anyways, I'm not sure of such distinctions are necessary in general.

- The quantization is performed entry-wise. Well, that is a valid case, but what is more advanced approaches, e.g. vector quantization, are deployed? I guess the problem finds the same form. However, since the dimensions do not easily decouple, it seems hard to use EM, unless a proxy decoupled function can be defined. Any clarification in this respect is appreciated.

**Limitations:**

- Model is rather basic. Extension to more realistic models has not been discussed.

**Strengths And Weaknesses:**

Strengths:
- Problem is well formulated.
- Writing is clear and the main technical contributions are well illustrated.
- Linear convergence has been shown.
- Various applications have been considered.

Weaknesses: Generally, there is no much I could claim regarding the work, as it's well presented. There are though a single major point and few minor points. The major point is that
- The work's novelty is limited as it has adapted a known approach, namely the EM method, to the underlying problem.
Some minor points are also as follows:
- The work has provided limited motivation to the contributions. This is though related to the nature of the studied topic.
- Literature review is rather too long. Some less related lines of work can be less discussed.
- Computational complexity of the approach has not been discussed.

---

> ### Author Rebuttal · Authors · 2026-03-31
>
> **Response to W1:** We agree with the reviewer that EM is a well-established framework; however, we emphasize that our derivation over the twice expectation and the expectation computations introduces substantial new technical insights. The classical EM estimation method constructs a surrogate function by taking the expectation with respect to a random variable and its corresponding latent variable. In contrast, the present paper introduces a two-level nested latent-variable structure, where $\\boldsymbol{x}$ is a latent variable of $\\boldsymbol{y}$, and $z$ is in turn a latent variable of $x$, thereby yielding a tractable surrogate function through two successive expectations. Moreover, although the EM algorithm itself has a fixed structural framework, the expectation step requires problem-specific analysis. In particular, the present study involves taking two expectations over quantized variables, and thus the closed-form derivation of these expectations is likewise non-trivial.
>
> **Response to W3:** We thank the reviewer for this comment. We included a broader literature review is to clarify the relationship between our work and some specific applications. To further improve readability, we will shorten the discussion in the final version.
>
> **Response to W4:** Regarding the scalability, since $\\boldsymbol{y}$ is constrained to discrete values, parameter updates require statistics for only $n_ 0 = \\min\\{(e+1)^d, n\\}$ distinct samples. The per-iteration complexity is dominated by the numerical integration (using $K$-node Gauss-Legendre quadrature) and the multivariate normal CDF evaluation ($\\mathcal{O}(d^3)$).
>
> For the parameters $\\boldsymbol{\\mu}$ and $\\boldsymbol{\\xi}$, the update complexity is $\\mathcal{O}(n_ 0 K d^3)$. For $\\boldsymbol{\\varSigma}$, a naive computation of the Hessian $\\boldsymbol{H}_ 1 + \\boldsymbol{D}_ 1$ leads to an expensive $\\mathcal{O}(n_ 0 K d^5)$; however, by applying the numerical differentiation method provided in Lemma 10 (Appendix B.3), this is reduced to $\\mathcal{O}(n_ 0 d^3)$. Consequently, the total per-iteration complexity is $\\mathcal{O}(n_ 0 K d^3)$, which scales linearly with $n_ 0$. In scenarios with coarse quantization or low dimensionality, $n_ 0$ becomes a small constant independent of $n$, ensuring the algorithm remains highly efficient for large datasets.
>
> A more detailed step-by-step complexity analysis will be added to the final version.
>
> **Response to Q1:** We sincerely thank the reviewer for pointing out this direction for extension. When the latent variable $z$ is scalar, the proposed model naturally reduces to a normal mean--variance mixture model. This is a classical model in the literature that is sufficiently general and fundamentally important. Establishing rigorous algorithmic and theoretical properties for this class of models is itself of substantial value, which is also the reason why this paper focuses on systematically addressing the one-dimensional case.
>
> When $z$ becomes a vector, a more general model can be obtained, which is capable of accommodating different directions of skewness and scatter. From a derivational perspective, a surrogate objective function for the case of multivariate $z$ can indeed be constructed by applying Jensen's inequality twice. However, the explicit computation of expectations with respect to the posterior distribution of $z$ conditional on the observed data $\\boldsymbol{y}$ is non-trivial.
>
> **Response to Q2:** We confirm that your understanding of our notation is exactly correct: bold lowercase denotes 1D vectors, bold uppercase denotes 2D matrices, and standard lowercase denotes scalars. We agree that setting this out explicitly will improve readability. In the final version, we will add an overall introduction to notations.
>
> **Response to Q3:** We thank the reviewer for this highly insightful perspective. We completely agree with your intuition: extending our method to vector quantization introduces dimensional coupling. Similar to extending $z$ to the multivariate case, the use of vector quantization also allows the surrogate function to be derived by applying Jensen's inequality twice. However, in the actual optimization, it is necessary to compute the expectation with respect to $\\boldsymbol{x}\\mid \\boldsymbol{y}$. Specifically, this requires evaluating the integral of $\\boldsymbol{x}$ over a particular hyperrectangle determined jointly by the element-wise quantization function and $\\boldsymbol{y}$. Under vector quantization, however, the conditional region for $\\boldsymbol{x}$ is no longer a coordinate-wise hyperrectangle but a generally polyhedral region, so the corresponding expectation is much more difficult to evaluate.
>
> We thank the reviewer again for the valuable comments. We hope our responses have addressed your concerns. If there are any further questions, we would be happy to provide additional clarification.

---

> > ### Author Rebuttal · Reviewer_WZcp · 2026-04-03
> >
> > I acknowledge that the authors have answered my main concerns and also confirmed the limitations of the work. I hence keep my score as it is.

---

> > > ### Author Response · Authors · 2026-04-08
> > >
> > > We sincerely thank the reviewer for the very positive assessment and for carefully reading our rebuttal. We are glad that our responses have adequately addressed the main concerns and that the limitations of the current work are now more clearly understood.
> > >
> > > We would just like to briefly clarify one point: the current paper focuses on the **classical scalar-latent normal mean-variance mixture** setting, which is already a broad and widely used framework for modeling **asymmetric** and **heavy-tailed** data. More general extensions, such as **multi-dimensional latent**, are in principle possible within the same overall ECM framework, but would require substantially more involved numerical treatment and are therefore beyond the scope of the present paper rather than excluded by the methodology itself.
> > >
> > > We are grateful for the reviewer’s positive evaluation, and we hope that this clarified understanding of the paper will be taken into account in the final evaluation.

---

### Official Review · Reviewer_faoF · 2026-03-13

**Soundness:** 3
**Presentation:** 3
**Significance:** 3
**Originality:** 2
**Overall Recommendation:** 5
**Confidence:** 4

**Summary:**

The paper studies maximum likelihood estimation from quantized observations when the latent unquantized data follow a normal mean-variance mixture model. This extends the usual Gaussian setting to a richer family that can capture heavy tails and skewness. The main methodological contribution is an ECM-based estimation procedure that derives a surrogate objective for the quantized likelihood, and obtains explicit conditional maximization updates for the location, skewness, and scatter parameters. The paper also provides a convergence analysis claiming strict concavity in each parameter block and linear convergence of the ECM iterations and shows how the same framework can be adapted to quantized linear regression, matrix completion, and compressive sensing. Overall, the paper presents a coherent and reasonably broad framework for robust quantized likelihood-based estimation.

**Compliance With Llm Reviewing Policy:**

Affirmed.

**Final Justification:**

All the technical questions have been resolved, the only question that remains concerns scope and relevance. Here I argue in favor of a balanced view: As I stated in the review, I do not consider the paper to be a breakthrough. However, I still recommend acceptance, as the general problem of theoretically understanding quantization effects in machine learning is very timely in general and moving slowly.
In particular, most works rely on a lot of simplifying assumptions, such as the Gaussian distribution. Gradually removing or generalizing these is very important towards a unified understanding, and I feel that this paper contributes to that.

**Key Questions For Authors:**

1. The derivation of equations (10)-(13) would benefit from a substantially more explicit presentation. At present, this part of the paper is difficult to follow: for example, equation (10) is attributed to Jensen's inequality, but there appear to be several intermediate manipulations and notation conventions that are not made fully transparent. Please include a detailed derivation of equations (10)-(13) in the response letter and also place it in an appendix of the camera-ready version with a reference to it in the main text. It would also help to clarify the notation for conditional distributions and expectations, such as $x_t \mid y_t$, when these expressions are first introduced, since the meaning of these objects is not immediately clear to the reader in the current presentation.

2. The E-step requires the computation of conditional expectations under mixed distributions, while many of the technical details are deferred to the appendix. Could the authors include a brief discussion in the main text about the computational cost and numerical stability of these calculations, especially as the ambient dimension or the number of quantization levels increases? A short clarification here would help readers better assess the practical scalability of the method.

3. The paper motivates the normal mean-variance mixture framework as a more robust alternative to Gaussian modeling. Could the authors comment more explicitly on the regimes in which this added flexibility is most beneficial in practice---for example, in the presence of heavy-tailed noise, skewed latent data, low-bit quantization, or model mismatch? A sharper discussion here would make it easier to understand when the proposed framework offers a substantial advantage over Gaussian-based approaches.

4. Since the proposed method is based on an ECM procedure in a latent-variable setting, could the authors comment on the sensitivity of the algorithm to initialization and on its local behavior in practice? In particular, it would be helpful to know whether multiple starts were needed in the experiments, whether the method was generally robust to initialization, and whether there were settings in which convergence depended noticeably on the starting point.

5. The discussion of one-bit identifiability is appreciated, but it would be helpful if the authors could state more explicitly what normalization or parameter constraints are imposed in practice to make the estimation problem well posed, both in theory and in experiments. A clearer explanation would make the one-bit case easier to interpret and would increase confidence that the reported results are directly comparable and not dependent on implicit conventions.

Update after rebuttal: The authors have sufficiently answered my questions.

**Limitations:**

This is a theoretical/algorithmic work, so I do not see the need to add such a discussion.

**Strengths And Weaknesses:**

{\bf Soundness:} The paper is technically solid at a high level and the proposed method is well motivated. The latent-variable formulation is natural for the normal mean-variance mixture family, and the resulting ECM procedure is appealing because it converts a difficult quantized likelihood problem into a sequence of more explicit updates. In particular, the surrogate formulation and the closed-form CM updates for $\mu$, $\xi$, and $\Sigma$ give the method a concrete algorithmic structure rather than leaving optimization entirely to generic numerical routines. The theoretical discussion also strengthens the paper, especially the treatment of blockwise concavity and convergence behavior. While some details of the E-step computations are deferred to the appendix, I did not see obvious conceptual gaps in the main development. Overall, the method appears sound and carefully constructed.

\textbf{Presentation.} The paper is generally well written and organized. The motivation is clear, the progression from model setup to algorithm is natural, and the paper does a good job of communicating the central message: namely, that one can go beyond Gaussian assumptions in quantized maximum likelihood estimation without losing a reasonably structured optimization procedure. I also appreciated that the paper connects the general estimation framework to several downstream problems, which helps the reader see the broader relevance of the approach. The exposition does become somewhat notation-heavy in the E-step and surrogate derivation, and a few passages could be made easier to parse for readers who are not already comfortable with latent-variable methods. The convergence section would benefit from a minor editorial clarification, as Proposition~1 is referenced without a clearly visible statement, and Proposition~2 uses $\mu_\ast$, $\xi_\ast$, and $\Sigma_\ast$ without explicitly defining their role. Still, on the whole I found the paper clear and professionally presented.

\textbf{Significance.} The paper addresses a relevant problem, and the proposed extension from Gaussian models to the normal mean-variance mixture family is meaningful rather than cosmetic. Quantized observations arise naturally in many modern settings, and robustness to skewness and heavy tails can matter in practice. The fact that the same framework extends to regression, matrix completion, and compressive sensing makes the contribution broader and more useful than a narrowly specialized methodological note. I would not describe the paper as a major breakthrough, but I do think it offers a solid and potentially valuable step forward for quantized likelihood methods.

\textbf{Originality.} There is genuine novelty in the way the paper combines quantized maximum likelihood estimation with the normal mean-variance mixture family and derives a unified ECM treatment with explicit updates. That is a worthwhile contribution. At the same time, the underlying ingredients---latent-variable modeling, mixture-based robustness, and ECM-style optimization---are themselves well established. For that reason, the paper feels more like a thoughtful and useful synthesis and extension than a highly original conceptual leap. In my view, this still reflects positively on the paper, but it places the novelty in the incremental rather than groundbreaking category.

---

> ### Author Rebuttal · Authors · 2026-03-31
>
> **Response to Weakness.** We thank the reviewer for noting these omissions. Proposition 1 should be revised to Theorem 1. Moreover, $\\boldsymbol{\\mu}^ {\\star}$, $\\boldsymbol{\\xi}^ {\\star}$, and $\\boldsymbol{\\varSigma}^ {\\star}$ denote the stationary points of the original problem. We will correct these in the final version.
>
> **Response to Q1.** We appreciate the reviewer's concern about the E-step and surrogate derivation. Below we detail the derivation of (10)–(13). Consider the log-likelihood in (6). Since $\\boldsymbol{y}$ depends on $\\boldsymbol{x}$ and both are random variables, we have
>
> $$
> \\begin{aligned}
> L(\\boldsymbol{\\theta})
> &=\\sum_ {t=1}^ n\\log\\int_ {\\mathbb{R}} p(\\boldsymbol{y}_ {t},\\boldsymbol{x}_ {t};\\boldsymbol{\\theta})\\,\\mathrm{d}\\boldsymbol{x}_ {t} \\\\
> &=\\sum_ {t=1}^ n\\log\\int_ {\\mathbb{R}} \\frac{p(\\boldsymbol{x}_ {t}\\mid \\boldsymbol{y}_ {t};\\underline{\\boldsymbol{\\theta}})}{p(\\boldsymbol{x}_ {t}\\mid \\boldsymbol{y}_ {t};\\underline{\\boldsymbol{\\theta}})} p(\\boldsymbol{y}_ {t},\\boldsymbol{x}_ {t};\\boldsymbol{\\theta})\\,\\mathrm{d}\\boldsymbol{x}_ {t} \\\\
> &=\\sum_ {t=1}^ n\\log\\int_ {\\mathbb{R}} \\mathsf{E}_ {\\boldsymbol{x}_ {t}\\mid \\boldsymbol{y}_ {t};\\underline{\\boldsymbol{\\theta}}}
> \\frac{p(\\boldsymbol{y}_ {t},\\boldsymbol{x}_ {t};\\boldsymbol{\\theta})}{p(\\boldsymbol{x}_ {t}\\mid \\boldsymbol{y}_ {t};\\underline{\\boldsymbol{\\theta}})}\\,\\mathrm{d}\\boldsymbol{x}_ {t} \\\\
> &\\geq \\sum_ {t=1}^ n\\int_ {\\mathbb{R}} \\mathsf{E}_ {\\boldsymbol{x}_ {t}\\mid \\boldsymbol{y}_ {t};\\underline{\\boldsymbol{\\theta}}}
> \\log\\frac{p(\\boldsymbol{y}_ {t},\\boldsymbol{x}_ {t};\\boldsymbol{\\theta})}{p(\\boldsymbol{x}_ {t}\\mid \\boldsymbol{y}_ {t};\\underline{\\boldsymbol{\\theta}})}\\,\\mathrm{d}\\boldsymbol{x}_ {t} \\\\
> &=\\sum_ {t=1}^ {n}\\mathsf{E}_ {\\boldsymbol{x}_ {t}\\mid \\boldsymbol{y}_ {t};\\underline{\\boldsymbol{\\theta}}}\\log p(\\boldsymbol{y}_ {t},\\boldsymbol{x}_ {t};\\boldsymbol{\\theta})+\\mathrm{const.}
> \\end{aligned}
> $$
>
> The inequality follows from Jensen's inequality: $\\log \\mathbb{E}[X] \\geq \\mathbb{E}[\\log X]$. Since $\\boldsymbol{x}$ depends on $z$ and both are random variables, the same argument yields (12). By definition of a normal mean-variance mixture, $\\boldsymbol{x} \\mid z$ is Gaussian, which gives (13).
>
> **Response to Q2.** Since $\\boldsymbol{y}$ is discrete, each iteration depends on at most $n_ 0=\\min\\{(e+1)^ d,n\\}$ distinct samples. The overall per-iteration complexity is $\\mathcal{O}(n_ 0 K d^ 3)$, linear in $n_ 0$. Hence, the algorithm remains efficient for large datasets, especially when coarse quantization or low dimensionality makes $n_ 0$ nearly independent of $n$.
> For numerical stability, the main error comes from evaluating the CDF of the normal mean-variance mixture distribution. The Gauss-Legendre quadrature error is $\\mathcal{O}(e^{-K})$. For $d > 3$, the multivariate normal CDF is computed numerically, and the quasi-Monte Carlo error is bounded by $\\mathcal{O}(N^{-1}\\log^ d N)$.
>
> **Response to Q3.** Thank you for the comment. The normal mean-variance mixture framework is most useful when data deviate from Gaussian assumptions: (1) heavy-tailed noise or outliers, where Gaussian models underestimate large deviations; (2) skewed or asymmetric distributions that symmetric Gaussian priors cannot capture; and (3) quantized observations at different bit depths.
> A recommender system is one example. User ratings are quantized preferences (e.g., positive/negative as 1-bit, or a 1-5 scale as $\\log_ 2 5$-bit). In practice, ratings and latent factors often exhibit asymmetry (users skew toward high ratings) and heavy tails (due to heterogeneity in user activity). Gaussian models are therefore too restrictive, while the mixture formulation captures skewness and tail behavior. We will clarify these scenarios in the revision.
>
> **Response to Q4.** To assess sensitivity to initialization, we ran the ECM algorithm under the generalized hyperbolic distribution assumption from 50 random starting points and recorded the converged objective values. As shown in https://anonymous.4open.science/api/repo/afoed_2131/file/boxplot.pdf?v=76ebadab, most runs converged to nearly the same objective value, while only 4 of 50 ended at noticeably inferior values. This suggests that the proposed method is generally robust to initialization in practice.
>
> **Response to Q5.** Identifiability is inherent under one-bit quantization. To address this, existing one-bit Gaussian estimation methods typically fix one parameter, either the location parameter or the scatter matrix, to identify the other. We adopt the same strategy here. Since this work focuses on algorithm design, we do not further study identifiability treatments. However, the normalization or parameter constraints suggested by the reviewer can be incorporated into our framework.
>
> Due to the page limit, additional details will be included in the final version.

---

> > ### Author Rebuttal · Reviewer_faoF · 2026-04-04
> >
> > Thank you for the detailed rebuttal. The clarification regarding the notation issue in the convergence section is helpful; the correction from Proposition 1 to Theorem 1, as well as the clarification that $\mu^*$, $\xi^*$, and $\Sigma^*$ denote stationary points of the original problem, resolves my concern there.
> >
> >
> > The response to Q1 is also satisfactory. The added derivation makes the intended Jensen-based construction of the E-step surrogate much clearer, and I now understand the logic behind equations (10)--(13) at a sufficient level. I think this concern is adequately addressed, although I would still encourage the authors to include a cleaner and slightly more polished version of this derivation in the appendix of the final paper.
> >
> >
> > I also find the responses to Q3, Q4 and Q5 acceptable. The discussion in Q3 helps clarify the regimes where the normal mean-variance mixture framework is expected to provide practical advantages over Gaussian modeling. Regarding Q4, the additional information is helpful in principle, and the reported experiments suggest that the method may be reasonably robust to initialization in practice. The clarification in Q5 about the one-bit identifiability issue and the normalization strategy adopted from prior one-bit Gaussian methods is also sufficient for me.
> >
> >
> > The only point where I still think a bit more explanation would be valuable is Q2. The rebuttal usefully indicates that the per-iteration complexity depends on the number of distinct quantized samples and that the main numerical difficulty lies in evaluating CDF-type quantities. However, I think the final version would benefit from a slightly more explicit explanation of the quantities $n_0$, $K$, and $N$, the justification for the bound $n_0=\min\{(e+1)^d,n\}$, and a brief indication of where the $d^3$ dependence arises. A short practical discussion of how these quantities are chosen and how runtime behaves as the dimension or number of quantization levels increases would make the scalability discussion easier to assess.
> >
> >
> > Overall, the rebuttal addresses most of my concerns, and I consider Q1, Q3, Q4 and Q5 resolved. Q2 is directionally helpful, but I would still encourage the authors to expand on that point more in their answer.

---

> > > ### Author Response · Authors · 2026-04-06
> > >
> > > We sincerely thank the reviewer for the very positive assessment and for carefully reading our rebuttal. We are glad that the reviewer now considers Q1, Q3, Q4, and Q5 resolved. We also appreciate the suggestion regarding Q2. We will revise this part accordingly.
> > >
> > > **On computational complexity.**
> > > Since the proposed method is iterative, we analyze its per-iteration complexity. Let $n$ denote the total number of observed quantized samples, and let $n_0$ denote the number of *distinct* quantized sample patterns. Because $\boldsymbol{y}$ takes discrete values, parameter updates only need to consider these $n_0$ distinct samples. If each coordinate is quantized into $e+1$ output intervals, the total number of possible quantized vectors is at most $(e+1)^d$. Therefore,
> > > $$
> > > n_0 \le \min\\{(e+1)^d, n\\}.
> > > $$
> > > Here, $(e+1)^d$ follows because each of the $d$ coordinates has at most $e+1$ quantized outcomes, while the minimum with $n$ reflects that distinct observed samples cannot exceed the dataset size. When quantization is coarse or the dimension is moderate, many observations collapse to the same quantized pattern, so **$n_0$ can be much smaller than $n$**.
> > >
> > > The per-iteration complexity is dominated by two operations:
> > > *   (i) **$K$-node Gauss-Legendre quadrature** for numerical integration over the mixing variable, where $K$ denotes the number of quadrature nodes ($K=20$ in practice);
> > > *   (ii) **Multivariate normal CDF evaluation** via `mvncdf`, costing $\mathcal{O}(d^3)$ per call due to operations such as Cholesky decomposition of a $d\times d$ covariance matrix.
> > >
> > > Evaluating $p(\boldsymbol{y};\boldsymbol{\theta})$ for $n_0$ distinct samples therefore costs
> > > $$
> > > \mathcal{O}(n_0 K d^3 + K^2 + d^2).
> > > $$
> > >
> > > *   **For updating $\boldsymbol{\mu}$** (eq. (43)), computing $p(\boldsymbol{y};\underline{\boldsymbol{\theta}})$ and $p_{-1}(\boldsymbol{y};\boldsymbol{\theta})$ each costs $\mathcal{O}(n_0 K d^3 + K^2 + d^2)$; computing $\boldsymbol{q}$ via $p(x_i,\boldsymbol{y}_{\backslash i};\boldsymbol{\theta})$ costs $\mathcal{O}(n_0(Kd^3+d^3)+K^2+d^2)$; and the final update costs $\mathcal{O}(d^2)$. The update of $\boldsymbol{\xi}$ (eq. (50)) has the same complexity.
> > > *   **For updating $\boldsymbol{\varSigma}$** (eq. (57)), naive construction of $\boldsymbol{H}_ 1$ and $\boldsymbol{D}_ 1$ requires evaluating $\mathcal{O}(d^2)$ entries per sample, each with cost $\mathcal{O}(Kd^3)$, leading to $\mathcal{O}(n_0 K d^5)$ overall. By using the numerical differentiation (Lemma 10 of Appendix B.3), this is reduced to $\mathcal{O}(n_0 d^3)$. The remaining steps—computing $\boldsymbol{D}$, the matrix product
> > > $$
> > > \frac{1}{n}\sum_ {t} p^{-1}(\boldsymbol{y}_ t)\underline{\boldsymbol{\varSigma}}(\boldsymbol{H}_ {1,t}+\boldsymbol{D}_ {1,t})\underline{\boldsymbol{\varSigma}},
> > > $$
> > > and the update of $\boldsymbol{\varSigma}^+$—cost $\mathcal{O}(n_0(Kd^3+d^3)+K^2+d^2)$, $\mathcal{O}(n_0 d^3)$, and $\mathcal{O}(n_0 d^2)$, respectively.
> > >
> > > Combining all steps, the total per-iteration complexity is
> > > $$
> > > \mathcal{O}(n_0 K d^3).
> > > $$
> > > Since $K$ is fixed in practice, the method **scales essentially linearly in $n_0$**. This is particularly favorable for large datasets with coarse quantization, where $n_0$ may be effectively independent of $n$. Runtime increases with $d$ through $\mathcal{O}(d^3)$ covariance operations, and may also increase with $e$, since finer quantization generally increases the number of distinct observed patterns.
> > >
> > > To illustrate how per-iteration runtime scales with $d$ and $e$, we include numerical runtime experiments in "d_versus_time.pdf" and "e_versus_time.pdf" at https://anonymous.4open.science/r/afoed_2131.
> > >
> > > **On numerical stability.**
> > > Numerical error mainly arises from two sources. First, the $K$-node Gauss--Legendre quadrature truncation error decays exponentially as $\mathcal{O}(e^{-K})$, so $K=20$ already provides very high accuracy. Second, for multivariate normal CDF evaluation: when $d\leq 3$, it is computed analytically; when $d>3$, `mvncdf` uses a quasi-Monte Carlo routine with error bounded by $\mathcal{O}(N^{-1}\log^d N)$, where $N$ is the number of internal quasi-Monte Carlo samples. The default tolerance in `mvncdf` was sufficient to ensure stable convergence of the ECM iterations.
> > >
> > > Following the reviewer's suggestion, we will add a brief discussion of the above points in the final version. We again sincerely thank the reviewer for the positive assessment and careful reading of our paper. If our response has adequately addressed the remaining concern, we would sincerely appreciate it if this could be reflected in the final score.

---

### Decision · Program_Chairs · 2026-04-30

**Decision:**

Accept (regular)

**Comment:**

The paper is technically sound and well executed. The reviewers were largely positive about the ECM development, the convergence analysis, and the breadth of the normal mean-variance mixture framework. The main reservation is about novelty: this is not a conceptual breakthrough so much as a careful and nontrivial extension of classical EM/ECM methodology to a broader and more useful quantized estimation setting. I agree with that characterization. Still, I think the scope limitations here are acceptable. Overall, while the novelty is somewhat mild, the paper provides a solid technical advance on an important problem, and I recommend accept.